# Online Clustering of Bandits with Misspecified User Models

**Zhiyong Wang**
The Chinese University of Hong Kong
zywang21@cse.cuhk.edu.hk

**Jize Xie**
Shanghai Jiao Tong University
xjzzjl@sjtu.edu.cn

**Xutong Liu**
The Chinese University of Hong Kong
liuxt@cse.cuhk.edu.hk

**Shuai Li**[*]
Shanghai Jiao Tong University
shuaili8@sjtu.edu.cn

**John C.S. Lui**
The Chinese University of Hong Kong
cslui@cse.cuhk.edu.hk

## Abstract

The contextual linear bandit is an important online learning problem where given arm features, a learning agent selects an arm at each round to maximize the cumulative rewards in the long run. A line of works, called the clustering of bandits (CB), utilize the collaborative effect over user preferences and have shown significant improvements over classic linear bandit algorithms. However, existing CB algorithms require well-specified linear user models and can fail when this critical assumption does not hold. Whether robust CB algorithms can be designed for more practical scenarios with misspecified user models remains an open problem. In this paper, we are the first to present the important problem of clustering of bandits with misspecified user models (CBMUM), where the expected rewards in user models can be perturbed away from perfect linear models. We devise two robust CB algorithms, RCLUMB and RSCLUMB (representing the learned clustering structure with dynamic graph and sets, respectively), that can accommodate the inaccurate user preference estimations and erroneous clustering caused by model misspecifications. We prove regret upper bounds of $O(\epsilon_* T\sqrt{md\log T} + d\sqrt{mT}\log T)$ for our algorithms under milder assumptions than previous CB works (notably, we move past a restrictive technical assumption on the distribution of the arms), which match the lower bound asymptotically in $T$ up to logarithmic factors, and also match the state-of-the-art results in several degenerate cases. The techniques in proving the regret caused by misclustering users are quite general and may be of independent interest. Experiments on both synthetic and real-world data show our outperformance over previous algorithms.

## 1 Introduction

Stochastic multi-armed bandit (MAB) [2, 4, 22] is an online sequential decision-making problem, where the learning agent selects an action and receives a corresponding reward at each round, so as to maximize the cumulative reward in the long run. MAB algorithms have been widely applied in

---

[*]Corresponding author.

37th Conference on Neural Information Processing Systems (NeurIPS 2023).

recommendation systems and computer networks to handle the exploration and exploitation trade-off [20, 30, 38, 5].

To deal with large-scale applications, the contextual linear bandits [24, 9, 1, 29, 21] have been studied, where the expected reward of each arm is assumed to be perfectly linear in their features. Leveraging the contextual side information about the user and arms, linear bandits can provide more personalized recommendations [16]. Classical linear bandit approaches, however, ignore the often useful tool of collaborative filtering. To utilize the relationships among users, the problem of clustering of bandits (CB) has been proposed [12]. Specifically, CB algorithms adaptively partition users into clusters and utilize the collaborative effect of users to enhance learning performance.

Although existing CB algorithms have shown great success in improving recommendation qualities, there exist two major limitations. First, all previous works on CB [12, 25, 27, 39] assume that for each user, the expected rewards follow a *perfectly linear* model with respect to the user preference vector and arms' feature vectors. In many real-world scenarios, due to feature noises or uncertainty [15], the reward may not necessarily conform to a perfectly linear function, or even deviates a lot from linearity [14]. Second, previous CB works assume that for users within the same cluster, their preferences are exactly the same. Due to the heterogeneity in users' personalities and interests, similar users may not have identical preferences, invalidating this strong assumption.

To address these issues, we propose a novel problem of clustering of bandits with misspecified user models (CBMUM). In CBMUM, the expected reward model of each user does not follow a perfectly linear function but with possible additive deviations. We assume users in the same underlying cluster share a common preference vector, meaning they have the same linear part in reward models, but the deviation parts are allowed to be different, better reflecting the varieties of user personalities.

The relaxation of perfect linearity and the reward homogeneity within the same cluster bring many challenges to the CBMUM problem. In CBMUM, we not only need to handle the uncertainty from the *unknown* user preference vectors, but also have to tackle the additional uncertainty from model misspecifications. Due to such uncertainties, it becomes highly challenging to design a robust algorithm that can cluster the users appropriately and utilize the clustered information judiciously. On the one hand, the algorithm needs to be more tolerant in the face of misspecifications so that more similar users can be clustered together to utilize the collaborative effect. On the other hand, it has to be more selective to rule out the possibility of *misclustering* users with large preference gaps.

## 1.1 Our Contributions

This paper makes the following four contributions.

**New Model Formulation.** We are the first to formulate the clustering of bandits with misspecified user models (CBMUM) problem, which is more practical by removing the perfect linearity assumption in previous CB works.

**Novel Algorithm Designs.** We design two novel algorithms, RCLUMB and RSCLUMB, which robustly learn the clustering structure and utilize this collaborative information for faster user preference elicitation. Specifically, RCLUMB keeps updating a dynamic graph over all users, where users connected directly by edges are supposed to be in the same cluster. RCLUMB adaptively removes edges and recommends items based on historical interactions. RSCLUMB represents the clustering structure with sets, which are dynamicly merged and split during the learning process. Due to the page limit, we only illustrate the RCLUMB algorithm in the main paper. We leave the exposition, illustration, and regret analysis of the RSCLUMB algorithm in Appendix K.

To overcome the challenges brought by model misspecifications, we do the following key steps in the RCLUMB algorithm. (i) To ensure that with high probability, similar users will not be partitioned apart, we design a more tolerant edge deletion rule by taking model misspecifications into consideration. (ii) Due to inaccurate user preference estimations caused by model misspecifications, trivially following previous CB works [12, 25, 28] to directly use connected components in the maintained graph as clusters would *miscluster* users with big preference gaps, causing a large regret. To be discriminative in cluster assignments, we filter users directly linked with the current user in the graph to form the cluster used in this round. With these careful designs of (i) and (ii), we can guarantee that with high probability, information of all similar users can be leveraged, and only users with close enough preferences might be *misclustered*, which will only mildly impair the

learning accuracy. Additionally: (iii) we design an enlarged confidence radius to incorporate both the exploration bonus and the additional uncertainty from misspecifications when recommending arms. The design of RSCLUMB follows similar ideas, which we leave in the Appendix K due to page limit.

**Theoretical Analysis with Milder Assumptions**. We prove regret upper bounds for our algorithms of $O(\epsilon_* T\sqrt{md\log T} + d\sqrt{mT}\log T)$ in CBMUM under much milder and practical assumptions (in arm generation distribution) than previous CB works, which match the state-of-the-art results in degenerate cases. Our proof is quite different from the typical proof flow of previous CB works (details in Appendix C). One key challenge is to bound the regret caused by *misclustering* users with close but not the same preference vectors and use the inaccurate cluster-based information to recommend arms. To handle the challenge, we prove a key lemma (Lemma 5.7) to bound this part of regret. We defer its details in Section 5 and Appendix G. The techniques and results for bounding this part are quite general and may be of independent interest. We also give a regret lower bound of $\Omega(\epsilon_* T\sqrt{d})$ for CBMUM, showing that our upper bounds are asymptotically tight with respect to $T$ up to logarithmic factors. We leave proving a tighter lower bound for CBMUM as an open problem.

**Good Experimental Performance.** Extensive experiments on both synthetic and real-world data show the advantages of our proposed algorithms over the existing algorithms.

## 2   Related Work

Our work is closely related to two lines of research: online clustering of bandits (CB) and misspecified linear bandits (MLB). More discussions on related works can be found in Appendix A.

The paper [12] first formulates the CB problem and proposes a graph-based algorithm. The work [26] further considers leveraging the collaborative effects on items to guide the clustering of users. The work [25] considers the CB problem in the cascading bandits setting with random prefix feedback. The paper [27] also considers users with different arrival frequencies. A recent work [28] proposes the setting of clustering of federated bandits, considering both privacy protection and communication requirements. However, all these works assume that the reward model for each user follows a perfectly linear model, which is unrealistic in many real-world applications. To the best of our knowledge, this paper is the first work to consider user model misspecifications in the CB problem.

The work [14] first proposes the misspecified linear bandits (MLB) problem, shows the vulnerability of linear bandit algorithms under deviations, and designs an algorithm RLB that is only robust to non-sparse deviations. The work [23] proposes two algorithms to handle general deviations, which are modifications of the phased elimination algorithm [22] and LinUCB [1]. Some recent works [31, 11] use model selection methods to deal with unknown exact maximum model misspecification level. Note that the work [11] has an additional assumption on the access to an online regression oracle, and the paper [31] still needs to know an upper bound of the unknown exact maximum model deviation level. None of them consider the CB setting with multiple users, thus differing from ours.

We are the first to initialize the study of the important CBMUM problem, and propose a general framework for dealing with model misspecifications in CB problems. Our study is based on fundamental models on CB [12, 27] and MLB [23], the algorithm design ideas and theoretical analysis are pretty general. We leave incorporating the model selection methods [31, 11] into our framework to address the unknown exact maximum model misspecification level as an interesting future work.

## 3   Problem Setup

This section formulates the problem of "clustering of bandits with misspecified user models" (CB-MUM). We use boldface **lowercase** and boldface **CAPITALIZED** letters for vectors and matrices. We use $|\mathcal{A}|$ to denote the number of elements in $\mathcal{A}$, $[m]$ to denote $\{1, \ldots, m\}$, and $\|\boldsymbol{x}\|_{\boldsymbol{M}} = \sqrt{\boldsymbol{x}^\top \boldsymbol{M} \boldsymbol{x}}$ to denote the matrix norm of vector $\boldsymbol{x}$ regarding the positive semi-definite (PSD) matrix $\boldsymbol{M}$.

In CBMUM, there are $u$ users denoted by $\mathcal{U} = \{1, 2, \ldots, u\}$. Each user $i \in \mathcal{U}$ is associated with an *unknown* preference vector $\boldsymbol{\theta}_i \in \mathbb{R}^d$, with $\|\boldsymbol{\theta}_i\|_2 \leq 1$. We assume there is an *unknown* underlying clustering structure over users representing the similarity of their behaviors. Specifically, $\mathcal{U}$ can be partitioned into a small number $m$ (i.e., $m \ll u$) clusters, $V_1, V_2, \ldots V_m$, where $\cup_{j \in [m]} V_j = \mathcal{U}$, and $V_j \cap V_{j'} = \emptyset$, for $j \neq j'$. We call these clusters *ground-truth clusters* and use $\mathcal{V} = \{V_1, V_2, \ldots, V_m\}$ to denote the set of these clusters. Users in the same *ground-truth cluster* share the same preference vector, while users from different *ground-truth clusters* have different preference vectors. Let $\boldsymbol{\theta}^j$

denote the common preference vector for $V_j$ and $j(i) \in [m]$ denote the index of the *ground-truth cluster* that user $i$ belongs to. For any $\ell \in \mathcal{U}$, if $\ell \in V_{j(i)}$, then $\boldsymbol{\theta}_\ell = \boldsymbol{\theta}_i = \boldsymbol{\theta}^{j(i)}$.

At each round $t \in [T]$, a user $i_t \in \mathcal{U}$ comes to be served. The learning agent receives a finite arm set $\mathcal{A}_t \subseteq \mathcal{A}$ to choose from (with $|\mathcal{A}_t| \leq C, \forall t$), where each arm $a \in \mathcal{A}$ is associated with a feature vector $\boldsymbol{x}_a \in \mathbb{R}^d$, and $\|\boldsymbol{x}_a\|_2 \leq 1$. The agent assigns an appropriate cluster $\overline{V}_t$ for user $i_t$ and recommends an item $a_t \in \mathcal{A}_t$ based on the aggregated historical information gathered from cluster $\overline{V}_t$. After receiving the recommended item $a_t$, user $i_t$ gives a random reward $r_t \in [0,1]$ to the agent. To better model real-world scenarios, we assume that the reward $r_t$ follows a misspecified linear function of the item feature vector $\boldsymbol{x}_{a_t}$ and the *unknown* user preference vector $\boldsymbol{\theta}_{i_t}$. Formally,

$$r_t = \boldsymbol{x}_{a_t}^\top \boldsymbol{\theta}_{i_t} + \boldsymbol{\epsilon}_{a_t}^{i_t,t} + \eta_t\,, \tag{1}$$

where $\boldsymbol{\epsilon}^{i_t,t} = [\boldsymbol{\epsilon}_1^{i_t,t}, \boldsymbol{\epsilon}_2^{i_t,t}, \ldots, \boldsymbol{\epsilon}_{|\mathcal{A}_t|}^{i_t,t}]^\top \in \mathbb{R}^{|\mathcal{A}_t|}$ denotes the *unknown* deviation in the expected rewards of arms in $\mathcal{A}_t$ from linearity for user $i_t$ at $t$, and $\eta_t$ is the 1-sub-Gaussian noise. We allow the deviation vectors for users in the same *ground-truth cluster* to be different.

We assume the clusters, users, items, and model misspecifications satisfy the following assumptions.

**Assumption 3.1** (Gap between different clusters). The gap between any two preference vectors for different *ground-truth clusters* is at least an *unknown* positive constant $\gamma$

$$\left\|\boldsymbol{\theta}^j - \boldsymbol{\theta}^{j'}\right\|_2 \geq \gamma > 0\,, \forall j, j' \in [m]\,, j \neq j'\,.$$

**Assumption 3.2** (Uniform arrival of users). At each round $t$, a user $i_t$ comes uniformly at random from $\mathcal{U}$ with probability $1/u$, independent of the past rounds.

**Assumption 3.3** (Item regularity). At each time step $t$, the feature vector $\boldsymbol{x}_a$ of each arm $a \in \mathcal{A}_t$ is drawn independently from a fixed but unknown distribution $\rho$ over $\{\boldsymbol{x} \in \mathbb{R}^d : \|\boldsymbol{x}\|_2 \leq 1\}$, where $\mathbb{E}_{\boldsymbol{x} \sim \rho}[\boldsymbol{x}\boldsymbol{x}^\top]$ is full rank with minimal eigenvalue $\lambda_x > 0$. Additionally, at any time $t$, for any fixed unit vector $\boldsymbol{\theta} \in \mathbb{R}^d$, $(\boldsymbol{\theta}^\top \boldsymbol{x})^2$ has sub-Gaussian tail with variance upper bounded by $\sigma^2$.

**Assumption 3.4** (Bounded misspecification level). We assume that there is a pre-specified maximum misspecification level parameter $\epsilon_*$ such that $\left\|\boldsymbol{\epsilon}^{i,t}\right\|_\infty \leq \epsilon_*, \forall i \in \mathcal{U}, t \in [T]$.

**Remark 1.** All these assumptions basically follow previous works on CB [12, 13, 25, 3, 28] and MLB [23]. Note that Assumption 3.3 is less stringent and more practical than previous CB works which also put restrictions on the variance upper bound $\sigma^2$. For Assumption 3.2, our results can easily generalize to the case where the user arrival follows any distributions with minimum arrival probability greater than $p_{min}$. For Assumption 3.4, note that $\epsilon_*$ can be an upper bound on the maximum misspecification level, not the exact maximum itself. In real-world applications, the deviations are usually small [14], and we can set a relatively big $\epsilon_*$ as an upper bound. For more discussions please refer to Appendix B

Let $a_t^* \in \arg\max_{a \in \mathcal{A}_t} \boldsymbol{x}_a^\top \boldsymbol{\theta}_{i_t} + \boldsymbol{\epsilon}_a^{i_t,t}$ denote an optimal arm which gives the highest expected reward at $t$. The goal of the agent is to minimize the expected cumulative regret

$$R(T) = \mathbb{E}[\textstyle\sum_{t=1}^T (\boldsymbol{x}_{a_t^*}^\top \boldsymbol{\theta}_{i_t} + \boldsymbol{\epsilon}_{a_t^*}^{i_t,t} - \boldsymbol{x}_{a_t}^\top \boldsymbol{\theta}_{i_t} - \boldsymbol{\epsilon}_{a_t}^{i_t,t})]\,. \tag{2}$$

## 4  Algorithm

This section introduces our algorithm called "Robust CLUstering of Misspecified Bandits" (RCLUMB) (Algo.1). RCLUMB is a graph-based algorithm. The ideas and techniques of RCLUMB can be easily generalized to set-based algorithms. To illustrate this generalizability, we also design a set-based algorithm RSCLUMB. We leave the exposition and analysis of RSCLUMB in Appendix K.

For ease of interpretation, we define the coefficient

$$\zeta \triangleq 2\epsilon_* \sqrt{\frac{2}{\tilde{\lambda}_x}}\,, \tag{3}$$

where $\tilde{\lambda}_x \triangleq \int_0^{\lambda_x}(1 - e^{-\frac{(\lambda_x - x)^2}{2\sigma^2}})^C dx$. $\zeta$ is theoretically the minimum gap between two users' preference vectors that an algorithm can distinguish with high probability, as supported by Eq.(50) in the proof of Lemma H.1 in Appendix H. Note that the algorithm does not require knowledge of $\zeta$. We also make the following definition for illustration.

---

**Algorithm 1** Robust Clustering of Misspecified Bandits Algorithm (RCLUMB)

---

1: **Input:** Deletion parameter $\alpha_1, \alpha_2 > 0$, $f(T) = \sqrt{\frac{1+\ln(1+T)}{1+T}}$, $\lambda, \beta, \epsilon_* > 0$.
2: **Initialization:** $\boldsymbol{M}_{i,0} = 0_{d \times d}, \boldsymbol{b}_{i,0} = 0_{d \times 1}, T_{i,0} = 0$, $\forall i \in \mathcal{U}$; a complete Graph $G_0 = (\mathcal{U}, E_0)$ over $\mathcal{U}$.
3: **for all** $t = 1, 2, \ldots, T$ **do**
4:     Receive the index of the current user $i_t \in \mathcal{U}$, and the current feasible arm set $\mathcal{A}_t$;
5:     Filter user $i_t$ and users $i \in \mathcal{U}$ that are *directly* connected with user $i_t$ via edge $(i, i_t) \in E_{t-1}$, to form the cluster $\overline{V}_t$;
6:     Compute the estimated statistics for cluster $\overline{V}_t$

$$\overline{\boldsymbol{M}}_{\overline{V}_t, t-1} = \lambda \boldsymbol{I} + \sum_{i \in \overline{V}_t} \boldsymbol{M}_{i,t-1}, \overline{\boldsymbol{b}}_{\overline{V}_t, t-1} = \sum_{i \in \overline{V}_t} \boldsymbol{b}_{i,t-1}, \hat{\boldsymbol{\theta}}_{\overline{V}_t, t-1} = \overline{\boldsymbol{M}}_{\overline{V}_t, t-1}^{-1} \overline{\boldsymbol{b}}_{\overline{V}_t, t-1};$$

7:     Recommend an arm $a_t$ with the largest UCB index (Eq.(5)), and receive the reward $r_t \in [0,1]$;

8:     Update the statistics for user $i_t$ $\boldsymbol{M}_{i_t,t} = \boldsymbol{M}_{i_t,t-1} + \boldsymbol{x}_{a_t} \boldsymbol{x}_{a_t}^\top, \boldsymbol{b}_{i_t,t} = \boldsymbol{b}_{i_t,t-1} + r_t \boldsymbol{x}_{a_t}, T_{i_t,t} = T_{i_t,t-1} + 1, \hat{\boldsymbol{\theta}}_{i_t,t} = (\lambda \boldsymbol{I} + \boldsymbol{M}_{i_t,t})^{-1} \boldsymbol{b}_{i_t,t};$
9:     Keep the statistics of other users unchanged
    $\boldsymbol{M}_{\ell,t} = \boldsymbol{M}_{\ell,t-1}, \boldsymbol{b}_{\ell,t} = \boldsymbol{b}_{\ell,t-1}, T_{\ell,t} = T_{\ell,t-1}, \hat{\boldsymbol{\theta}}_{\ell,t} = \hat{\boldsymbol{\theta}}_{\ell,t-1}$, for all $\ell \in \mathcal{U}, \ell \neq i_t$;
10:    Delete the edge $(i_t, \ell) \in E_{t-1}$, if

$$\left\| \hat{\boldsymbol{\theta}}_{i_t,t} - \hat{\boldsymbol{\theta}}_{\ell,t} \right\|_2 \geq \alpha_1 \left( f(T_{i_t,t}) + f(T_{\ell,t}) \right) + \alpha_2 \epsilon_*,$$

    and get an updated graph $G_t = (\mathcal{U}, E_t)$;

---

**Definition 4.1** ($\zeta$-close users and $\zeta$-good clusters). Two users $i, i' \in \mathcal{U}$ are $\zeta$-close if $\|\boldsymbol{\theta}_i - \boldsymbol{\theta}_{i'}\|_2 \leq \zeta$. Cluster $\overline{V}$ is a $\zeta$-good cluster at time $t$, if $\forall i \in \overline{V}$, user $i$ and the coming user $i_t$ are $\zeta$-close.

We also say that two *ground-truth clusters* are "$\zeta$-close" if their preference vectors' gap is less than $\zeta$.

Now we introduce the process and intuitions of RCLUMB (Algo.1). The algorithm maintains an undirected user graph $G_t = (\mathcal{U}, E_t)$, where users are connected with edges if they are inferred to be in the same cluster. We denote the connected component in $G_{t-1}$ containing user $i_t$ at round $t$ as $\tilde{V}_t$.

**Cluster Detection.** $G_0$ is initialized to be a complete graph, and will be updated adaptively based on the interactive information. At round $t$, user $i_t \in \mathcal{U}$ comes to be served with a feasible arm set $\mathcal{A}_t$ (Line 4). Due to model misspecifications, it is impossible to cluster users with exactly the same preference vector $\boldsymbol{\theta}$, but similar users whose preference vectors are within the distance of $\zeta$. According to the proof of Lemma H.1, after a sufficient time, with high probability, any pair of users directly connected by an edge in $E_{t-1}$ are $\zeta$-close. However, if we trivially follow previous CB works [12, 25, 28] to directly use the connected component $\tilde{V}_t$ as the inferred cluster for user $i_t$ at round $t$, it will cause a large regret. The reason is that in the worst case, the preference vector $\boldsymbol{\theta}$ of the user in $\tilde{V}_t$ who is $h$-hop away from user $i_t$ could deviate by $h\zeta$ from $\boldsymbol{\theta}_{i_t}$, where $h$ can be as large as $|\tilde{V}_t|$. Based on this reasoning, our key point is to select the cluster $\overline{V}_t$ as the users at most 1-hop away from $i_t$ in the graph. In other words, after some interactions, $\overline{V}_t$ forms a $\zeta$-good cluster with high probability; thus, RCLUMB can avoid using misleading information from dissimilar users for recommendations.

**Cluster-based Recommendation.** After finding the appropriate cluster $\overline{V}_t$ for $i_t$, the agent estimates the common user preference vector based on the historical information associated with cluster $\overline{V}_t$ by

$$\hat{\boldsymbol{\theta}}_{\overline{V}_t, t-1} = \arg\min_{\boldsymbol{\theta} \in \mathbb{R}^d} \sum_{\substack{s \in [t-1] \\ i_s \in \overline{V}_t}} (r_s - \boldsymbol{x}_{a_s}^\top \boldsymbol{\theta})^2 + \lambda \|\boldsymbol{\theta}\|_2^2, \tag{4}$$

where $\lambda > 0$ is a regularization coefficient. Its closed-form solution is $\hat{\boldsymbol{\theta}}_{\overline{V}_t, t-1} = \overline{\boldsymbol{M}}_{\overline{V}_t, t-1}^{-1} \overline{\boldsymbol{b}}_{\overline{V}_t, t-1}$, where $\overline{\boldsymbol{M}}_{\overline{V}_t, t-1} = \lambda \boldsymbol{I} + \sum_{\substack{s \in [t-1] \\ i_s \in \overline{V}_t}} \boldsymbol{x}_{a_s} \boldsymbol{x}_{a_s}^\top$, $\overline{\boldsymbol{b}}_{\overline{V}_t, t-1} = \sum_{\substack{s \in [t-1] \\ i_s \in \overline{V}_t}} r_{a_s} \boldsymbol{x}_{a_s}$.

Based on this estimation, in Line 7, the agent recommends an arm using the UCB strategy

$$a_t = \arg\max_{a \in \mathcal{A}_t} \min\{1, \underbrace{\boldsymbol{x}_a^\top \hat{\boldsymbol{\theta}}_{\overline{V}_t, t-1}}_{\hat{R}_{a,t}} + \underbrace{\beta \|\boldsymbol{x}_a\|_{\overline{\boldsymbol{M}}_{\overline{V}_t, t-1}^{-1}} + \epsilon_* \sum_{\substack{s \in [t-1] \\ i_s \in \overline{V}_t}} \left| \boldsymbol{x}_a^\top \overline{\boldsymbol{M}}_{\overline{V}_t, t-1}^{-1} \boldsymbol{x}_{a_s} \right|}_{C_{a,t}} \}, \tag{5}$$

where $\beta = \sqrt{\lambda} + \sqrt{2\log(\frac{1}{\delta}) + d\log(1 + \frac{T}{\lambda d})}$, $\hat{R}_{a,t}$ denotes the estimated reward of arm $a$ at $t$, $C_{a,t}$ denotes the confidence radius of arm $a$ at round $t$.

Due to deviations from linearity, the estimation $\hat{R}_{a,t}$ computed by a linear function is no longer accurate. To handle the estimation uncertainty of model misspecifications, we design an enlarged confidence radius $C_{a,t}$. The first term of $C_{a,t}$ in Eq.(5) captures the uncertainty of online learning for the linear part, and the second term related to $\epsilon_*$ reflects the additional uncertainty from deviations from linearity. The design of $C_{a,t}$ theoretically relies on Lemma 5.6 which will be given in Section 5.

**Update User Statistics.** Based the feedback $r_t$, in Line 8 and 9, the agent updates the statistics for user $i_t$. Specifically, the agent estimates the preference vector $\boldsymbol{\theta}_{i_t}$ by

$$\hat{\boldsymbol{\theta}}_{i_t,t} = \underset{\boldsymbol{\theta} \in \mathbb{R}^d}{\arg\min} \sum\nolimits_{\substack{s \in [t] \\ i_s = i_t}} (r_s - \boldsymbol{x}_{a_s}^\top \boldsymbol{\theta})^2 + \lambda \|\boldsymbol{\theta}\|_2^2 \,, \tag{6}$$

with solution $\hat{\boldsymbol{\theta}}_{i_t,t} = (\lambda \boldsymbol{I} + \boldsymbol{M}_{i_t,t})^{-1} \boldsymbol{b}_{i_t,t}$, where $\boldsymbol{M}_{i_t,t} = \sum\nolimits_{\substack{s \in [t] \\ i_s = i_t}} \boldsymbol{x}_{a_s} \boldsymbol{x}_{a_s}^\top$, $\boldsymbol{b}_{i_t,t} = \sum\nolimits_{\substack{s \in [t] \\ i_s = i_t}} r_{a_s} \boldsymbol{x}_{a_s}$.

**Update the Graph $G_t$.** Finally, in Line 10, the agent verifies whether the similarities between user $i_t$ and other users are still true based on the updated estimation $\hat{\boldsymbol{\theta}}_{i_t,t}$. For every user $\ell \in \mathcal{U}$ connected with user $i_t$ via edge $(i_t, \ell) \in E_{t-1}$, if the gap between her estimated preference vector $\hat{\boldsymbol{\theta}}_{\ell,t}$ and $\hat{\boldsymbol{\theta}}_{i_t,t}$ is larger than a threshold supported by Lemma H.1, the agent will delete the edge $(i_t, \ell)$ to split them apart. The threshold in Line 10 is carefully designed, taking both estimation uncertainty in a linear model and deviations from linearity into consideration. As shown in the proof of Lemma H.1 (in Appendix H), using this threshold, with high probability, edges between users in the same *ground-truth clusters* will not be deleted, and edges between users that are not $\zeta$-close will always be deleted. Together with the filtering step in Line 5, with high probability, the algorithm will leverage all the collaborative information of similar users and avoid misusing the information of dissimilar users. The updated graph $G_t$ will be used in the next round.

## 5 Theoretical Analysis

In this section, we theoretically analyze the performance of the RCLUMB algorithm by giving an upper bound of the expected regret defined in Eq.(2). Due to the space limitation, we only show the main result (Theorem 5.3), key lemmas, and a sketched proof for Theorem 5.3. Detailed proofs, other technical lemmas, and the regret analysis of the RSLUMB algorithm can be found in the Appendix.

To state our main result, we first give two definitions as follows. The first definition is about the minimum separable gap constant $\gamma_1$ of a CBMUM problem instance.

**Definition 5.1** (Minimum separable gap $\gamma_1$). The minimum separable gap constant $\gamma_1$ of a CBMUM problem instance is the minimum gap over the gaps among users that are greater than $\zeta$ (Eq. (3))

$$\gamma_1 = \min\{\|\boldsymbol{\theta}_i - \boldsymbol{\theta}_\ell\|_2 : \|\boldsymbol{\theta}_i - \boldsymbol{\theta}_\ell\|_2 > \zeta, \forall i, \ell \in \mathcal{U}\}, \text{ with } \min \emptyset = \infty.$$

**Remark 2.** In CBMUM, the role of $\gamma_1 - \zeta$ is similar to that of $\gamma$ (given in Assumption 3.1) in the previous CB problem with perfectly linear models, quantifying the hardness of performing clustering on the problem instance. Intuitively, users are easier to cluster if $\gamma_1$ is larger, and the deduction of $\zeta$ shows the additional difficulty due to model diviations. If there are no misspecifications, i.e., $\zeta = 2\epsilon_*\sqrt{\frac{2}{\lambda_x}} = 0$, then $\gamma_1 = \gamma$, recovering the minimum separable gap between clusters in the classic CB problem [12, 25] without model misspecifications.

The second definition is about the number of "hard-to-cluster users" $\tilde{u}$.

**Definition 5.2** (Number of "hard-to-cluster users" $\tilde{u}$). The number of "hard-to-cluster users" $\tilde{u}$ is the number of users in the *ground-truth clusters* which are $\zeta$-close to some other *ground-truth cluster*s

$$\tilde{u} = \sum_{j \in [m]} |V_j| \times \mathbb{I}\{\exists j' \in [m], j' \neq j : \left\|\boldsymbol{\theta}^{j'} - \boldsymbol{\theta}^j\right\|_2 \leq \zeta\},$$

where $\mathbb{I}\{\cdot\}$ denotes the indicator function of the argument, $|V_j|$ denotes the number of users in $V_j$.

**Remark 3.** $\tilde{u}$ captures the number of users who belong to different *ground-truth clusters* but their gaps are less than $\zeta$. These users may be merged into one cluster by mistake and cause certain regret.

The following theorem gives an upper bound on the expected regret achieved by RCLUMB.

**Theorem 5.3** (Main result on regret bound). *Suppose that the assumptions in Section 3 are satisfied. Then the expected regret of the RCLUMB algorithm for $T$ rounds satisfies*

$$R(T) \leq O\left(u\left(\frac{d}{\tilde{\lambda}_x(\gamma_1 - \zeta)^2} + \frac{1}{\tilde{\lambda}_x^2}\right)\log T + \frac{\tilde{u}}{u}\frac{\epsilon_* \sqrt{d}T}{\tilde{\lambda}_x^{1.5}} + \epsilon_* T\sqrt{md\log T} + d\sqrt{mT}\log T\right) \quad (7)$$

$$\leq O(\epsilon_* T\sqrt{md\log T} + d\sqrt{mT}\log T), \quad (8)$$

*where $\gamma_1$ is defined in Definition 5.1, and $\tilde{u}$ is defined in Definition 5.2).*

**Discussion and Comparison.** The bound in Eq.(7) has four terms. The first term is the time needed to gather enough information to assign appropriate clusters for users. The second term is the regret caused by *misclustering* $\zeta$-close but not precisely similar users together, which is unavoidable with model misspecifications. The third term is from the preference estimation errors caused by model deviations. The last term is the usual term in CB with perfectly linear models [12, 25, 27].

Let us discuss how the parameters affect this regret bound.
• If $\gamma_1 - \zeta$ is large, the gaps between clusters that are not "$\zeta$-close" are much greater than the minimum gap $\zeta$ for the algorithm to distinguish, the first term in Eq.(7) will be small as it is easy to identify their dissimilarities. The role of $\gamma_1 - \zeta$ in CBMUM is similar to that of $\gamma$ in the previous CB.
• If $\tilde{u}$ is small, indicating that few *ground-truth clusters* are "$\zeta$-close", RCLUMB will hardly *miscluster* different *ground-truth clusters* together thus the second term in Eq.(7) will be small.
• If the deviation level $\epsilon_*$ is small, the user models are close to linearity and the misspecifications will not affect the estimations much, then both the second and third term in Eq.(7) will be small.
The following theorem gives a regret lower bound of the CBMUM problem.

**Theorem 5.4** (Regret lower bound for CBMUM). *There exists a problem instance for the CBMUM problem such that for any algorithm $R(T) \geq \Omega(\epsilon_* T\sqrt{d})$.*

The proof can be found in Appendix F. The upper bounds in Theorem 5.3 asymptotically match this lower bound with respect to $T$ up to logarithmic factors (and a constant factor of $\sqrt{m}$ where $m$ is typically small in real-applications), showing the tightness of our theoretical results. Additionally, we conjecture the gap for the $m$ factor is due to the strong assumption that cluster structures are known to prove this lower bound, and whether there exists a tighter lower bound is left for future work.

We then compare our results with two degenerate cases. First, when $m = 1$ (indicating $\tilde{u} = 0$), our setting degenerates to the MLB problem where all users share the same preference vector. In this case, our regret bound is $O(\epsilon_* T\sqrt{d\log T} + d\sqrt{T}\log T)$, exactly matching the current best bound of MLB [23]. Second, when $\epsilon_* = 0$, our setting reduces to the CB problem with perfectly linear user models and our bounds become $O(d\sqrt{mT}\log T)$, also perfectly match the existing best bound of the CB problem [25, 27]. The above discussions and comparisons show the tightness of our regret bounds. Additionally, we also provide detailed discussions on why trivially combining existing works on CB and MLB would not get any non-vacuous regret upper bound in Appendix D.

We define the following "good partition" for ease of interpretation.

**Definition 5.5** (Good partition). RCLUMB does a "good partition" at $t$, if the cluster $\overline{V}_t$ assigned to $i_t$ is a $\zeta$-good cluster, and it contains all the users in the same *ground-truth cluster* as $i_t$, i.e.,

$$\|\boldsymbol{\theta}_{i_t} - \boldsymbol{\theta}_\ell\|_2 \leq \zeta, \forall \ell \in \overline{V}_t, \text{ and } V_{j(i_t)} \subseteq \overline{V}_t. \quad (9)$$

Note that when the algorithm does a "good partition" at $t$, $\overline{V}_t$ will contain all the users in the same *ground-truth cluster* as $i_t$ and may only contain some other $\zeta$-close users with respect to $i_t$, which means the gathered information associated with $\overline{V}_t$ can be used to infer user $i_t$'s preference with high accuracy. Also, it is obvious that under a "good partition", if $\overline{V}_t \in \mathcal{V}$, then $\overline{V}_t = V_{j(i_t)}$ by definition.

Next, we give a sketched proof for Theorem 5.3.

*Proof.* **[Sketch for Theorem 5.3]** The proof mainly contains two parts. First, we prove there is a sufficient time $T_0$ for RCLUMB to get a "good partition" with high probability. Second, we prove the regret upper bound for RCLUMB after maintaining a "good partition". The most challenging part is to bound the regret caused by *misclustering* $\zeta$-close users after getting a "good partition".

**1. Sufficient time to maintain a "good partition".** With the item regularity (Assumption 3.3), we can prove after some $T_0$ (defined in Lemma H.1 in Appendix H), RCLUMB will always have a

"good partition". Specifically, after $t \geq O\left(u\left(\frac{d}{\tilde{\lambda}_x(\gamma_1-\zeta)^2} + \frac{1}{\tilde{\lambda}_x^2}\right)\log T\right)$, for any user $i \in \mathcal{U}$, the gap between the estimated $\hat{\boldsymbol{\theta}}_{i,t}$ and the ground-truth $\boldsymbol{\theta}^{j(i)}$ is less than $\frac{\gamma_1}{4}$ with high probability. With this, we can get: for any two users $i$ and $\ell$, if their gap is greater than $\zeta$, it will trigger the deletion of the edge $(i,\ell)$ (Line 10 of Algo.1) with high probability; on the other hand, when the deletion condition of the edge $(i,\ell)$ is satisfied, then $\left\|\boldsymbol{\theta}^{j(i)} - \boldsymbol{\theta}^{j(\ell)}\right\|_2 > 0$, which means user $i$ and $\ell$ belong to different *ground-truth clusters* by Assumption 3.1 with high probability. Therefore, we can get that with high probability, all those users in the same *ground-truth cluster* as $i_t$ will be directly connected with $i_t$, and users directly connected with $i_t$ must be $\zeta$-close to $i_t$. By filtering users directly linked with $i_t$ as the cluster $\overline{V}_t$ (Algo.1 Line 5) and the definition of "good partition", we can ensure that RCLUMB will keep a "good partition" afterward with high probability.

**2. Bounding the regret after getting a "good partition".** After $T_0$, with the "good partition", we can prove the following lemma that gives a bound of the difference between $\hat{\boldsymbol{\theta}}_{\overline{V}_t,t-1}$ and ground-truth $\boldsymbol{\theta}_{i_t}$ in direction of action vector $\boldsymbol{x}_a$, and supports the design of the confidence radius $C_{a,t}$ in Eq.(5).

**Lemma 5.6.** *With probability at least* $1 - 5\delta$ *for some* $\delta \in (0, \frac{1}{5})$, $\forall t \geq T_0$

$$\left|\boldsymbol{x}_a^\top(\boldsymbol{\theta}_{i_t} - \hat{\boldsymbol{\theta}}_{\overline{V}_t,t-1})\right| \leq \frac{\epsilon_*\sqrt{2d}}{\tilde{\lambda}_x^{\frac{3}{2}}}\mathbb{I}\{\overline{V}_t \notin \mathcal{V}\} + \epsilon_* \sum_{\substack{s \in [t-1] \\ i_s \in \overline{V}_t}} \left|\boldsymbol{x}_a^\top \overline{\boldsymbol{M}}_{\overline{V}_t,t-1}^{-1} \boldsymbol{x}_{a_s}\right| + \beta \left\|\boldsymbol{x}_a\right\|_{\overline{\boldsymbol{M}}_{\overline{V}_t,t-1}^{-1}} .$$

To prove this lemma, we consider the following two situations.

**(i) Assigning a perfect cluster for** $i_t$. In this case, $\overline{V}_t \in \mathcal{V}$, meaning the cluster assigned for user $i_t$ is the same as her *ground-truth cluster*, i.e., $\overline{V}_t = V_{j(i_t)}$. Therefore, we have that $\forall \ell \in \overline{V}_t, \boldsymbol{\theta}_\ell = \boldsymbol{\theta}_{i_t}$. With careful analysis, we can bound $\left|\boldsymbol{x}_a^\top(\boldsymbol{\theta}_{i_t} - \hat{\boldsymbol{\theta}}_{\overline{V}_t,t-1})\right|$ by $C_{a,t}$ (defined in Eq.(5)).

**(ii) Bounding the term of** *misclustering* $i_t$**'s** $\zeta$**-close users.** In this case, $\overline{V}_t \notin \mathcal{V}$, meaning the algorithm *misclusters* user $i_t$, i.e., $\overline{V}_t \neq V_{j(i_t)}$. Thus, we do not have $\forall \ell \in \overline{V}_t, \boldsymbol{\theta}_\ell = \boldsymbol{\theta}_{i_t}$ anymore, but we have all the users in $\overline{V}_t$ are $\zeta$-close to $i_t$ (by "good partition"), i.e., $\left\|\boldsymbol{\theta}_{i_s} - \boldsymbol{\theta}_{i_t}\right\|_2 \leq \zeta, \forall \ell \in \overline{V}_t$. Then an additional term can be caused by using the information of $i_t$'s $\zeta$-close users in $\overline{V}_t$ lying in different *ground-truth clusters* from $i_t$ to estimate $\boldsymbol{\theta}_{i_t}$. It is highly challenging to bound this part.

We will get an extra term $\left|\boldsymbol{x}_a^\top \overline{\boldsymbol{M}}_{\overline{V}_t,t-1}^{-1} \sum_{\substack{s \in [t-1] \\ i_s \in \overline{V}_t}} \boldsymbol{x}_{a_s} \boldsymbol{x}_{a_s}^\top(\boldsymbol{\theta}_{i_s} - \boldsymbol{\theta}_{i_t})\right|$ when bounding the regret in this case, where $\left\|\boldsymbol{\theta}_\ell - \boldsymbol{\theta}_{i_t}\right\|_2 \leq \zeta, \forall \ell \in \overline{V}_t$. It is an easy-to-be-made mistake to directly drag $\left\|\boldsymbol{\theta}_{i_s} - \boldsymbol{\theta}_{i_t}\right\|_2$ out to bound it by $\left\|\boldsymbol{x}_a^\top \overline{\boldsymbol{M}}_{\overline{V}_t,t-1}^{-1} \sum_{\substack{s \in [t-1] \\ i_s \in \overline{V}_t}} \boldsymbol{x}_{a_s} \boldsymbol{x}_{a_s}^\top\right\|_2 \times \zeta$. With subtle analysis, we propose the following lemma to bound the above term.

**Lemma 5.7** (Bound of error caused by *misclustering*). $\forall t \geq T_0$, *if the current partition by RCLUMB is a "good partition", and* $\overline{V}_t \notin \mathcal{V}$, *then for all* $\boldsymbol{x}_a \in \mathbb{R}^d, \left\|\boldsymbol{x}_a\right\|_2 \leq 1$, *with probability at least* $1 - \delta$:

$$\left|\boldsymbol{x}_a^\top \overline{\boldsymbol{M}}_{\overline{V}_t,t-1}^{-1} \sum_{\substack{s \in [t-1] \\ i_s \in \overline{V}_t}} \boldsymbol{x}_{a_s} \boldsymbol{x}_{a_s}^\top(\boldsymbol{\theta}_{i_s} - \boldsymbol{\theta}_{i_t})\right| \leq \frac{\epsilon_*\sqrt{2d}}{\tilde{\lambda}_x^{\frac{3}{2}}} .$$

This lemma is quite general. Please see Appendix G for details about its proof.

The expected occurrences of $\{\overline{V}_t \notin \mathcal{V}\}$ is bounded by $\frac{\tilde{u}}{u}T$ with Assumption 3.2, Definition 5.2 and 5.5. The result follows by bounding the expected sum of the bounds for the instantaneous regret using Lemma 5.6 with delicate analysis due to the time-varying clustering structure kept by RCLUMB. $\square$

# 6 Experiments

This section compares RCLUMB and RSCLUMB with CLUB [12], SCLUB [27], LinUCB with a single estimated vector for all users, LinUCB-Ind with separate estimated vectors for each user, and two modifications of LinUCB in [23] which we name as RLinUCB and RLinUCB-Ind. We use averaged reward as the evaluation metric, where the average is taken over ten independent trials.

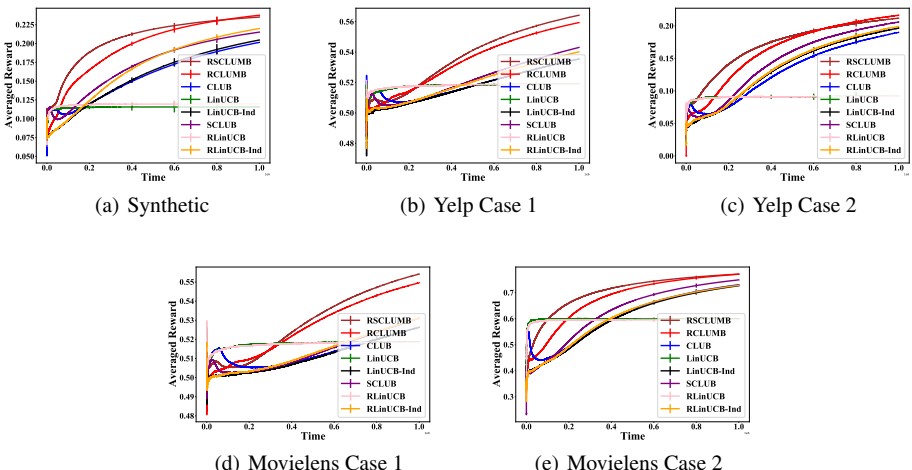

(a) Synthetic  (b) Yelp Case 1  (c) Yelp Case 2

(d) Movielens Case 1  (e) Movielens Case 2

Figure 1: The figures compare RCLUMB and RSCLUMB with the baselines. (a) shows the result on synthetic data, (b) and (c) show the results on Yelp dataset, (d) and (e) show the results on Movielens dataset. All experiments are under the setting of $u = 1,000$ users, $m = 10$ clusters, and $d = 50$. All results are averaged under 10 random trials. The error bars are standard deviations divided by $\sqrt{10}$.

## 6.1 Synthetic Experiments

We consider a setting with $u = 1,000$ users, $m = 10$ clusters and $T = 10^6$ rounds. The preference and feature vectors are in $d = 50$ dimension with each entry drawn from a standard Gaussian distribution, and are normalized to vectors with $\|.\|_2 = 1$ [27]. We fix an arm set with $|\mathcal{A}| = 1000$ items, at each round $t$, 20 items are randomly selected to form a set $\mathcal{A}_t$ for the user to choose from. We construct a matrix $\epsilon \in \mathbb{R}^{1,000 \times 1,000}$ in which each element $\epsilon(i, j)$ is drawn uniformly from the range $(-0.2, 0.2)$ to represent the deviation. At $t$, for user $i_t$ and the item $a_t$, $\epsilon(i_t, a_t)$ will be added to the feedback as the deviation, which corresponds to the $\epsilon_{a_t}^{i_t, t}$ defined in Eq.(1).

The result is provided in Figure 1(a), showing that our algorithms have clear advantages: RCLUMB improves over CLUB by 21.9%, LinUCB by 194.8%, LinUCB-Ind by 20.1%, SCLUB by 12.0%, RLinUCB by 185.2% and RLinUCB-Ind by 10.6%. The performance difference between RCLUMB and RSCLUMB is very small as expected. RLinUCB performs better than LinUCB; RLinUCB-Ind performs better than LinUCB-Ind and CLUB, showing that the modification of the recommendation policy is effective. The set-based RSCLUMB and SCLUB can separate clusters quicker and have advantages in the early period, but eventually RCLUMB catches up with RSCLUMB, and SCLUB is surpassed by RLinUCB-Ind because it does not consider misspecifications. RCLUMB and RSCLUMB perform better than RLinUCB-Ind, which shows the advantage of the clustering. So it can be concluded that both the modification for misspecification and the clustering structure are critical to improving the algorithm's performance. We also have done some ablation experiments on different scales of $\epsilon^*$ in Appendix P , and we can notice that under different $\epsilon^*$ , our algorithms always outperform the baselines, and some baselines will perform worse as $\epsilon^*$ increases.

## 6.2 Experiments on Real-world Datasets

We conduct experiments on the Yelp data and the $20m$ MovieLens data [17]. For both data, we have two cases due to the different methods for generating feedback. For case 1, we extract 1,000 items with most ratings and 1,000 users who rate most; then we construct a binary matrix $\boldsymbol{H}^{1,000 \times 1,000}$ based on the user rating [40, 42]: if the user rating is greater than 3, the feedback is 1; otherwise, the feedback is 0. Then we use this binary matrix to generate the preference and feature vectors by singular-value decomposition (SVD) [27, 25, 40]. Similar to the synthetic experiment, we construct a matrix $\boldsymbol{\epsilon} \in \mathbb{R}^{1,000 \times 1,000}$ in which each element is drawn uniformly from the range $(-0.2, 0.2)$. For case 2, we extract 1,100 users who rate most and 1000 items with most ratings. We construct a binary feedback matrix $\boldsymbol{H}^{1,100 \times 1,000}$ based on the same rule as case 1. Then we select the first 100 rows $\boldsymbol{H}_1^{100 \times 1,000}$ to generate the feature vectors by SVD. The remaining 1,000 rows $\boldsymbol{F}^{1,000 \times 1,000}$

is used as the feedback matrix, meaning user $i$ receives $\boldsymbol{F}(i, j)$ as feedback while choosing item $j$. In both cases, at time $t$, we randomly select 20 items for the algorithms to choose from. In case 1, the feedback is computed by the preference and feature vector with misspecification, in case 2, the feedback is from the feedback matrix.

The results on Yelp are shown in Fig 1(b) and Fig 1(c). In case 1, RCLUMB improves CLUB by 45.1%, SCLUB by 53.4%, LinUCB-One by 170.1% , LinUCB-Ind by 46.2%, RLinUCB by 171.0% and RLinUCB-Ind by 21.5%. In case 2, RCLUMB improves over CLUB by 13.9%, SCLUB by 5.1%, LinUCB-One by 135.6% , LinUCB-Ind by 10.1%, RLinUCB by 138.6% and RLinUCB by 8.5%. It is notable that our modeling assumption 3.4 is violated in case 2 since the misspecification range is unknown. We set $\epsilon_* = 0.2$ following our synthetic dataset and it can still perform better than other algorithms. When the misspecification level is known as in case 1, our algorithms' improvement is significantly enlarged, e.g., RCLUMB improves over SCLUB from 5.1% to 53.4%.

The results on Movielens are shown in Fig 1(d) and 1(e). In case 1, RCLUMB improves CLUB by 58.8%, SCLUB by 92.1%, LinUCB-One by 107.7%, LinUCB-Ind by 61.5 %, RLinUCB by 109.5%, and RLinUCB-Ind by 21.3%. In case 2, RCLUMB improves over CLUB by 5.5%, SCLUB by 2.9%, LinUCB-One by 28.5%, LinUCB-Ind by 6.1%, RLinUCB by 29.3% and RLinUCB-Ind by 5.8%. The results are consistent with the Yelp data, confirming our superior performance.

## 7   Conclusion

We present a new problem of clustering of bandits with misspecified user models (CBMUM), where the agent has to adaptively assign appropriate clusters for users under model misspecifications. We propose two robust CB algorithms, RCLUMB and RSCLUMB. Under milder assumptions than previous CB works, we prove the regret bounds of our algorithms, which match the lower bound asymptotically in $T$ up to logarithmic factors, and match the state-of-the-art results in several degenerate cases. It is challenging to bound the regret caused by *misclustering* users with close but not the same preference vectors and use inaccurate cluster-based information to select arms. Our analysis to bound this part of the regret is quite general and may be of independent interest. Experiments on synthetic and real-world data demonstrate the advantage of our algorithms. We would like to state some interesting future works: (1) Prove a tighter regret lower bound for CBMUM, (2) Incorporate recent model selection methods into our fundamental framework to design robust algorithms for CBMUM with unknown exact maximum model misspecification level, and (3) Consider the setting with misspecifications in the underlying user clustering structure rather than user models.

## 8   Acknowledgement

The corresponding author Shuai Li is supported by National Key Research and Development Program of China (2022ZD0114804) and National Natural Science Foundation of China (62376154, 62006151, 62076161). The work of John C.S. Lui was supported in part by the RGC's GRF 14215722.

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

# Appendix

## A More Discussions on Related Work

In this section, we will give more comparisions and discussions on some previous works that are related to our work to some extent.

There are some other works on bandits leveraging user (or task) relations, which have some relations with the clustering of bandits (CB) works to some extent, but are in different lines of research from CB, and are quite different from our work. First, besides CB, the work [41] also leverages user relations. Specifically, it utilizes a *known* user adjacency graph to share context and payoffs among neighbors, whereas in CB, the user relations are *unknown* and need to be learnt, thus the setting differs a lot from CB. Second, there are lines of works on multi-task learning [6, 10, 33, 8, 37, 36], meta-learning [35, 18, 7] and federated learning [32, 19], where multiple different tasks are solved jointly and share information. Note that all of these works do not assume an underlying *unknown* user clustering structure which needs to be inferred by the agent to speed up learning. For works on multi-task learning [6, 10, 33, 8, 37, 36], they assume the tasks are related but no user clustering structures, and to the best of our knowledge, none of them consider model misspefications, thus differing a lot from ours. For some recent works on meta-learning [35, 18, 34], they propose general Bayesian hierarchical models to share knowledge across tasks, and design Thompson-Sampling-based algorithms to optimize the Bayes regret, which are quite different from the line of CB works, and differ a lot from ours. And additionally, as supported by the discussions in the works [7, 36], multi-task learning and meta-learning are different lines of research from CB. For the works on federated learning [32, 19], they consider the privacy and communication costs among multiple servers, whose setting is also very different from the previous CB works and our work.

**Remark.** Again, we emphasize that the goal of this work is to initialize the study of the important CBMUM problem, and propose general design ideas for dealing with model misspecifications in CB problems. Therefore, our study is based on fundamental models on CB [12, 27] and MLB [23], and the algorithm design ideas and theoretical analysis are pretty general. We leave incorporating the more recent model selection methods [31, 11] into our framework to address the unknown exact maximum model misspecification level as an interesting future work. It would also be interesting to consider incorporating our methods and ideas of tackling model misspecifications into the studies of multi-task learning, meta learning and federated learning.

## B More Discussions on Assumptions

All the assumptions (Assumptions 3.1,3.2,3.3,3.4)in this work are natural and basically follow (or less strigent than) previous works on CB and MLB [12, 25, 27, 28, 23].

### B.1 Less Strigent Assumption on on the Generating Distribution of Arm Vectors

We also make some contributions to relax a widely-used but stringent assumption on the generating distribution of arm vectors. Specifically, our Assumption 3.3 on item regularity relaxes the previous one used in previous CB works [12, 25, 27, 28] by removing the condition that the variance should be upper bounded by $\frac{\lambda^2}{8\log(4|\mathcal{A}_t|)}$. For technical details on this, please refer to the theoretical analysis and discussions in Appendix J.

### B.2 Discussions on Assumption 3.4 about Bounded Misspecification Level

This assumption follows [23]. Note that this $\epsilon_*$ can be an upper bound on the maximum misspecification level, not the exact maximum itself. In real-world applications, the deviations are usually small [14], and we can set a relatively big $\epsilon_*$ (e.g., 0.2) to be the upper bound. Our experimental results support this claim. As shown in our experimental results on real-data case 2, even when $\epsilon_*$ is unknown, our algorithms still perform well by setting $\epsilon_* = 0.2$. Some recent studies [31, 11] use model selection methods to theoretically deal with unknown exact maximum misspecification level in the single-user case, which is not the emphasis of this work. Additionally, the work [11] assumes that the learning agent has access to a regression oracle. And for the work [31], though their regret bound

is dependent on the exact maximum misspecification level that needs not to be known by the agent, an upper bound of the exact maximum misspecification level is still needed. We leave incorporating their methods to deal with unknown exact maximum misspecification level as an interesting future work.

### B.3 Discussions on Assumption 3.2 about the Theoretical Results under General User Arrival Distributions

The uniform arrival in Assumption 3.2 follows previous CB works [12, 25, 28], it only affects the $T_0$ term, which is the time after which the algorithm maintains a "good partition" and is of $O(u \log T)$. For an arbitrary arrival distribution, $T_0$ becomes $O(1/p_{min} \log T)$, where $p_{min}$ is the minimal arrival probability of a user. And since it is a lower-order term (of $O(\log T)$), it will not affect the main order of our regret upper bound which is of $O(\epsilon_* T \sqrt{md \log T} + d\sqrt{mT} \log T)$. The work [27] studies arbitrary arrivals and aims to remove the $1/p_{min}$ factor in this term, but their setting is different. They make an additional assumption that users in the same cluster not only have the same preference vector, but also the same arrival probability, which is different from our setting and other classic CB works [12, 25, 28] where we only assume users in the same cluster share the same preference vector.

## C Highlight of the Theoretical Analysis

Our proof flow and methodologies are novel in clustering of bandits (CB), which are expected to inspire future works on model misspecifications and CB. The main challenge of the regret analysis in CBMUM is that due to the estimation inaccuracy caused by misspecifications, it is impossible to cluster all users exactly correctly, and it is highly non-trivial to bound the regret caused by **"misclustering" $\zeta$-close users**.

To the best of our knowledge, the common proof flow of previous CB works (e.g., [12, 25, 28]) can be summarized in two steps: The first is to prove a sufficient time $T_0'$ after which the algorithms can cluster all users **exactly correctly** with high probability. Note that the inferred clustering structure remains static after $T_0'$, making the analysis easy. Second, after the **correct static clustering**, the regret can be trivially bounded by bounding $m$ (number of underlying clusters) independent linear bandit algorithms, resulting in a $O(d\sqrt{mT} \log T)$ regret.

The above common proof flow is straightforward in CB with perfectly linear models, but it would fail to get a non-vacuous regret bound for CBMUM. In CBMUM, it is impossible to learn an exactly correct static clustering structure with model misspecifications. In particular, we prove that we can only expect the algorithm to cluster $\zeta$-close users together rather than cluster all users exactly correctly. Therefore, the previous flow can not be applied to the more challenging CBMUM problem.

We do the following to address the challenges in obtaining a tight regret bound for CBMUM. With the carefully-designed novel key components of RCLUMB, we can prove a sufficient time $T_0$ after which RCLUMB can get a "good partition" (Definition 5.5) with high probability, which means the cluster $\overline{V}_t$ assigned to $i_t$ contains all users in the same ground-truth cluster as $i_t$, and possibly some other $i_t$'s $\zeta$-close users. Intuitively, after $T_0$, the algorithm can leverage all the information from the users' ground-truth clusters but may misuse some information from other $\zeta$-close users with preference gaps up to $\zeta$, causing a regret of **"misclustering" $\zeta$-close users**. It is highly non-trivial to bound this part of regret, and the proof methods would be beneficial for future studies in CB in challenging cases when it is impossible to cluster all users exactly correctly. For details, please refer to the discussions "(ii) Bounding the term of misclustering it's $\zeta$-close users" in Section 5, the key Lemma 5.7 (Bound of error caused by misclustering), its proof and tightness discussion in Appendix G. Also, a more subtle analysis is needed to handle the time-varying inferred clustering structure since the "good partition" may change over time, whereas in the previous CB works, the clustering structure remains static after $T_0'$. For theoretical details on this, please refer to Appendix E.

## D Discussions on why Trivially Combining Existing CB and MLB Works Could Not Achieve a Non-vacuous Regret Upper Bound

We consider discussing regret upper bounds for CB without considering misspecifications for three cases: (1) neither the clustering process nor the decision process considers misspecifications (previous

CB algorithms); (2) the decision process does not consider misspecifications; (3) the clustering process does not consider misspecifications.

For cases (1) and (2), the decision process could contribute to the leading regret. We consider the case where there are $m$ underlying clusters, with each cluster's arrival being $T/m$, and the agent knows the underlying clustering structure. For this case, there exist some instances where the regret upper bound $R(T)$ is strictly larger than $\epsilon_* T\sqrt{m\log T}$ asymptotically in $T$. Formally, in the discussion of "Failure of unmodified algorithm" in Appendix E in [23], they give an example to show that in the single-user case, the regret $R_1(T)$ of the classic linear bandit algorithms without considering misspecifications will have: $\lim_{T\to+\infty}\frac{R_1(T)}{\epsilon_* T\sqrt{m\log T}} = +\infty$. In our problem with multiple users and $m$ underlying clusters, even if we know the underlying clustering structure and keep $m$ independent linear bandit algorithms with $T_i$ for the cluster $i\in[m]$ to leverage the common information of clusters, the best we can get is $R_2(T) = \sum_{i\in[m]} R_1(T_i)$. By the above results, if the decision process does not consider misspecifications, we have $\lim_{T\to+\infty}\frac{R_2(T)}{\epsilon_* T\sqrt{m\log T}} = \lim_{T\to+\infty}\frac{mR_1(T/m)}{\epsilon_* T\sqrt{m\log T}} = +\infty$. Recall that the regret upper bound $R(T)$ of our proposed algorithms is of $O(\epsilon_* T\sqrt{md\log T} + d\sqrt{mT}\log T)$ (thus, we have $\lim_{T\to+\infty}\frac{R(T)}{\epsilon_* T\sqrt{m\log T}} < +\infty$), which gives a proof that that the regret upper bound of our proposed algorithms is asymptotically much better than CB algorithms in cases (1)(2).

For case (3), if the clustering process does not use the more tolerant deletion rule in Line 10 of Algo.1, the gap between users linked by edges would possibly exceed $\zeta$ ($\zeta = 2\epsilon_*\sqrt{\frac{2}{\tilde{\lambda}_x}}$) even after $T_0$, which will result in a regret upper bound no better than $O(\epsilon_* u\sqrt{dT})$. As the number of users $u$ is usually huge in practice, this result is vacuous. The reasons for getting the above claim are as follows. Even if the clustering process further uses our deletion rule considering misspecifications, and the users linked by edges are within $\zeta$ distance, failing to extract 1-hop users (Line 5 in Algo.1) would cause the leading $O(\epsilon_* u\sqrt{dT})$ regret term, as in the worst case, the preference vector $\theta$ of the user in $\tilde{V}_t$ who is $h$-hop away from user $i_t$ could deviate by $h\zeta$ from $\theta_{i_t}$, where $h$ can be as large as $u$, and it would make the second term in Eq.(8) a $O(\epsilon_* u\sqrt{dT})$ term. If we completely do not consider the misspecifications in the clustering process, the above user gap between users linked by edges would possibly exceed $\zeta$, which will cause a regret upper bound worse than $O(\epsilon_* u\sqrt{dT})$.

# E  Proof of Theorem 5.3

We first prove the result in the case when $\gamma_1$ defined in Definition 5.1 is not infinity, i.e., $4\epsilon_*\sqrt{\frac{2}{\tilde{\lambda}_x}} < \gamma_1 < \infty$. The proof of the special case when $\gamma_1 = \infty$ will directly follow the proof of this case.

For the instantaneous regret $R_t$ at round $t$, with probability at least $1 - 5\delta$ for some $\delta \in (0, \frac{1}{5})$, at $\forall t \geq T_0$:

$$
\begin{aligned}
R_t &= (\boldsymbol{x}_{a_t^*}^\top \boldsymbol{\theta}_{i_t} + \boldsymbol{\epsilon}_{a_t^*}^{i_t,t}) - (\boldsymbol{x}_{a_t}^\top \boldsymbol{\theta}_{i_t} + \boldsymbol{\epsilon}_{a_t}^{i_t,t}) \\
&= \boldsymbol{x}_{a_t^*}^\top(\boldsymbol{\theta}_{i_t} - \hat{\boldsymbol{\theta}}_{\overline{V}_t,t-1}) + (\boldsymbol{x}_{a_t^*}^\top \hat{\boldsymbol{\theta}}_{\overline{V}_t,t-1} + C_{a_t^*,t}) - (\boldsymbol{x}_{a_t}^\top \hat{\boldsymbol{\theta}}_{\overline{V}_t,t-1} + C_{a_t,t}) \\
&\quad + \boldsymbol{x}_{a_t}^\top(\hat{\boldsymbol{\theta}}_{\overline{V}_t,t-1} - \boldsymbol{\theta}_{i_t}) + C_{a_t,t} - C_{a_t^*,t} + (\boldsymbol{\epsilon}_{a_t^*}^{i_t,t} - \boldsymbol{\epsilon}_{a_t}^{i_t,t}) \\
&\leq 2C_{a_t,t} + \frac{2\epsilon_*\sqrt{2d}}{\tilde{\lambda}_x^{\frac{3}{2}}}\mathbb{I}\{\overline{V}_t \notin \mathcal{V}\} + 2\epsilon_* \,,
\end{aligned}
\tag{10}
$$

where the last inequality holds by the UCB arm selection strategy in Eq.(5), the concentration bound given in Lemma 5.6, and the fact that $\left\|\boldsymbol{\epsilon}^{i,t}\right\|_\infty \leq \epsilon_*, \forall i \in \mathcal{U}, \forall t$.

We define the following events. Let

$$\mathcal{E}_0 = \left\{R_t \leq 2C_{a_t,t} + \frac{2\epsilon_*\sqrt{2d}}{\tilde{\lambda}_x^{\frac{3}{2}}}\mathbb{I}\{\overline{V}_t \notin \mathcal{V}\} + 2\epsilon_*, \text{ for all } \{t : t \geq T_0, \text{ and the algorithm maintains a "good partition" at } t\}\right\},$$

$\mathcal{E}_1 = \{\text{the algorithm maintains a "good partition" for all } t \geq T_0\}$,
$\mathcal{E} = \mathcal{E}_0 \cap \mathcal{E}_1$.

$\mathbb{P}(\mathcal{E}_0) \geq 1 - 2\delta$. According to Lemma H.1, $\mathbb{P}(\mathcal{E}_1) \geq 1 - 3\delta$. Thus, $\mathbb{P}(\mathcal{E}) \geq 1 - 5\delta$ for some $\delta \in (0, \frac{1}{5})$. Take $\delta = \frac{1}{T}$, we can get that

$$
\begin{aligned}
\mathbb{E}[R(T)] &= \mathbb{P}(\mathcal{E})\mathbb{I}\{\mathcal{E}\}R(T) + \mathbb{P}(\overline{\mathcal{E}})\mathbb{I}\{\overline{\mathcal{E}}\}R(T) \\
&\leq \mathbb{I}\{\mathcal{E}\}R(T) + 5 \times \frac{1}{T} \times T \\
&= \mathbb{I}\{\mathcal{E}\}R(T) + 5\,,
\end{aligned}
\tag{11}
$$

where $\overline{\mathcal{E}}$ denotes the complementary event of $\mathcal{E}$, $\mathbb{I}\{\mathcal{E}\}R(T)$ denotes $R(T)$ under event $\mathcal{E}$, $\mathbb{I}\{\overline{\mathcal{E}}\}R(T)$ denotes $R(T)$ under event $\overline{\mathcal{E}}$, and we use $R(T) \leq T$ to bound $R(T)$ under event $\overline{\mathcal{E}}$.

Then it remains to bound $\mathbb{I}\{\mathcal{E}\}R(T)$:

$$
\begin{aligned}
\mathbb{I}\{\mathcal{E}\}R(T) &\leq R(T_0) + \mathbb{E}[\mathbb{I}\{\mathcal{E}\} \sum_{t=T_0+1}^{T} R_t] \\
&\leq T_0 + 2\mathbb{E}[\mathbb{I}\{\mathcal{E}\} \sum_{t=T_0+1}^{T} C_{a_t,t}] + \frac{2\epsilon_* \sqrt{2d}}{\tilde{\lambda}_x^{\frac{3}{2}}} \sum_{t=T_0+1}^{T} \mathbb{E}[\mathbb{I}\{\mathcal{E}, \overline{V}_t \notin \mathcal{V}\}] + 2\epsilon_* T \quad (12) \\
&= T_0 + 2\mathbb{E}[\mathbb{I}\{\mathcal{E}\} \sum_{t=T_0+1}^{T} C_{a_t,t}] + \frac{2\epsilon_* \sqrt{2d}}{\tilde{\lambda}_x^{\frac{3}{2}}} \sum_{t=T_0+1}^{T} \mathbb{P}(\mathbb{I}\{\mathcal{E}, \overline{V}_t \notin \mathcal{V}\}) + 2\epsilon_* T \\
&\leq T_0 + 2\mathbb{E}[\mathbb{I}\{\mathcal{E}\} \sum_{t=T_0+1}^{T} C_{a_t,t}] + \frac{2\epsilon_* \sqrt{2d}}{\tilde{\lambda}_x^{\frac{3}{2}}} \times \frac{\tilde{u}}{u} T + 2\epsilon_* T\,, \quad (13)
\end{aligned}
$$

where Eq.(12) follows from Eq.(10). Eq.(13) holds since under Assumption 3.2 about user arrival uniformness and by Definition 5.5 of "good partition", $\mathbb{P}(\mathbb{I}\{\mathcal{E}, \overline{V}_t \notin \mathcal{V}\}) \leq \frac{\tilde{u}}{u}, \forall t \geq T_0$, where $\tilde{u}$ is defined in Definition 5.2.

Then we need to bound $\mathbb{E}[\mathbb{I}\{\mathcal{E}\} \sum_{t=T_0+1}^{T} C_{a_t,t}]$:

$$
\begin{aligned}
\mathbb{I}\{\mathcal{E}\} \sum_{t=T_0+1}^{T} C_{a_t,t} &= \Big(\sqrt{\lambda} + \sqrt{2\log(\frac{1}{\delta}) + d\log(1 + \frac{T}{\lambda d})}\Big)\mathbb{I}\{\mathcal{E}\} \sum_{t=T_0+1}^{T} \|\boldsymbol{x}_{a_t}\|_{\overline{\boldsymbol{M}}_{\overline{V}_t,t-1}^{-1}} \\
&\quad + \mathbb{I}\{\mathcal{E}\}\epsilon_* \sum_{t=T_0+1}^{T} \sum_{\substack{s \in [t-1] \\ i_s \in \overline{V}_t}} \Big|\boldsymbol{x}_{a_t}^\top \overline{\boldsymbol{M}}_{\overline{V}_t,t-1}^{-1} \boldsymbol{x}_{a_s}\Big|\,. \quad (14)
\end{aligned}
$$

Next, we bound the $\mathbb{I}\{\mathcal{E}\} \sum_{t=T_0+1}^{T} \|\boldsymbol{x}_{a_t}\|_{\overline{\boldsymbol{M}}_{\overline{V}_t,t-1}^{-1}}$ term in Eq.(14):

$$
\begin{aligned}
\mathbb{I}\{\mathcal{E}\} \sum_{t=T_0+1}^{T} \|\boldsymbol{x}_{a_t}\|_{\overline{\boldsymbol{M}}_{\overline{V}_t,t-1}^{-1}} &= \mathbb{I}\{\mathcal{E}\} \sum_{t=T_0+1}^{T} \sum_{k=1}^{m_t} \mathbb{I}\{i_t \in \tilde{V}'_{t,k}\} \|\boldsymbol{x}_{a_t}\|_{\overline{\boldsymbol{M}}_{\overline{V}'_{t,k},t-1}^{-1}} \\
&\leq \mathbb{I}\{\mathcal{E}\} \sum_{t=T_0+1}^{T} \sum_{j=1}^{m} \mathbb{I}\{i_t \in V_j\} \|\boldsymbol{x}_{a_t}\|_{\overline{\boldsymbol{M}}_{V_j,t-1}^{-1}} \quad (15) \\
&\leq \mathbb{I}\{\mathcal{E}\} \sum_{j=1}^{m} \sqrt{\sum_{t=T_0+1}^{T} \mathbb{I}\{i_t \in V_j\} \sum_{t=T_0+1}^{T} \mathbb{I}\{i_t \in V_j\} \|\boldsymbol{x}_{a_t}\|_{\overline{\boldsymbol{M}}_{V_j,t-1}^{-1}}^2} \\
&\qquad\qquad\qquad\qquad\qquad\qquad\qquad\qquad\qquad\qquad (16) \\
&\leq \mathbb{I}\{\mathcal{E}\} \sum_{j=1}^{m} \sqrt{2T_{V_j,T} d\log(1 + \frac{T}{\lambda d})} \quad (17) \\
&\leq \mathbb{I}\{\mathcal{E}\} \sqrt{2 \sum_{j=1}^{m} 1 \sum_{j=1}^{m} T_{V_j,T} d\log(1 + \frac{T}{\lambda d})} = \mathbb{I}\{\mathcal{E}\}\sqrt{2mdT\log(1 + \frac{T}{\lambda d})}\,, \\
&\qquad\qquad\qquad\qquad\qquad\qquad\qquad\qquad\qquad\qquad (18)
\end{aligned}
$$

where we use $m_t$ to denote the number of connected components partitioned by the algorithm at $t$, $\tilde{V}'_{t,k}, k \in [m_t]$ to denote the connected components partitioned by the algorithm at $t$, $\overline{V}'_{t,k} \subseteq \tilde{V}'_{t,k}$ to denote the subset extracted to be the cluster $\overline{V}_t$ for $i_t$ from $\tilde{V}'_{t,k}$ conditioned on $i_t \in \tilde{V}'_{t,k}$, and $T_{V_j,T}$ to denote the number of times that the served users lie in the *ground-truth cluster* $V_j$ up to time $T$, i.e., $T_{V_j,T} = \sum_{t\in[T]} \mathbb{I}\{i_t \in V_j\}$.

The reasons for having Eq.(15) are as follows. Under event $\mathcal{E}$, the algorithm will always have a "good partition" after $T_0$. By Definition 5.5 and the proof process of Lemma H.1 about the edge deletion conditions, we can get $m_t \leq m$ and if $i_t \in \tilde{V}'_{t,k}, i_t \in V_j$, then $V_j \subseteq \overline{V}'_{t,k}$ since $\overline{V}'_{t,k}$ contains $V_j$ and possibly other *ground-truth clusters* $V_n, n \in [m]$, whose preference vectors are $\zeta$-close to $\boldsymbol{\theta}^j$. Therefore, by the definition of the regularized Gramian matrix, we can get $M_{\overline{V}'_{t,k},t-1} \succeq M_{V_j,t-1}, \forall t \geq T_0 + 1$. Thus by the above reasoning, $\sum_{k=1}^{m_t} \mathbb{I}\{i_t \in \tilde{V}'_{t,k}\} \|\boldsymbol{x}_{a_t}\|_{\overline{M}^{-1}_{\tilde{V}'_{t,k},t-1}} \leq \sum_{j=1}^{m} \mathbb{I}\{i_t \in V_j\} \|\boldsymbol{x}_{a_t}\|_{\overline{M}^{-1}_{V_j,t-1}}, \forall t \geq T_0 + 1$. Eq.(16) holds by the Cauchy–Schwarz inequality; Eq.(17) follows by the following technical Lemma J.2. Eq.(18) is from the Cauchy–Schwarz inequality and the fact that $\sum_{j=1}^{m} T_{V_j,T} = T$.

We then bound the last term in Eq.(14):

$$\mathbb{I}\{\mathcal{E}\}\epsilon_* \sum_{t=T_0+1}^{T} \sum_{\substack{s\in[t-1]\\i_s\in\overline{V}_t}} \left|\boldsymbol{x}_{a_t}^\top \overline{M}^{-1}_{\overline{V}_t,t-1}\boldsymbol{x}_{a_s}\right| = \mathbb{I}\{\mathcal{E}\}\epsilon_* \sum_{t=T_0+1}^{T} \sum_{k=1}^{m_t} \mathbb{I}\{i_t \in \tilde{V}'_{t,k}\} \sum_{\substack{s\in[t-1]\\i_s\in\overline{V}'_{t,k}}} \left|\boldsymbol{x}_{a_t}^\top \overline{M}^{-1}_{\overline{V}'_{t,k},t-1}\boldsymbol{x}_{a_s}\right|$$

$$\leq \mathbb{I}\{\mathcal{E}\}\epsilon_* \sum_{t=T_0+1}^{T} \sum_{k=1}^{m_t} \mathbb{I}\{i_t \in \tilde{V}'_{t,k}\} \sqrt{\sum_{\substack{s\in[t-1]\\i_s\in\overline{V}'_{t,k}}} 1 \sum_{\substack{s\in[t-1]\\i_s\in\overline{V}'_{t,k}}} \left|\boldsymbol{x}_{a_t}^\top \overline{M}^{-1}_{\overline{V}'_{t,k},t-1}\boldsymbol{x}_{a_s}\right|^2}$$

$$(19)$$

$$\leq \mathbb{I}\{\mathcal{E}\}\epsilon_* \sum_{t=T_0+1}^{T} \sum_{k=1}^{m_t} \mathbb{I}\{i_t \in \tilde{V}'_{t,k}\} \sqrt{T_{\overline{V}'_{t,k},t-1} \|\boldsymbol{x}_{a_t}\|^2_{\overline{M}^{-1}_{\overline{V}'_{t,k},t-1}}}$$

$$(20)$$

$$\leq \mathbb{I}\{\mathcal{E}\}\epsilon_* \sum_{t=T_0+1}^{T} \sqrt{\sum_{k=1}^{m_t} \mathbb{I}\{i_t \in \tilde{V}'_{t,k}\} \sum_{k=1}^{m_t} \mathbb{I}\{i_t \in \tilde{V}'_{t,k}\} T_{\overline{V}'_{t,k},t-1} \|\boldsymbol{x}_{a_t}\|^2_{\overline{M}^{-1}_{\overline{V}'_{t,k},t-1}}}$$

$$(21)$$

$$\leq \mathbb{I}\{\mathcal{E}\}\epsilon_* \sqrt{T} \sum_{t=T_0+1}^{T} \sqrt{\sum_{k=1}^{m_t} \mathbb{I}\{i_t \in \tilde{V}'_{t,k}\} \|\boldsymbol{x}_{a_t}\|^2_{\overline{M}^{-1}_{\overline{V}'_{t,k},t-1}}}$$

$$(22)$$

$$\leq \mathbb{I}\{\mathcal{E}\}\epsilon_* \sqrt{T} \sqrt{\sum_{t=T_0+1}^{T} 1 \sum_{t=T_0+1}^{T} \sum_{k=1}^{m_t} \mathbb{I}\{i_t \in \tilde{V}'_{t,k}\} \|\boldsymbol{x}_{a_t}\|^2_{\overline{M}^{-1}_{\overline{V}'_{t,k},t-1}}}$$

$$(23)$$

$$\leq \mathbb{I}\{\mathcal{E}\}\epsilon_* \sqrt{T} \sqrt{T \sum_{t=T_0+1}^{T} \sum_{j=1}^{m} \mathbb{I}\{i_t \in V_j\} \|\boldsymbol{x}_{a_t}\|^2_{\overline{M}^{-1}_{V_j,t-1}}}$$

$$(24)$$

$$= \mathbb{I}\{\mathcal{E}\}\epsilon_* T \sqrt{\sum_{j=1}^{m} \sum_{t=T_0+1}^{T} \mathbb{I}\{i_t \in V_j\} \|\boldsymbol{x}_{a_t}\|^2_{\overline{M}^{-1}_{V_j,t-1}}}$$

$$\leq \mathbb{I}\{\mathcal{E}\}\epsilon_* T \sqrt{2md\log(1+\frac{T}{\lambda d})}, \qquad (25)$$

where Eq.(19), Eq.(21) and Eq.(23) hold because of the Cauchy–Schwarz inequality, Eq.(20) holds since $\overline{M}_{\overline{V}'_{t,k},t-1} \succeq \sum_{\substack{s\in[t-1] \\ i_s \in \overline{V}'_{t,k}}} x_{a_s} x_{a_s}^\top$, Eq.(22) is because $T_{\overline{V}'_{t,k},t-1} \leq T$, Eq. (24) follows from the same reasoning as Eq.(15), and Eq.(25) comes from the following technical Lemma J.2.

Finally, plugging Eq.(18) and Eq.(25) into Eq.(14), take expectation and plug it into Eq.(13), we can get:

$$
\begin{aligned}
R(T) \leq & 5 + T_0 + \frac{\tilde{u}}{u} \times \frac{2\epsilon_* \sqrt{2dT}}{\tilde{\lambda}_x^{\frac{3}{2}}} + 2\epsilon_* T\left(1 + \sqrt{2md\log(1 + \frac{T}{\lambda d})}\right) \\
& + 2\left(\sqrt{\lambda} + \sqrt{2\log(T) + d\log(1 + \frac{T}{\lambda d})}\right) \times \sqrt{2mdT\log(1 + \frac{T}{\lambda d})},
\end{aligned}
\tag{26}
$$

where

$$
T_0 = 16u\log(\frac{u}{\delta}) + 4u\max\max\{\frac{8d}{\tilde{\lambda}_x(\frac{\gamma_1}{4} - \epsilon_*\sqrt{\frac{1}{2\tilde{\lambda}_x}})^2}\log(\frac{u}{\delta}), \frac{16}{\tilde{\lambda}_x^2}\log(\frac{8d}{\tilde{\lambda}_x^2\delta})\}
$$

is given in the following Lemma H.1 in Appendix H.

## F  Proof and Discussions of Theorem 5.4

In the work [23], they give a lower bound for misspecified linear bandits with a single user. The lower bound of $R(T)$ is given by: $R_3(T) \geq \epsilon_* T\sqrt{d}$. Therefore, suppose our problem with multiple users and $m$ underlying clusters where the arrival times are $T_i$ for each cluster, then for any algorithms, even if they know the underlying clustering structure and keep $m$ independent linear bandit algorithms to leverage the common information of clusters, the best they can get is $R(T) = \sum_{i\in[m]} R_3(T_i) \geq \epsilon_* \sum_{i\in[m]} T_i\sqrt{d} = \epsilon_* T\sqrt{d}$, which gives a lower bound of $O(\epsilon_* T\sqrt{d})$ for the CBMUM problem. Recall that the regret upper bound of our algorithms is of $O(\epsilon_* T\sqrt{md\log T} + d\sqrt{mT}\log T)$, asymptotically matching this lower bound with respect to $T$ up to logarithmic factors and with respect to $m$ up to $O(\sqrt{m})$ factors, showing the tightness of our theoretical results (where $m$ are typically very small for real applications).

We conjecture that the gap for the $m$ factor is due to the strong assumption that cluster structures are known to prove our lower bound, and whether there exists a tighter lower bound will be left for future work.

## G  Proof of the key Lemma 5.7

In Lemma 5.7, we want to bound the term $\left|x_a^\top \overline{M}_{\overline{V}_t,t-1}^{-1} \sum_{\substack{s\in[t-1] \\ i_s \in \overline{V}_t}} x_{a_s} x_{a_s}^\top (\theta_{i_s} - \theta_{i_t})\right|$. By the definition of "good partition", we have $\|\theta_{i_s} - \theta_{i_t}\|_2 \leq \zeta, \forall i_s \in \overline{V}_t$. It is an easy-to-be-made mistake to directly drag $\|\theta_{i_s} - \theta_{i_t}\|_2$ out to upper bound it by $\left\|x_a^\top \overline{M}_{\overline{V}_t,t-1}^{-1} \sum_{\substack{s\in[t-1] \\ i_s \in \overline{V}_t}} x_{a_s} x_{a_s}^\top\right\|_2 \times \zeta$ and then proceed. We need more careful analysis.

We first prove the following general lemma.

**Lemma G.1.** *For vectors $x_1, x_2, \ldots, x_k \in \mathbb{R}^d, \|x_i\|_2 \leq 1, \forall i \in [k]$, and vectors $\theta_1, \theta_2, \ldots, \theta_k \in \mathbb{R}^d, \|\theta_i\|_2 \leq C, \forall i \in [k]$, where $C > 0$ is a constant, we have:*

$$
\left\|\sum_{i=1}^k x_i x_i^\top \theta_i\right\|_2 \leq C\sqrt{d}\left\|\sum_{i=1}^k x_i x_i^\top\right\|_2.
$$

*Proof.* Let $\boldsymbol{X} \in \mathbb{R}^{d \times k}$ be a matrix such that it has $\boldsymbol{x}_i$ s as its columns, i.e., $\boldsymbol{X} = [\boldsymbol{x}_1, \dots, \boldsymbol{x}_k] =$
$\begin{bmatrix} x_{11} & x_{21} & \cdots & x_{k1} \\ x_{12} & x_{22} & \cdots & x_{k2} \\ \vdots & \vdots & \ddots & \vdots \\ x_{1d} & x_{2d} & \cdots & x_{kd} \end{bmatrix}$.

Let $\boldsymbol{y} \in \mathbb{R}^{k \times 1}$ be a vector that has $\boldsymbol{x}_i^\top \boldsymbol{\theta}_i$ s as its elements, i.e., $\boldsymbol{y} = [\boldsymbol{x}_1^\top \boldsymbol{\theta}_1, \dots, \boldsymbol{x}_k^\top \boldsymbol{\theta}_k]^\top$. Then we have:

$$\left\| \sum_{i=1}^{k} \boldsymbol{x}_i \boldsymbol{x}_i^\top \boldsymbol{\theta}_i \right\|_2^2 = \|\boldsymbol{X} \boldsymbol{y}\|_2^2 \leq \|\boldsymbol{X}\|_2^2 \|\boldsymbol{y}\|_2^2 \tag{27}$$

$$= \|\boldsymbol{X}\|_2^2 \sum_{i=1}^{k} (\boldsymbol{x}_i^\top \boldsymbol{\theta}_i)^2$$

$$\leq \|\boldsymbol{X}\|_2^2 \sum_{i=1}^{k} \|\boldsymbol{x}_i\|_2^2 \|\boldsymbol{\theta}_i\|_2^2 \tag{28}$$

$$\leq C^2 \|\boldsymbol{X}\|_2^2 \sum_{i=1}^{k} \|\boldsymbol{x}_i\|_2^2$$

$$= C^2 \|\boldsymbol{X}\|_2^2 \|\boldsymbol{X}\|_F^2$$

$$\leq C^2 d \|\boldsymbol{X}\|_2^4 \tag{29}$$

$$= C^2 d \left\| \boldsymbol{X} \boldsymbol{X}^\top \right\|_2^2 \tag{30}$$

$$= C^2 d \left\| \sum_{i=1}^{k} \boldsymbol{x}_i \boldsymbol{x}_i^\top \right\|_2^2, \tag{31}$$

where Eq. (27) follows by the matrix operator norm inequality, Eq. (28) follows by the Cauchy–Schwarz inequality, Eq. (29) follows by $\|\boldsymbol{X}\|_F \leq \sqrt{d} \|\boldsymbol{X}\|_2$, Eq. (30) follows from $\|\boldsymbol{X}\|_2^2 = \left\| \boldsymbol{X} \boldsymbol{X}^\top \right\|_2$. $\qquad \square$

The above result is tight. We can show that the lower bound of $\left\| \sum_{i=1}^{k} \boldsymbol{x}_i \boldsymbol{x}_i^\top \boldsymbol{\theta}_i \right\|_2$ under the conditions in the lemma is exactly $C \sqrt{d} \left\| \sum_{i=1}^{k} \boldsymbol{x}_i \boldsymbol{x}_i^\top \right\|_2$. Specifically, let $k = 2$, $C = 1$, $d = 2$, $\boldsymbol{x}_1 = [0, 1]^\top$, $\boldsymbol{x}_2 = [1, 0]^\top$, $\boldsymbol{\theta}_1 = [1, 0]^\top$, $\boldsymbol{\theta}_2 = [0, 1]^\top$, then we have $\left\| \sum_{i=1}^{2} \boldsymbol{x}_i \boldsymbol{x}_i^\top \boldsymbol{\theta}_i \right\|_2 = \left\| [1, 1]^\top \right\|_2 = \sqrt{2}$, and $C \sqrt{d} \left\| \sum_{i=1}^{2} \boldsymbol{x}_i \boldsymbol{x}_i^\top \right\|_2 = 1 \times \sqrt{2} \times \left\| \begin{bmatrix} 1 & 0 \\ 0 & 1 \end{bmatrix} \right\|_2 = \sqrt{2}$. Therefore, we have that the upper bound given in Lemma G.1 matches the lower bound.

We are now ready to prove the key Lemma 5.7 with the above Lemma G.1.

At any $t \geq T_0$, if the current partition is a "good partition", and $\overline{V}_t \notin \mathcal{V}$, then for all $\boldsymbol{x}_a \in \mathbb{R}^d, \|\boldsymbol{x}_a\|_2 \leq 1$, with probability at least $1 - \delta$:

$$\left| \boldsymbol{x}_a^\top \overline{\boldsymbol{M}}_{\overline{V}_t,t-1}^{-1} \sum_{\substack{s \in [t-1] \\ i_s \in \overline{V}_t}} \boldsymbol{x}_{a_s} \boldsymbol{x}_{a_s}^\top (\boldsymbol{\theta}_{i_s} - \boldsymbol{\theta}_{i_t}) \right| \leq \|\boldsymbol{x}_a\|_2 \left\| \overline{\boldsymbol{M}}_{\overline{V}_t,t-1}^{-1} \sum_{\substack{s \in [t-1] \\ i_s \in \overline{V}_t}} \boldsymbol{x}_{a_s} \boldsymbol{x}_{a_s}^\top (\boldsymbol{\theta}_{i_s} - \boldsymbol{\theta}_{i_t}) \right\|_2 \quad (32)$$

$$\leq \left\| \overline{\boldsymbol{M}}_{\overline{V}_t,t-1}^{-1} \right\|_2 \left\| \sum_{\substack{s \in [t-1] \\ i_s \in \overline{V}_t}} \boldsymbol{x}_{a_s} \boldsymbol{x}_{a_s}^\top (\boldsymbol{\theta}_{i_s} - \boldsymbol{\theta}_{i_t}) \right\|_2 \quad (33)$$

$$\leq 2\epsilon_* \sqrt{\frac{2d}{\tilde{\lambda}_x}} \times \left\| \overline{\boldsymbol{M}}_{\overline{V}_t,t-1}^{-1} \right\|_2 \left\| \sum_{\substack{s \in [t-1] \\ i_s \in \overline{V}_t}} \boldsymbol{x}_{a_s} \boldsymbol{x}_{a_s}^\top \right\|_2 \quad (34)$$

$$\leq 2\epsilon_* \sqrt{\frac{2d}{\tilde{\lambda}_x}} \times \frac{\lambda_{max}(\sum_{\substack{s \in [t-1] \\ i_s \in \overline{V}_t}} \boldsymbol{x}_{a_s} \boldsymbol{x}_{a_s}^\top)}{\lambda_{\min}(\overline{\boldsymbol{M}}_{\overline{V}_t,t-1})}$$

$$\leq 2\epsilon_* \sqrt{\frac{2d}{\tilde{\lambda}_x}} \times \frac{T_{\overline{V}_t,t-1}}{2T_{\overline{V}_t,t-1}\tilde{\lambda}_x + \lambda} \quad (35)$$

$$\leq \frac{\epsilon_* \sqrt{2d}}{\tilde{\lambda}_x^{\frac{3}{2}}},$$

where Eq.(32) follows by the Cauchy–Schwarz inequality, Eq.(33) follows from the inequality of matrix's operator norm, Eq.(34) follows from the fact that in a "good partition", $\|\boldsymbol{\theta}_{i_t} - \boldsymbol{\theta}_l\|_2 \leq 2\epsilon_* \sqrt{\frac{2}{\tilde{\lambda}_x}}, \forall l \in \overline{V}_t$ and Lemma G.1, Eq.(35) follows by Eq.(47) with probability $\geq 1 - \delta$.

# H   Lemma H.1 of the sufficient time $T_0$ and its proof

The following lemma gives a sufficient time $T_0$ for the algorithm to get a "good partition".

**Lemma H.1.** *With the carefully designed edge deletion rule, after*

$$T_0 \triangleq 16u \log(\frac{u}{\delta}) + 4u \max \max \{ \frac{8d}{\tilde{\lambda}_x(\frac{\gamma_1}{4} - \epsilon_* \sqrt{\frac{1}{2\tilde{\lambda}_x}})^2} \log(\frac{u}{\delta}), \frac{16}{\tilde{\lambda}_x^2} \log(\frac{8d}{\tilde{\lambda}_x^2 \delta}) \}$$

$$= O\left( u \left( \frac{d}{\tilde{\lambda}_x(\gamma_1 - \zeta)^2} + \frac{1}{\tilde{\lambda}_x^2} \right) \log \frac{1}{\delta} \right)$$

*rounds, with probability at least $1 - 3\delta$ for some $\delta \in (0, \frac{1}{3})$, RCLUMB can always get a "good partition".*

Below is the detailed proof of Lemma H.1.

*Proof.* We first prove the following result:
With probability at least $1 - \delta$ for some $\delta \in (0, 1)$, at any $t \in [T]$:

$$\left\| \hat{\boldsymbol{\theta}}_{i,t} - \boldsymbol{\theta}^{j(i)} \right\|_2 \leq \frac{\beta(T_{i,t}, \frac{\delta}{u}) + \epsilon_* \sqrt{T_{i,t}}}{\sqrt{\lambda + \lambda_{\min}(\boldsymbol{M}_{i,t})}}, \forall i \in \mathcal{U}, \quad (36)$$

where $\beta(T_{i,t}, \frac{\delta}{u}) \triangleq \sqrt{\lambda} + \sqrt{2\log(\frac{u}{\delta}) + d\log(1 + \frac{T_{i,t}}{\lambda d})}$.

$$\hat{\boldsymbol{\theta}}_{i,t} - \boldsymbol{\theta}^{j(i)} = (\sum_{\substack{s \in [t] \\ i_s = i}} \boldsymbol{x}_{a_s} \boldsymbol{x}_{a_s}^\top + \lambda \boldsymbol{I})^{-1} \left( \sum_{\substack{s \in [t] \\ i_s = i}} \boldsymbol{x}_{a_s} (\boldsymbol{x}_{a_s}^\top \boldsymbol{\theta}^{j(i)} + \boldsymbol{\epsilon}_{a_s}^{i_s, s} + \eta_s) \right) - \boldsymbol{\theta}^{j(i)} \tag{37}$$

$$= (\sum_{\substack{s \in [t] \\ i_s = i}} \boldsymbol{x}_{a_s} \boldsymbol{x}_{a_s}^\top + \lambda \boldsymbol{I})^{-1}[(\sum_{\substack{s \in [t] \\ i_s = i}} \boldsymbol{x}_{a_s} \boldsymbol{x}_{a_s}^\top + \lambda \boldsymbol{I})\boldsymbol{\theta}^{j(i)} - \lambda\boldsymbol{\theta}^{j(i)} + \sum_{\substack{s \in [t] \\ i_s = i}} \boldsymbol{x}_{a_s} \boldsymbol{\epsilon}_{a_s}^{i_s, s} + \sum_{\substack{s \in [t] \\ i_s = i}} \boldsymbol{x}_{a_s} \eta_s] - \boldsymbol{\theta}^{j(i)}$$

$$= -\lambda \tilde{\boldsymbol{M}}_{i,t}^{-1} \boldsymbol{\theta}^{j(i)} + \tilde{\boldsymbol{M}}_{i,t}^{-1} \sum_{\substack{s \in [t] \\ i_s = i}} \boldsymbol{x}_{a_s} \boldsymbol{\epsilon}_{a_s}^{i_s, s} + \tilde{\boldsymbol{M}}_{i,t}^{-1} \sum_{\substack{s \in [t] \\ i_s = i}} \boldsymbol{x}_{a_s} \eta_s \,,$$

where we denote $\tilde{\boldsymbol{M}}_{i,t} = \boldsymbol{M}_{i,t} + \lambda \boldsymbol{I}$, and Eq.(37) holds by definition.

Therefore,

$$\left\| \hat{\boldsymbol{\theta}}_{i,t} - \boldsymbol{\theta}^{j(i)} \right\|_2 \le \lambda \left\| \tilde{\boldsymbol{M}}_{i,t}^{-1} \boldsymbol{\theta}^{j(i)} \right\|_2 + \left\| \tilde{\boldsymbol{M}}_{i,t}^{-1} \sum_{\substack{s \in [t] \\ i_s = i}} \boldsymbol{x}_{a_s} \boldsymbol{\epsilon}_{a_s}^{i_s, s} \right\|_2 + \left\| \tilde{\boldsymbol{M}}_{i,t}^{-1} \sum_{\substack{s \in [t] \\ i_s = i}} \boldsymbol{x}_{a_s} \eta_s \right\|_2 . \tag{38}$$

We then bound the three terms in Eq.(38) one by one. For the first term:

$$\lambda \left\| \tilde{\boldsymbol{M}}_{i,t}^{-1} \boldsymbol{\theta}^{j(i)} \right\|_2 \le \lambda \left\| \tilde{\boldsymbol{M}}_{i,t}^{-\frac{1}{2}} \right\|_2^2 \left\| \boldsymbol{\theta}^{j(i)} \right\|_2 \le \frac{\sqrt{\lambda}}{\sqrt{\lambda_{\min}(\tilde{\boldsymbol{M}}_{i,t})}} \,, \tag{39}$$

where we use the Cauchy–Schwarz inequality, the inequality for the operator norm of matrices, and the fact that $\lambda_{\min}(\tilde{\boldsymbol{M}}_{i,t}) \ge \lambda$.

For the second term in Eq.(38):

$$\left\| \tilde{\boldsymbol{M}}_{i,t}^{-1} \sum_{\substack{s \in [t] \\ i_s = i}} \boldsymbol{x}_{a_s} \boldsymbol{\epsilon}_{a_s}^{i_s, s} \right\|_2 = \max_{\boldsymbol{x} \in S^{d-1}} \sum_{\substack{s \in [t] \\ i_s = i}} \boldsymbol{x}^\top \tilde{\boldsymbol{M}}_{i,t}^{-1} \boldsymbol{x}_{a_s} \boldsymbol{\epsilon}_{a_s}^{i_s, s}$$

$$\le \max_{\boldsymbol{x} \in S^{d-1}} \sum_{\substack{s \in [t] \\ i_s = i}} \left| \boldsymbol{x}^\top \tilde{\boldsymbol{M}}_{i,t}^{-1} \boldsymbol{x}_{a_s} \boldsymbol{\epsilon}_{a_s}^{i_s, s} \right|$$

$$\le \max_{\boldsymbol{x} \in S^{d-1}} \sum_{\substack{s \in [t] \\ i_s = i}} \left| \boldsymbol{x}^\top \tilde{\boldsymbol{M}}_{i,t}^{-1} \boldsymbol{x}_{a_s} \right| \left\| \boldsymbol{\epsilon}_{a_s}^{i_s, s} \right\|_\infty \tag{40}$$

$$\le \epsilon_* \max_{\boldsymbol{x} \in S^{d-1}} \sum_{\substack{s \in [t] \\ i_s = i}} \left| \boldsymbol{x}^\top \tilde{\boldsymbol{M}}_{i,t}^{-1} \boldsymbol{x}_{a_s} \right|$$

$$\le \epsilon_* \max_{\boldsymbol{x} \in S^{d-1}} \sqrt{\sum_{\substack{s \in [t] \\ i_s = i}} 1 \sum_{\substack{s \in [t] \\ i_s = i}} \left| \boldsymbol{x}^\top \tilde{\boldsymbol{M}}_{i,t}^{-1} \boldsymbol{x}_{a_s} \right|^2} \tag{41}$$

$$\le \epsilon_* \sqrt{T_{i,t}} \sqrt{\max_{\boldsymbol{x} \in S^{d-1}} \boldsymbol{x}^\top \tilde{\boldsymbol{M}}_{i,t}^{-1} \boldsymbol{x}} \tag{42}$$

$$= \frac{\epsilon_* \sqrt{T_{i,t}}}{\sqrt{\lambda_{\min}(\tilde{\boldsymbol{M}}_{i,t})}} \,, \tag{43}$$

where we denote $S^{d-1} = \{ \boldsymbol{x} \in \mathbb{R}^d : \| \boldsymbol{x} \|_2 = 1 \}$, Eq.(40) follows from Holder's inequality, Eq.(41) follows by the Cauchy–Schwarz inequality, Eq.(42) holds because $\tilde{\boldsymbol{M}}_{i,t} \succeq \sum_{\substack{s \in [t] \\ i_s = i}} \boldsymbol{x}_{a_s} \boldsymbol{x}_{a_s}^\top$, Eq.(43) follows from the Courant-Fischer theorem.

For the last term in Eq.(38)

$$\left\| \tilde{\boldsymbol{M}}_{i,t}^{-1} \sum_{\substack{s \in [t] \\ i_s = i}} \boldsymbol{x}_{a_s} \eta_s \right\|_2 \leq \left\| \tilde{\boldsymbol{M}}_{i,t}^{-\frac{1}{2}} \sum_{\substack{s \in [t] \\ i_s = i}} \boldsymbol{x}_{a_s} \eta_s \right\|_2 \left\| \tilde{\boldsymbol{M}}_{i,t}^{-\frac{1}{2}} \right\|_2 \tag{44}$$

$$= \frac{\left\| \sum_{\substack{s \in [t] \\ i_s = i}} \boldsymbol{x}_{a_s} \eta_s \right\|_{\tilde{\boldsymbol{M}}_{i,t}^{-1}}}{\sqrt{\lambda_{\min}(\tilde{\boldsymbol{M}}_{i,t})}}, \tag{45}$$

where Eq.(44) follows by the Cauchy–Schwarz inequality and the inequality for the operator norm of matrices, and Eq.(45) follows by the Courant-Fischer theorem.

Following Theorem 1 in [1], with probability at least $1 - \delta$ for some $\delta \in (0,1)$, for any $i \in \mathcal{U}$, we have:

$$\left\| \sum_{\substack{s \in [t] \\ i_s = i}} \boldsymbol{x}_{a_s} \eta_s \right\|_{\tilde{\boldsymbol{M}}_{i,t}^{-1}} \leq \sqrt{2 \log(\frac{u}{\delta}) + \log(\frac{\det(\tilde{\boldsymbol{M}}_{i,t})}{\det(\lambda \boldsymbol{I})})}$$

$$\leq \sqrt{2 \log(\frac{u}{\delta}) + d \log(1 + \frac{T_{i,t}}{\lambda d})}, \tag{46}$$

where $\det(\boldsymbol{M})$ denotes the determinant of matrix $\boldsymbol{M}$, Eq.(46) is because $\det(\tilde{\boldsymbol{M}}_{i,t}) \leq \left( \frac{\text{trace}(\lambda \boldsymbol{I} + \sum_{\substack{s \in [t] \\ i_s = i}} \boldsymbol{x}_{a_s} \boldsymbol{x}_{a_s}^\top)}{d} \right)^d \leq \left( \frac{\lambda d + T_{i,t}}{d} \right)^d$, and $\det(\lambda \boldsymbol{I}) = \lambda^d$.

Plugging Eq.(46) into Eq. (45), then plugging Eq. (39), Eq.(43) and Eq.(45) into Eq.(38), we can get that Eq.(73) holds with probability $\geq 1 - \delta$.

Then, with the item regularity assumption stated in Assumption 3.3, the technical Lemma J.1, together with Lemma 7 in [25], with probability at least $1 - \delta$, for a particular user $i$, at any $t$ such that $T_{i,t} \geq \frac{16}{\tilde{\lambda}_x^2} \log(\frac{8d}{\tilde{\lambda}_x^2 \delta})$, we have:

$$\lambda_{\min}(\tilde{\boldsymbol{M}}_{i,t}) \geq 2 \tilde{\lambda}_x T_{i,t} + \lambda. \tag{47}$$

Based on the above reasoning, we have: if $T_{i,t} \geq \frac{16}{\tilde{\lambda}_x^2} \log(\frac{8d}{\tilde{\lambda}_x^2 \delta})$, then with probability $\geq 1 - 2\delta$, we have:

$$\left\| \hat{\boldsymbol{\theta}}_{i,t} - \boldsymbol{\theta}^{j(i)} \right\|_2 \leq \frac{\beta(T_{i,t}, \frac{\delta}{u}) + \epsilon_* \sqrt{T_{i,t}}}{\sqrt{\lambda_{\min}(\tilde{\boldsymbol{M}}_{i,t})}}$$

$$\leq \frac{\beta(T_{i,t}, \frac{\delta}{u}) + \epsilon_* \sqrt{T_{i,t}}}{\sqrt{2 \tilde{\lambda}_x T_{i,t} + \lambda}}$$

$$\leq \frac{\sqrt{\lambda} + \sqrt{2 \log(\frac{u}{\delta}) + d \log(1 + \frac{T_{i,t}}{\lambda d})}}{\sqrt{2 \tilde{\lambda}_x T_{i,t} + \lambda}} + \epsilon_* \sqrt{\frac{1}{2 \tilde{\lambda}_x}}, \tag{48}$$

for any $i \in \mathcal{U}$.

Let

$$\frac{\sqrt{\lambda} + \sqrt{2 \log(\frac{u}{\delta}) + d \log(1 + \frac{T_{i,t}}{\lambda d})}}{\sqrt{2 \tilde{\lambda}_x T_{i,t} + \lambda}} + \epsilon_* \sqrt{\frac{1}{2 \tilde{\lambda}_x}} < \frac{\gamma_1}{4}, \tag{49}$$

which is equivalent to

$$\frac{\sqrt{\lambda} + \sqrt{2 \log(\frac{u}{\delta}) + d \log(1 + \frac{T_{i,t}}{\lambda d})}}{\sqrt{2 \tilde{\lambda}_x T_{i,t} + \lambda}} < \frac{\gamma_1}{4} - \epsilon_* \sqrt{\frac{1}{2 \tilde{\lambda}_x}}, \tag{50}$$

where $\gamma_1$ is given in Definition 5.1.

Assume $\lambda \le 2\log(\frac{u}{\delta}) + d\log(1 + \frac{T_{i,t}}{\lambda d})$, which is typically held, then a sufficient condition for Eq. (50) is:

$$\frac{2\log(\frac{u}{\delta}) + d\log(1 + \frac{T_{i,t}}{\lambda d})}{2\tilde{\lambda}_x T_{i,t}} < \frac{1}{4}(\frac{\gamma_1}{4} - \epsilon_* \sqrt{\frac{1}{2\tilde{\lambda}_x}})^2 . \tag{51}$$

To satisfy the condition in Eq. (51), it is sufficient to show

$$\frac{2\log(\frac{u}{\delta})}{2\tilde{\lambda}_x T_{i,t}} < \frac{1}{8}(\frac{\gamma_1}{4} - \epsilon_* \sqrt{\frac{1}{2\tilde{\lambda}_x}})^2 \tag{52}$$

and

$$\frac{d\log(1 + \frac{T_{i,t}}{\lambda d})}{2\tilde{\lambda}_x T_{i,t}} < \frac{1}{8}(\frac{\gamma_1}{4} - \epsilon_* \sqrt{\frac{1}{2\tilde{\lambda}_x}})^2 . \tag{53}$$

From Eq. (52), we can get:

$$T_{i,t} \ge \frac{8\log(\frac{u}{\delta})}{\tilde{\lambda}_x(\frac{\gamma_1}{4} - \epsilon_* \sqrt{\frac{1}{2\tilde{\lambda}_x}})^2} . \tag{54}$$

Following Lemma 9 in [25], we can get the following sufficient condition for Eq. (53):

$$T_{i,t} \ge \frac{8d\log(\frac{4}{\lambda\tilde{\lambda}_x(\frac{\gamma_1}{4} - \epsilon_* \sqrt{\frac{1}{2\tilde{\lambda}_x}})^2})}{\tilde{\lambda}_x(\frac{\gamma_1}{4} - \epsilon_* \sqrt{\frac{1}{2\tilde{\lambda}_x}})^2} . \tag{55}$$

Assume $\frac{u}{\delta} \ge \frac{4}{\lambda\tilde{\lambda}_x(\frac{\gamma_1}{4} - \epsilon_* \sqrt{\frac{1}{2\tilde{\lambda}_x}})^2}$, which is typically held, we can get that

$$T_{i,t} \ge \frac{8d}{\tilde{\lambda}_x(\frac{\gamma_1}{4} - \epsilon_* \sqrt{\frac{1}{2\tilde{\lambda}_x}})^2} \log(\frac{u}{\delta}) \tag{56}$$

is a sufficient condition for Eq. (49). Together with the condition that $T_{i,t} \ge \frac{16}{\tilde{\lambda}_x^2}\log(\frac{8d}{\tilde{\lambda}_x^2 \delta})$, we can get that if

$$T_{i,t} \ge \max\{\frac{8d}{\tilde{\lambda}_x(\frac{\gamma_1}{4} - \epsilon_* \sqrt{\frac{1}{2\tilde{\lambda}_x}})^2} \log(\frac{u}{\delta}), \frac{16}{\tilde{\lambda}_x^2}\log(\frac{8d}{\tilde{\lambda}_x^2 \delta})\}, \forall i \in \mathcal{U}, \tag{57}$$

then with probability $\ge 1 - 2\delta$:

$$\left\|\hat{\boldsymbol{\theta}}_{i,t} - \boldsymbol{\theta}^{j(i)}\right\|_2 < \frac{\gamma_1}{4}, \forall i \in \mathcal{U} .$$

By Lemma 8 in [25], and Assumption 3.2 of user arrival uniformness, we have that for all

$$t \ge T_0 \triangleq 16u\log(\frac{u}{\delta}) + 4u\max\{\frac{8d}{\tilde{\lambda}_x(\frac{\gamma_1}{4} - \epsilon_* \sqrt{\frac{1}{2\tilde{\lambda}_x}})^2} \log(\frac{u}{\delta}), \frac{16}{\tilde{\lambda}_x^2}\log(\frac{8d}{\tilde{\lambda}_x^2 \delta})\}, \tag{58}$$

with probability at least $1 - \delta$, condition in Eq. (57) is satisfied.

Therefore we have that for all $t \ge T_0$, with probability $\ge 1 - 3\delta$:

$$\left\|\hat{\boldsymbol{\theta}}_{i,t} - \boldsymbol{\theta}^{j(i)}\right\|_2 < \frac{\gamma_1}{4}, \forall i \in \mathcal{U} . \tag{59}$$

Next, we show that with Eq. (59), we can get that the RCLUMB keeps a "good partition". First, if we delete the edge $(i, l)$, then user $i$ and user $j$ belong to different *ground-truth clusters*, i.e., $\|\boldsymbol{\theta}_i - \boldsymbol{\theta}_l\|_2 > 0$. This is because by the deletion rule of the algorithm, the concentration bound, and triangle inequality, $\|\boldsymbol{\theta}_i - \boldsymbol{\theta}_l\|_2 = \left\|\boldsymbol{\theta}^{j(i)} - \boldsymbol{\theta}^{j(l)}\right\|_2 \ge \left\|\hat{\boldsymbol{\theta}}_{i,t} - \hat{\boldsymbol{\theta}}_{l,t}\right\|_2 - \left\|\boldsymbol{\theta}^{j(l)} - \boldsymbol{\theta}_{l,t}\right\|_2 -$

$\left\|\boldsymbol{\theta}^{j(i)} - \boldsymbol{\theta}_{i,t}\right\|_2 > 0$. Second, we show that if $\|\boldsymbol{\theta}_i - \boldsymbol{\theta}_l\| \geq \gamma_1 > 2\epsilon_*\sqrt{\frac{2}{\tilde{\lambda}_x}}$, the RCLUMB algorithm will delete the edge $(i, l)$. This is because if $\|\boldsymbol{\theta}_i - \boldsymbol{\theta}_l\| \geq \gamma_1$, then by the triangle inequality, and $\left\|\hat{\boldsymbol{\theta}}_{i,t} - \boldsymbol{\theta}^{j(i)}\right\|_2 < \frac{\gamma_1}{4}$, $\left\|\hat{\boldsymbol{\theta}}_{l,t} - \boldsymbol{\theta}^{j(l)}\right\|_2 < \frac{\gamma_1}{4}$, $\boldsymbol{\theta}_i = \boldsymbol{\theta}^{j(i)}$, $\boldsymbol{\theta}_l = \boldsymbol{\theta}^{j(l)}$, we have $\left\|\hat{\boldsymbol{\theta}}_{i,t} - \hat{\boldsymbol{\theta}}_{l,t}\right\|_2 \geq \|\boldsymbol{\theta}_i - \boldsymbol{\theta}_l\| - \left\|\hat{\boldsymbol{\theta}}_{i,t} - \boldsymbol{\theta}^{j(i)}\right\|_2 - \left\|\hat{\boldsymbol{\theta}}_{l,t} - \boldsymbol{\theta}^{j(l)}\right\|_2 > \gamma_1 - \frac{\gamma_1}{4} - \frac{\gamma_1}{4} = \frac{\gamma_1}{2} > \frac{\sqrt{\lambda} + \sqrt{2\log(\frac{u}{\delta}) + d\log(1 + \frac{T_{i,t}}{\lambda d})}}{\sqrt{\lambda + 2\tilde{\lambda}_x T_{i,t}}} + \epsilon_*\sqrt{\frac{1}{2\tilde{\lambda}_x}} + \frac{\sqrt{\lambda} + \sqrt{2\log(\frac{u}{\delta}) + d\log(1 + \frac{T_{l,t}}{\lambda d})}}{\sqrt{\lambda + 2\tilde{\lambda}_x T_{l,t}}} + \epsilon_*\sqrt{\frac{1}{2\tilde{\lambda}_x}}$, which will trigger the deletion condition Line 10 in Algo.1.

From the above reasoning, we can get that at round $t$, any user within $\overline{V}_t$ is $\zeta$-close to $i_t$, and all the users belonging to $V_{j(i)}$ are contained in $\overline{V}_t$, which means the algorithm has done a "good partition" at $t$ by Definition 5.5. $\qquad\square$

# I  Proof of Lemma 5.6

We prove the result in two situations: when $\overline{V}_t \in \mathcal{V}$ and when $\overline{V}_t \notin \mathcal{V}$.

(1) Situation 1: for any $t \geq T_0$ and $\overline{V}_t \in \mathcal{V}$, which means that the current user $i_t$ is clustered completely correctly, i.e., $\overline{V}_t = V_{j(i_t)}$, therefore $\boldsymbol{\theta}_l = \boldsymbol{\theta}_{i_t}, \forall l \in \overline{V}_t$, then we have:

$$
\begin{aligned}
\hat{\boldsymbol{\theta}}_{\overline{V}_t, t-1} - \boldsymbol{\theta}_{i_t} &= \Big( \sum_{\substack{s \in [t-1] \\ i_s \in \overline{V}_t}} \boldsymbol{x}_{a_s} \boldsymbol{x}_{a_s}^\top + \lambda \boldsymbol{I} \Big)^{-1} \Big( \sum_{\substack{s \in [t-1] \\ i_s \in \overline{V}_t}} \boldsymbol{x}_{a_s} r_s \Big) - \boldsymbol{\theta}_{i_t} \\
&= \Big( \sum_{\substack{s \in [t-1] \\ i_s \in \overline{V}_t}} \boldsymbol{x}_{a_s} \boldsymbol{x}_{a_s}^\top + \lambda \boldsymbol{I} \Big)^{-1} \Big( \sum_{\substack{s \in [t-1] \\ i_s \in \overline{V}_t}} \boldsymbol{x}_{a_s} (\boldsymbol{x}_{a_s}^\top \boldsymbol{\theta}_{i_s} + \boldsymbol{\epsilon}_{a_s}^{i_s,s} + \eta_s) \Big) - \boldsymbol{\theta}_{i_t} \\
&= \Big( \sum_{\substack{s \in [t-1] \\ i_s \in \overline{V}_t}} \boldsymbol{x}_{a_s} \boldsymbol{x}_{a_s}^\top + \lambda \boldsymbol{I} \Big)^{-1} \Big( \sum_{\substack{s \in [t-1] \\ i_s \in \overline{V}_t}} \boldsymbol{x}_{a_s} (\boldsymbol{x}_{a_s}^\top \boldsymbol{\theta}_{i_t} + \boldsymbol{\epsilon}_{a_s}^{i_s,s} + \eta_s) \Big) - \boldsymbol{\theta}_{i_t} \\
&= \Big( \sum_{\substack{s \in [t-1] \\ i_s \in \overline{V}_t}} \boldsymbol{x}_{a_s} \boldsymbol{x}_{a_s}^\top + \lambda \boldsymbol{I} \Big)^{-1} \Big[ \Big( \sum_{\substack{s \in [t-1] \\ i_s \in \overline{V}_t}} \boldsymbol{x}_{a_s} \boldsymbol{x}_{a_s}^\top + \lambda \boldsymbol{I} \Big) \boldsymbol{\theta}_{i_t} - \lambda \boldsymbol{\theta}_{i_t} + \sum_{\substack{s \in [t-1] \\ i_s \in \overline{V}_t}} \boldsymbol{x}_{a_s} \boldsymbol{\epsilon}_{a_s}^{i_s,s} + \sum_{\substack{s \in [t-1] \\ i_s \in \overline{V}_t}} \boldsymbol{x}_{a_s} \eta_s \Big] - \boldsymbol{\theta}_{i_t} \\
&= -\lambda \overline{\boldsymbol{M}}_{\overline{V}_t, t-1}^{-1} \boldsymbol{\theta}_{i_t} + \sum_{\substack{s \in [t-1] \\ i_s \in \overline{V}_t}} \overline{\boldsymbol{M}}_{\overline{V}_t, t-1}^{-1} \boldsymbol{x}_{a_s} \boldsymbol{\epsilon}_{a_s}^{i_s,s} + \sum_{\substack{s \in [t-1] \\ i_s \in \overline{V}_t}} \overline{\boldsymbol{M}}_{\overline{V}_t, t-1}^{-1} \boldsymbol{x}_{a_s} \eta_s \, .
\end{aligned}
$$

Therefore we have

$$
\left| \boldsymbol{x}_a^\top (\hat{\boldsymbol{\theta}}_{\overline{V}_t, t-1} - \boldsymbol{\theta}_{i_t}) \right| \leq \lambda \left| \boldsymbol{x}_a^\top \overline{\boldsymbol{M}}_{\overline{V}_t, t-1}^{-1} \boldsymbol{\theta}_{i_t} \right| + \left| \sum_{\substack{s \in [t-1] \\ i_s \in \overline{V}_t}} \boldsymbol{x}_a^\top \overline{\boldsymbol{M}}_{\overline{V}_t, t-1}^{-1} \boldsymbol{x}_{a_s} \boldsymbol{\epsilon}_{a_s}^{i_s,s} \right| + \left| \boldsymbol{x}_a^\top \overline{\boldsymbol{M}}_{\overline{V}_t, t-1}^{-1} \sum_{\substack{s \in [t-1] \\ i_s \in \overline{V}_t}} \boldsymbol{x}_{a_s} \eta_s \right| \, .
\tag{60}
$$

Next, we bound the three terms in Eq.(60). For the first term:

$$
\lambda \left| \boldsymbol{x}_a^\top \overline{\boldsymbol{M}}_{\overline{V}_t, t-1}^{-1} \boldsymbol{\theta}_{i_t} \right| \leq \lambda \|\boldsymbol{x}_a\|_{\overline{\boldsymbol{M}}_{\overline{V}_t, t-1}^{-1}} \sqrt{\lambda_{\max}(\overline{\boldsymbol{M}}_{\overline{V}_t, t-1}^{-1})} \|\boldsymbol{\theta}_{i_t}\|_2 \leq \sqrt{\lambda} \|\boldsymbol{x}_a\|_{\overline{\boldsymbol{M}}_{\overline{V}_t, t-1}^{-1}} \, ,
\tag{61}
$$

where we use the inequality of matrix norm, the Cauchy–Schwarz inequality, $\|\boldsymbol{\theta}_{i_t}\|_2 \leq 1$, and the fact that $\lambda_{\max}(\overline{\boldsymbol{M}}_{\overline{V}_t, t-1}^{-1}) = \frac{1}{\lambda_{\min}(\overline{\boldsymbol{M}}_{\overline{V}_t, t-1})} \leq \frac{1}{\lambda}$.

For the second term in Eq.(60):

$$\left| \sum_{\substack{s \in [t-1] \\ i_s \in \overline{V}_t}} \boldsymbol{x}_a^\top \overline{\boldsymbol{M}}_{\overline{V}_t, t-1}^{-1} \boldsymbol{x}_{a_s} \boldsymbol{\epsilon}_{a_s}^{i_s, s} \right| \leq \sum_{\substack{s \in [t-1] \\ i_s \in \overline{V}_t}} \left| \boldsymbol{x}_a^\top \overline{\boldsymbol{M}}_{\overline{V}_t, t-1}^{-1} \boldsymbol{x}_{a_s} \boldsymbol{\epsilon}_{a_s}^{i_s, s} \right|$$

$$\leq \sum_{\substack{s \in [t-1] \\ i_s \in \overline{V}_t}} \left\| \boldsymbol{\epsilon}_{a_s}^{i_s, s} \right\|_\infty \left| \boldsymbol{x}_a^\top \overline{\boldsymbol{M}}_{\overline{V}_t, t-1}^{-1} \boldsymbol{x}_{a_s} \right|$$

$$\leq \epsilon_* \sum_{\substack{s \in [t-1] \\ i_s \in \overline{V}_t}} \left| \boldsymbol{x}_a^\top \overline{\boldsymbol{M}}_{\overline{V}_t, t-1}^{-1} \boldsymbol{x}_{a_s} \right|, \tag{62}$$

where in the second inequality we use the Holder's inequality.

For the last term, with probability at least $1 - \delta$:

$$\left| \boldsymbol{x}_a^\top \overline{\boldsymbol{M}}_{\overline{V}_t, t-1}^{-1} \sum_{\substack{s \in [t-1] \\ i_s \in \overline{V}_t}} \boldsymbol{x}_{a_s} \eta_s \right| \leq \|\boldsymbol{x}_a\|_{\overline{\boldsymbol{M}}_{\overline{V}_t, t-1}^{-1}} \left\| \sum_{\substack{s \in [t-1] \\ i_s \in \overline{V}_t}} \boldsymbol{x}_{a_s} \eta_s \right\|_{\overline{\boldsymbol{M}}_{\overline{V}_t, t-1}^{-1}} \tag{63}$$

$$\leq \|\boldsymbol{x}_a\|_{\overline{\boldsymbol{M}}_{\overline{V}_t, t-1}^{-1}} \sqrt{2 \log(\frac{1}{\delta}) + \log(\frac{\det(\overline{\boldsymbol{M}}_{\overline{V}_t, t-1})}{\det(\lambda \boldsymbol{I})})}$$

$$\leq \|\boldsymbol{x}_a\|_{\overline{\boldsymbol{M}}_{\overline{V}_t, t-1}^{-1}} \sqrt{2 \log(\frac{1}{\delta}) + d \log(1 + \frac{T}{\lambda d})}, \tag{64}$$

where the second inequality follows by Theorem 1 in [1], Eq.(64) is because $\det(\overline{\boldsymbol{M}}_{\overline{V}_t, t-1}) \leq \left( \frac{\text{trace}(\lambda \boldsymbol{I} + \sum_{\substack{s \in [t] \\ i_s \in \overline{V}_t}} \boldsymbol{x}_{a_s} \boldsymbol{x}_{a_s}^\top)}{d} \right)^d \leq \left( \frac{\lambda d + T_{\overline{V}_t, t}}{d} \right)^d \leq \left( \frac{\lambda d + T}{d} \right)^d$, and $\det(\lambda \boldsymbol{I}) = \lambda^d$.

Plugging Eq.(61), Eq.(62) and Eq.(64) into Eq.(60), we can prove Lemma 5.6 in situation 1, i.e., for any $t \geq T_0$ and $\overline{V}_t \in V$, with probability at least $1 - \delta$:

$$\left| \boldsymbol{x}_a^\top (\hat{\boldsymbol{\theta}}_{\overline{V}_t, t-1} - \boldsymbol{\theta}_{i_t}) \right| \leq \epsilon_* \sum_{\substack{s \in [t-1] \\ i_s \in \overline{V}_t}} \left| \boldsymbol{x}_a^\top \overline{\boldsymbol{M}}_{\overline{V}_t, t-1}^{-1} \boldsymbol{x}_{a_s} \right| + \|\boldsymbol{x}_a\|_{\overline{\boldsymbol{M}}_{\overline{V}_t, t-1}^{-1}} \left( \sqrt{\lambda} + \sqrt{2 \log(\frac{1}{\delta}) + d \log(1 + \frac{T}{\lambda d})} \right).$$

$$\tag{65}$$

(2) Situation 2: for any $t \geq T_0$ and $\overline{V}_t \notin \mathcal{V}$, which means that the current user is *misclustered* by the algorithm, i.e., $\overline{V}_t \neq V_{j(i_t)}$, but with Lemma H.1, with probability at least $1 - 3\delta$, the current

partition is a "good partition", i.e., $\|\boldsymbol{\theta}_l - \boldsymbol{\theta}_{i_t}\|_2 \le 2\epsilon_* \sqrt{\frac{2}{\lambda_x}}, \forall l \in \overline{V}_t$, we have:

$$\hat{\boldsymbol{\theta}}_{\overline{V}_t, t-1} - \boldsymbol{\theta}_{i_t} = \Big( \sum_{\substack{s \in [t-1] \\ i_s \in \overline{V}_t}} \boldsymbol{x}_{a_s} \boldsymbol{x}_{a_s}^\top + \lambda \boldsymbol{I} \Big)^{-1} \Big( \sum_{\substack{s \in [t-1] \\ i_s \in \overline{V}_t}} \boldsymbol{x}_{a_s} r_s \Big) - \boldsymbol{\theta}_{i_t}$$

$$= \Big( \sum_{\substack{s \in [t-1] \\ i_s \in \overline{V}_t}} \boldsymbol{x}_{a_s} \boldsymbol{x}_{a_s}^\top + \lambda \boldsymbol{I} \Big)^{-1} \Big( \sum_{\substack{s \in [t-1] \\ i_s \in \overline{V}_t}} \boldsymbol{x}_{a_s} (\boldsymbol{x}_{a_s}^\top \boldsymbol{\theta}_{i_s} + \boldsymbol{\epsilon}_{a_s}^{i_s, s} + \eta_s) \Big) - \boldsymbol{\theta}_{i_t}$$

$$= \overline{\boldsymbol{M}}_{\overline{V}_t, t-1}^{-1} \sum_{\substack{s \in [t-1] \\ i_s \in \overline{V}_t}} \boldsymbol{x}_{a_s} \boldsymbol{\epsilon}_{a_s}^{i_s, s} + \overline{\boldsymbol{M}}_{\overline{V}_t, t-1}^{-1} \sum_{\substack{s \in [t-1] \\ i_s \in \overline{V}_t}} \boldsymbol{x}_{a_s} \eta_s + \overline{\boldsymbol{M}}_{\overline{V}_t, t-1}^{-1} \sum_{\substack{s \in [t-1] \\ i_s \in \overline{V}_t}} \boldsymbol{x}_{a_s} \boldsymbol{x}_{a_s}^\top \boldsymbol{\theta}_{i_s} - \boldsymbol{\theta}_{i_t}$$

$$= \overline{\boldsymbol{M}}_{\overline{V}_t, t-1}^{-1} \sum_{\substack{s \in [t-1] \\ i_s \in \overline{V}_t}} \boldsymbol{x}_{a_s} \boldsymbol{\epsilon}_{a_s}^{i_s, s} + \overline{\boldsymbol{M}}_{\overline{V}_t, t-1}^{-1} \sum_{\substack{s \in [t-1] \\ i_s \in \overline{V}_t}} \boldsymbol{x}_{a_s} \eta_s + \overline{\boldsymbol{M}}_{\overline{V}_t, t-1}^{-1} \sum_{\substack{s \in [t-1] \\ i_s \in \overline{V}_t}} \boldsymbol{x}_{a_s} \boldsymbol{x}_{a_s}^\top (\boldsymbol{\theta}_{i_s} - \boldsymbol{\theta}_{i_t})$$

$$+ \overline{\boldsymbol{M}}_{\overline{V}_t, t-1}^{-1} \Big( \sum_{\substack{s \in [t-1] \\ i_s \in \overline{V}_t}} \boldsymbol{x}_{a_s} \boldsymbol{x}_{a_s}^\top + \lambda \boldsymbol{I} \Big) \boldsymbol{\theta}_{i_t} - \lambda \overline{\boldsymbol{M}}_{\overline{V}_t, t-1}^{-1} \boldsymbol{\theta}_{i_t} - \boldsymbol{\theta}_{i_t}$$

$$= \overline{\boldsymbol{M}}_{\overline{V}_t, t-1}^{-1} \sum_{\substack{s \in [t-1] \\ i_s \in \overline{V}_t}} \boldsymbol{x}_{a_s} \boldsymbol{\epsilon}_{a_s}^{i_s, s} + \overline{\boldsymbol{M}}_{\overline{V}_t, t-1}^{-1} \sum_{\substack{s \in [t-1] \\ i_s \in \overline{V}_t}} \boldsymbol{x}_{a_s} \eta_s + \overline{\boldsymbol{M}}_{\overline{V}_t, t-1}^{-1} \sum_{\substack{s \in [t-1] \\ i_s \in \overline{V}_t}} \boldsymbol{x}_{a_s} \boldsymbol{x}_{a_s}^\top (\boldsymbol{\theta}_{i_s} - \boldsymbol{\theta}_{i_t})$$

$$- \lambda \overline{\boldsymbol{M}}_{\overline{V}_t, t-1}^{-1} \boldsymbol{\theta}_{i_t} .$$

Thus, with Lemma 5.7 and with the previous reasoning, with probability at least $1 - 5\delta$, we have:

$$\Big| \boldsymbol{x}_a^\top (\hat{\boldsymbol{\theta}}_{\overline{V}_t, t-1} - \boldsymbol{\theta}_{i_t}) \Big| \le \lambda \Big| \boldsymbol{x}_a^\top \overline{\boldsymbol{M}}_{\overline{V}_t, t-1}^{-1} \boldsymbol{\theta}_{i_t} \Big| + \Big| \sum_{\substack{s \in [t-1] \\ i_s \in \overline{V}_t}} \boldsymbol{x}_a^\top \overline{\boldsymbol{M}}_{\overline{V}_t, t-1}^{-1} \boldsymbol{x}_{a_s} \boldsymbol{\epsilon}_{a_s}^{i_s, s} \Big| + \Big| \boldsymbol{x}_a^\top \overline{\boldsymbol{M}}_{\overline{V}_t, t-1}^{-1} \sum_{\substack{s \in [t-1] \\ i_s \in \overline{V}_t}} \boldsymbol{x}_{a_s} \eta_s \Big|$$

$$+ \Big| \boldsymbol{x}_a^\top \overline{\boldsymbol{M}}_{\overline{V}_t, t-1}^{-1} \sum_{\substack{s \in [t-1] \\ i_s \in \overline{V}_t}} \boldsymbol{x}_{a_s} \boldsymbol{x}_{a_s}^\top (\boldsymbol{\theta}_{i_s} - \boldsymbol{\theta}_{i_t}) \Big|$$

$$\le \epsilon_* \sum_{\substack{s \in [t-1] \\ i_s \in \overline{V}_t}} \Big| \boldsymbol{x}_a^\top \overline{\boldsymbol{M}}_{\overline{V}_t, t-1}^{-1} \boldsymbol{x}_{a_s} \Big| + \|\boldsymbol{x}_a\|_{\overline{\boldsymbol{M}}_{\overline{V}_t, t-1}^{-1}} \Big( \sqrt{\lambda} + \sqrt{2 \log(\frac{1}{\delta}) + d \log(1 + \frac{T}{\lambda d})} \Big)$$

$$+ \frac{\epsilon_* \sqrt{2d}}{\tilde{\lambda}_x^{\frac{3}{2}}} .$$

Therefore, combining situation 1 and situation 2, the result of Lemma 5.6 then follows.

## J Technical Lemmas and Their Proofs

We first prove the following technical lemma which is used to prove Lemma H.1.

**Lemma J.1.** *Under Assumption 3.3, at any time $t$, for any fixed unit vector $\boldsymbol{\theta} \in \mathbb{R}^d$*

$$\mathbb{E}_t[(\boldsymbol{\theta}^\top \boldsymbol{x}_{a_t})^2 | |\mathcal{A}_t|] \ge \tilde{\lambda}_x \triangleq \int_0^{\lambda_x} (1 - e^{-\frac{(\lambda_x - x)^2}{2\sigma^2}})^C dx . \tag{66}$$

*Proof.* The proof of this lemma mainly follows the proof of Claim 1 in [12], but with more careful analysis, since their assumption is more stringent than ours.

Denote the feasible arms at round $t$ by $\mathcal{A}_t = \{\boldsymbol{x}_{t,1}, \boldsymbol{x}_{t,2}, \ldots, \boldsymbol{x}_{t,|\mathcal{A}_t|}\}$. Consider the corresponding i.i.d. random variables $\theta_i = (\boldsymbol{\theta}^\top \boldsymbol{x}_{t,i})^2 - \mathbb{E}_t[(\boldsymbol{\theta}^\top \boldsymbol{x}_{t,i})^2 | |\mathcal{A}_t|], i = 1, 2, \ldots, |\mathcal{A}_t|$. By Assumption 3.3, $\theta_i$ s are sub-Gaussian random variables with variance bounded by $\sigma^2$. Therefore, we have that

for any $\alpha > 0$ and any $i \in [|\mathcal{A}_t|]$:

$$\mathbb{P}_t(\theta_i < -\alpha | \, |\mathcal{A}_t|) \leq e^{-\frac{\alpha^2}{2\sigma^2}},$$

where $\mathbb{P}_t(\cdot)$ is the shorthand for the conditional probability $\mathbb{P}(\cdot | (i_1, \mathcal{A}_1, r_1), \ldots, (i_{t-1}, \mathcal{A}_{t-1}, r_{t-1}), i_t)$.

We also have that $\mathbb{E}_t[(\boldsymbol{\theta}^\top \boldsymbol{x}_{t,i})^2 | \, |\mathcal{A}_t|] = \mathbb{E}_t[\boldsymbol{\theta}^\top \boldsymbol{x}_{t,i} \boldsymbol{x}_{t,i}^\top \boldsymbol{\theta} | \, |\mathcal{A}_t|] \geq \lambda_{\min}(\mathbb{E}_{\boldsymbol{x} \sim \rho}[\boldsymbol{x}\boldsymbol{x}^\top]) \geq \lambda_x$ by Assumption 3.3. With the above inequalities, we can get

$$\mathbb{P}_t(\min_{i=1,\ldots,|\mathcal{A}_t|} (\boldsymbol{\theta}^\top \boldsymbol{x}_{t,i})^2 \geq \lambda_x - \alpha | \, |\mathcal{A}_t|) \geq (1 - e^{-\frac{\alpha^2}{2\sigma^2}})^C,$$

where $C$ is the upper bound of $|\mathcal{A}_t|$.

Therefore, we have

$$\mathbb{E}_t[(\boldsymbol{\theta}^\top \boldsymbol{x}_{a_t})^2 | \, |\mathcal{A}_t|] \geq \mathbb{E}_t[\min_{i=1,\ldots,|\mathcal{A}_t|} (\boldsymbol{\theta}^\top \boldsymbol{x}_{t,i})^2 | \, |\mathcal{A}_t|]$$

$$\geq \int_0^\infty \mathbb{P}_t(\min_{i=1,\ldots,|\mathcal{A}_t|} (\boldsymbol{\theta}^\top \boldsymbol{x}_{t,i})^2 \geq x | \, |\mathcal{A}_t|) dx$$

$$\geq \int_0^{\lambda_x} (1 - e^{-\frac{(\lambda_x - x)^2}{2\sigma^2}})^C dx \triangleq \tilde{\lambda}_x$$

$\square$

Finally, we prove the following lemma which is used in the proof of Theorem 5.3.

**Lemma J.2.**

$$\sum_{t=T_0+1}^T \min\{\mathbb{I}\{i_t \in V_j\} \|\boldsymbol{x}_{a_t}\|^2_{\overline{\boldsymbol{M}}^{-1}_{V_j,t-1}}, 1\} \leq 2d \log(1 + \frac{T}{\lambda d}), \forall j \in [m]. \quad (67)$$

*Proof.*

$$\det(\overline{\boldsymbol{M}}_{V_j,T}) = \det\left(\overline{\boldsymbol{M}}_{V_j,T-1} + \mathbb{I}\{i_T \in V_j\}\boldsymbol{x}_{a_T}\boldsymbol{x}_{a_T}^\top\right)$$

$$= \det(\overline{\boldsymbol{M}}_{V_j,T-1})\det\left(\boldsymbol{I} + \mathbb{I}\{i_T \in V_j\}\overline{\boldsymbol{M}}^{-\frac{1}{2}}_{V_j,T-1}\boldsymbol{x}_{a_T}\boldsymbol{x}_{a_T}^\top\overline{\boldsymbol{M}}^{-\frac{1}{2}}_{V_j,T-1}\right)$$

$$= \det(\overline{\boldsymbol{M}}_{V_j,T-1})\left(1 + \mathbb{I}\{i_T \in V_j\} \|\boldsymbol{x}_{a_T}\|^2_{\overline{\boldsymbol{M}}^{-1}_{V_j,T-1}}\right)$$

$$= \det(\overline{\boldsymbol{M}}_{V_j,T_0}) \prod_{t=T_0+1}^T \left(1 + \mathbb{I}\{i_t \in V_j\} \|\boldsymbol{x}_{a_t}\|^2_{\overline{\boldsymbol{M}}^{-1}_{V_j,t-1}}\right)$$

$$\geq \det(\lambda\boldsymbol{I}) \prod_{t=T_0+1}^T \left(1 + \mathbb{I}\{i_t \in V_j\} \|\boldsymbol{x}_{a_t}\|^2_{\overline{\boldsymbol{M}}^{-1}_{V_j,t-1}}\right). \quad (68)$$

$\forall x \in [0,1]$, we have $x \leq 2\log(1+x)$. Therefore

$$\sum_{t=T_0+1}^T \min\{\mathbb{I}\{i_t \in V_j\} \|\boldsymbol{x}_{a_t}\|^2_{\overline{\boldsymbol{M}}^{-1}_{V_j,t-1}}, 1\} \leq 2 \sum_{t=T_0+1}^T \log\left(1 + \mathbb{I}\{i_t \in V_j\} \|\boldsymbol{x}_{a_t}\|^2_{\overline{\boldsymbol{M}}^{-1}_{V_j,t-1}}\right)$$

$$= 2\log\left(\prod_{t=T_0+1}^T \left(1 + \mathbb{I}\{i_t \in V_j\} \|\boldsymbol{x}_{a_t}\|^2_{\overline{\boldsymbol{M}}^{-1}_{V_j,t-1}}\right)\right)$$

$$\leq 2[\log(\det(\overline{\boldsymbol{M}}_{V_j,T})) - \log(\det(\lambda\boldsymbol{I}))]$$

$$\leq 2\log\left(\frac{\text{trace}(\lambda\boldsymbol{I} + \sum_{t=1}^T \mathbb{I}\{i_t \in V_j\}\boldsymbol{x}_{a_t}\boldsymbol{x}_{a_t}^\top)}{\lambda d}\right)^d$$

$$\leq 2d \log(1 + \frac{T}{\lambda d}). \quad (69)$$

$\square$

## K   Algorithms of RSCLUMB

This section introduces the Robust Set-based Clustering of Misspecified Bandits Algorithm (RSCLUMB). Unlike RCLUMB, which maintains a graph-based clustering structure, RSCLUMB maintains a set-based clustering structure. Besides, RCLUMB only splits clusters during the learning process, while RSCLUMB allows both split and merge operations. A brief illustration is that the agent will split a user out of its current set(cluster) if it finds an inconsistency between the user and its set, and if there are two clusters whose estimated preferences are close enough, the agent will merge them. A detailed discussion of the connection between the graph structure and the set structure can be found in [27].

Now we introduce the details of RSCLUMB. The algorithm first initializes a single set $S_1$ containing all users and updates it during the learning process. The whole learning process consists of phases (Algo. 2 Line 3), where the $s-th$ phase contains $2^s$ rounds. At the beginning of each phase, the agent marks all users as "unchecked", and if a user comes later, it will be marked as "checked". If all users in a cluster are checked, then this cluster will be marked as "checked" meaning it is an accurate cluster in the current phase. With this mechanism, every phase can maintain an accuracy level, and the agent can put the accurate clusters aside and focus on exploring the inaccurate ones. For each cluster $V_j$, the algorithm maintains two estimated vectors $\hat{\boldsymbol{\theta}}_{V_j}$ and $\tilde{\boldsymbol{\theta}}_{V_j}$, where the $\hat{\boldsymbol{\theta}}_{V_j}$ is similar to the $\hat{\boldsymbol{\theta}}_{\overline{V}_j}$ in RCLUMB and is used for the recommendation, while the $\tilde{\boldsymbol{\theta}}_{V_j}$ is the average of all the estimated user preference vectors in this cluster and is used for the split and merge operations.

At time $t$ in phase $s$, the user $i_\tau$ comes with the item set $\mathcal{D}_\tau$, where $\tau$ represents the index of total time steps. Then the algorithm determines the cluster and makes a cluster-based recommendation. This process is similar to RCLUMB. After updating the information (Algo. 2 Line12), the agent checks if a split or a merge is possible (Algo. 2 Line13-17).

By our assumption, users in the same cluster have the same vectors. So a cluster can be regarded as a good cluster only when all the estimated user vectors are close to the estimated cluster vector. We call a user is consistent with the cluster if their estimated vectors are close enough. If a user is inconsistent with its current cluster, the agent will split it out. Two clusters are consistent when their estimated vectors are close, and the agent will merge them.

RSCLUMB maintains two sets of estimated cluster vectors: (i) cluster-level estimation with integrated user information, which is for recommendations (Line 12 and Line 10 in Algo.2); (ii) the average of estimated user vectors, which is used for robust clustering (Line 3 in Algo.3 and Line 2 in Algo.4). The previous set-based CB work [27] only uses (i) for both recommendations and clustering, which would lead to erroneous clustering under misspecifications, and cannot get any non-vacuous regret bound in CBMUM.

## L   Main Theorem and Lemmas of RSCLUMB

**Theorem L.1** (main result on regret bound for RSCLUMB). *With the same assumptions in Theorem 5.3, the expected regret of the RSCLUMB algorithm for T rounds satisfies:*

$$R(T) \le O\left(u\left(\frac{d}{\tilde{\lambda}_x(\gamma_1-\zeta_1)^2}+\frac{1}{\tilde{\lambda}_x^2}\right)\log T + \frac{\epsilon_*\sqrt{dT}}{\tilde{\lambda}_x^{1.5}}+\epsilon_*T\sqrt{md\log T}+d\sqrt{mT}\log T+\epsilon_*\sqrt{\frac{1}{\tilde{\lambda}_x}T}\right)$$

$$\le O(\epsilon_*T\sqrt{md\log T}+d\sqrt{mT}\log T) \tag{70}$$

**Lemma L.2.** *For RSCLUMB, we use $T_1$ to represent the corresponding $T_0$ of RCLUMB. Then :*

$$T_1 \triangleq 16u\log(\frac{u}{\delta})+4u\max\{\frac{16}{\tilde{\lambda}_x^2}\log(\frac{8d}{\tilde{\lambda}_x^2\delta}),\frac{8d}{\tilde{\lambda}_x(\frac{\gamma_1}{6}-\epsilon_*\sqrt{\frac{1}{2\tilde{\lambda}_x}})^2}\log(\frac{u}{\delta})\}$$

$$= O\left(u\left(\frac{d}{\tilde{\lambda}_x(\gamma_1-\zeta_1)^2}+\frac{1}{\tilde{\lambda}_x^2}\right)\log\frac{1}{\delta}\right)$$

---

**Algorithm 2** Robust Set-based Clustering of Misspecified Bandits Algorithm (RSCLUMB)

---

1: **Input:** Deletion parameter $\alpha_1, \alpha_2 > 0$, $f(T) = \sqrt{\frac{1+\ln(1+T)}{1+T}}$, $\lambda, \beta, \epsilon_* > 0$.
2: **Initialization:**
   - $\boldsymbol{M}_{i,0} = 0_{d \times d}, \boldsymbol{b}_{i,0} = 0_{d \times 1}, T_{i,0} = 0$, $\forall i \in \mathcal{U}$;
   - Initialize the set of cluster indexes by $J = \{1\}$ and the single cluster $\boldsymbol{S}_1$ by $\boldsymbol{M}_1 = 0_{d \times d}$, $\boldsymbol{b}_1 = 0_{d \times 1}, T_1 = 0, C_1 = \mathcal{U}, j(i) = 1, \forall i$.
3: **for all** $s = 1, 2, \ldots$ **do**
4:    Mark every user unchecked for each cluster.
5:    For each cluster $V_j$, compute $\tilde{T}_{V_j} = T_{V_j}, \hat{\boldsymbol{\theta}}_{V_j} = (\lambda \boldsymbol{I} + \boldsymbol{M}_{V_j})^{-1} \boldsymbol{b}_{V_j}, \tilde{\boldsymbol{\theta}}_{V_j} = \frac{\sum_{i \in V_j} \hat{\boldsymbol{\theta}}_i}{[V_j]}$
6:    **for all** $t = 1, 2, \ldots, T$ **do**
7:       Compute $\tau = 2^s - 2 + t$
8:       Receive the user $i_\tau$ and the decision set $\mathcal{D}_\tau$
9:       Determine the cluster index $j = j(i_\tau)$
10:      Recommend item $a_\tau$ with the largest UCB index as shown in Eq. (5)
11:      Received the feedback $r_\tau$.
12:      Update the information:

$$\boldsymbol{M}_{i_\tau,\tau} = \boldsymbol{M}_{i_\tau,\tau-1} + \boldsymbol{x}_{a_\tau} \boldsymbol{x}_{a_\tau}^{\mathrm{T}}, \boldsymbol{b}_{i_\tau,\tau} = \boldsymbol{b}_{i_\tau,\tau-1} + r_\tau \boldsymbol{x}_{a_\tau},$$

$$T_{i_\tau,\tau} = T_{i_\tau,\tau-1} + 1, \hat{\boldsymbol{\theta}}_{i_\tau,\tau} = (\lambda \boldsymbol{I} + \boldsymbol{M}_{i_\tau,\tau})^{-1} \boldsymbol{b}_{i_\tau,\tau}$$

$$\boldsymbol{M}_{V_j,\tau} = \boldsymbol{M}_{V_j,\tau-1} + \boldsymbol{x}_{a_\tau} \boldsymbol{x}_{a_\tau}^{\mathrm{T}}, \boldsymbol{b}_{V_j,\tau} = \boldsymbol{b}_{V_j,\tau-1} + r_\tau \boldsymbol{x}_\tau,$$

$$T_{V_j,\tau} = T_{V_j,\tau-1} + 1, \hat{\boldsymbol{\theta}}_{V_j,\tau} = (\lambda \boldsymbol{I} + \boldsymbol{M}_{V_j,\tau})^{-1} \boldsymbol{b}_{V_j,\tau},$$

$$\tilde{\boldsymbol{\theta}}_{V_j,\tau} = \frac{\sum_{i \in V_j} \hat{\boldsymbol{\theta}}_i, \tau}{[V_j]}$$

13:      **if** $i_\tau$ is unchecked **then**
14:         Run **Split**
15:         Mark user $i_\tau$ has been checked
16:         Run **Merge**

---

**Algorithm 3** Split

---

1: Define $F(T) = \sqrt{\frac{1+\ln(1+T)}{1+T}}$
2: **if** $\left\| \hat{\boldsymbol{\theta}}_{i_\tau,\tau} - \tilde{\boldsymbol{\theta}}_{V_j,\tau} \right\| > \alpha_1(F(T_{i_\tau,\tau}) + F(T_{V_j,\tau})) + \alpha_2 \epsilon_*$ **then**
3:    Split user $i_\tau$ from cluster $V_j$ and form a new cluster $V_j'$ of user $i_\tau$

$$\boldsymbol{M}_{V_j,\tau} = \boldsymbol{M}_{V_j,\tau} - \boldsymbol{M}_{i_\tau,\tau}, \boldsymbol{b}_{V_j} = \boldsymbol{b}_{V_j} - \boldsymbol{b}_{i_\tau,\tau},$$

$$T_{V_j,\tau} = T_{V_j,\tau} - T_{i_\tau,\tau}, C_{j,\tau} = C_{j,\tau} - \{i_\tau\},$$

$$\boldsymbol{M}_{V_j',\tau} = \boldsymbol{M}_{i_\tau,\tau}, \boldsymbol{b}_{V_j',\tau} = \boldsymbol{b}_{i_\tau,\tau},$$

$$T_{V_j',\tau} = T_{i_\tau,\tau}, C_{j',\tau} = \{i_\tau\}$$

---

**Lemma L.3.** *For RSCLUMB, after $2T_1 + 1$ rounds: in each phase, after the first $u$ rounds, with probability at least $1 - 5\delta$:*

$$\left| \boldsymbol{x}_a^\top (\boldsymbol{\theta}_{i_t} - \hat{\boldsymbol{\theta}}_{\overline{V}_t,t-1}) \right| \leq (\frac{3\epsilon_* \sqrt{2d}}{2\tilde{\lambda}_x^{\frac{3}{2}}} + 6\epsilon_* \sqrt{\frac{1}{2\tilde{\lambda}_x}}) \mathbb{I}\{\overline{V}_t \notin V\} + \beta \|\boldsymbol{x}_a\|_{\overline{\boldsymbol{M}}_{\overline{V}_t,t-1}^{-1}} + \epsilon_* \sum_{\substack{s \in [t-1] \\ i_s \in \overline{V}_t}} \left| \boldsymbol{x}_a^\top \overline{\boldsymbol{M}}_{\overline{V}_t,t-1}^{-1} \boldsymbol{x}_{a_s} \right|$$

$$\triangleq (\frac{3\epsilon_* \sqrt{2d}}{2\tilde{\lambda}_x^{\frac{3}{2}}} + 6\epsilon_* \sqrt{\frac{1}{2\tilde{\lambda}_x}}) \mathbb{I}\{\overline{V}_t \notin V\} + C_{a,t}$$

**Algorithm 4** Merge

1: **for** any two checked clusters $V_{j_1}, V_{j_2}$ satisfying

$$\left\|\tilde{\boldsymbol{\theta}}_{j_1} - \tilde{\boldsymbol{\theta}}_{j_2}\right\| < \frac{\alpha_1}{2}(F(T_{V_{j_1}}) + F(T_{V_{j_2}})) + \frac{\alpha_2}{2}\epsilon_*$$

**do**

2:    Merge them:

$$\boldsymbol{M}_{V_{j_1}} = \boldsymbol{M}_{j_1} + \boldsymbol{M}_{j_2}, \boldsymbol{b}_{V_{j_1}} = \boldsymbol{b}_{V_{j_1}} + \boldsymbol{b}_{V_{j_2}},$$
$$T_{V_{j_1}} = T_{V_{j_1}} + T_{V_{j_2}}, C_{V_{j_1}} = C_{V_{j_1}} \cup C_{V_{j_2}}$$

3:    Set $j(i) = j_1, \forall i \in j_2$, delete $V_{j_2}$

---

# M  Proof of Lemma L.3

$$
\begin{aligned}
|\boldsymbol{x}_a^{\mathrm{T}}(\boldsymbol{\theta}_i - \hat{\boldsymbol{\theta}}_{\overline{V}_t,t})| &= |\boldsymbol{x}_a^{\mathrm{T}}(\boldsymbol{\theta}_i - \boldsymbol{\theta}_{V_t})| + |\boldsymbol{x}_a^{\mathrm{T}}(\hat{\boldsymbol{\theta}}_{\overline{V}_t,t} - \boldsymbol{\theta}_{V_t})| \\
&\leq \|\boldsymbol{x}_a^{\mathrm{T}}\| \|\boldsymbol{\theta}_i - \boldsymbol{\theta}_{V_t}\| + |\boldsymbol{x}_a^{\mathrm{T}}(\hat{\boldsymbol{\theta}}_{\overline{V}_t,t} - \boldsymbol{\theta}_{V_t})| \\
&\leq 6\epsilon_*\sqrt{\frac{1}{2\tilde{\lambda}_x}} + |\boldsymbol{x}_a^{\mathrm{T}}(\hat{\boldsymbol{\theta}}_{\overline{V}_t,t} - \boldsymbol{\theta}_{V_t})|
\end{aligned}
\tag{71}
$$

where the last inequality holds due to the fact $\|\boldsymbol{x}_a\| \leq 1$ and the condition of "split" and "merge". For $|\boldsymbol{x}_a^{\mathrm{T}}(\hat{\boldsymbol{\theta}}_{\overline{V}_t,t} - \boldsymbol{\theta}_{V_t})|$:

$$
\begin{aligned}
\hat{\boldsymbol{\theta}}_{\overline{V}_t,t-1} - \boldsymbol{\theta}_{V_t} &= (\sum_{\substack{s\in[t-1]\\i_s\in\overline{V}_t}} \boldsymbol{x}_{a_s}\boldsymbol{x}_{a_s}^{\top} + \lambda\boldsymbol{I})^{-1}(\sum_{\substack{s\in[t-1]\\i_s\in\overline{V}_t}} \boldsymbol{x}_{a_s}r_s) - \boldsymbol{\theta}_{V_t} \\
&= (\sum_{\substack{s\in[t-1]\\i_s\in\overline{V}_t}} \boldsymbol{x}_{a_s}\boldsymbol{x}_{a_s}^{\top} + \lambda\boldsymbol{I})^{-1}\left(\sum_{\substack{s\in[t-1]\\i_s\in\overline{V}_t}} \boldsymbol{x}_{a_s}(\boldsymbol{x}_{a_s}^{\top}\boldsymbol{\theta}_{i_s} + \boldsymbol{\epsilon}_{a_s}^{i_s,s} + \eta_s)\right) - \boldsymbol{\theta}_{V_t} \\
&= \overline{\boldsymbol{M}}_{\overline{V}_t,t-1}^{-1}\sum_{\substack{s\in[t-1]\\i_s\in\overline{V}_t}}\boldsymbol{x}_{a_s}\boldsymbol{\epsilon}_{a_s}^{i_s,s} + \overline{\boldsymbol{M}}_{\overline{V}_t,t-1}^{-1}\sum_{\substack{s\in[t-1]\\i_s\in\overline{V}_t}}\boldsymbol{x}_{a_s}\eta_s + \overline{\boldsymbol{M}}_{\overline{V}_t,t-1}^{-1}\sum_{\substack{s\in[t-1]\\i_s\in\overline{V}_t}}\boldsymbol{x}_{a_s}\boldsymbol{x}_{a_s}^{\top}\boldsymbol{\theta}_{i_s} - \boldsymbol{\theta}_{V_t} \\
&= \overline{\boldsymbol{M}}_{\overline{V}_t,t-1}^{-1}\sum_{\substack{s\in[t-1]\\i_s\in\overline{V}_t}}\boldsymbol{x}_{a_s}\boldsymbol{\epsilon}_{a_s}^{i_s,s} + \overline{\boldsymbol{M}}_{\overline{V}_t,t-1}^{-1}\sum_{\substack{s\in[t-1]\\i_s\in\overline{V}_t}}\boldsymbol{x}_{a_s}\eta_s + \overline{\boldsymbol{M}}_{\overline{V}_t,t-1}^{-1}\sum_{\substack{s\in[t-1]\\i_s\in\overline{V}_t}}\boldsymbol{x}_{a_s}\boldsymbol{x}_{a_s}^{\top}(\boldsymbol{\theta}_{i_s} - \boldsymbol{\theta}_{V_t}) \\
&\quad + \overline{\boldsymbol{M}}_{\overline{V}_t,t-1}^{-1}(\sum_{\substack{s\in[t-1]\\i_s\in\overline{V}_t}}\boldsymbol{x}_{a_s}\boldsymbol{x}_{a_s}^{\top} + \lambda\boldsymbol{I})\boldsymbol{\theta}_{V_t} - \lambda\overline{\boldsymbol{M}}_{\overline{V}_t,t-1}^{-1}\boldsymbol{\theta}_{V_t} - \boldsymbol{\theta}_{V_t} \\
&= \overline{\boldsymbol{M}}_{\overline{V}_t,t-1}^{-1}\sum_{\substack{s\in[t-1]\\i_s\in\overline{V}_t}}\boldsymbol{x}_{a_s}\boldsymbol{\epsilon}_{a_s}^{i_s,s} + \overline{\boldsymbol{M}}_{\overline{V}_t,t-1}^{-1}\sum_{\substack{s\in[t-1]\\i_s\in\overline{V}_t}}\boldsymbol{x}_{a_s}\eta_s + \overline{\boldsymbol{M}}_{\overline{V}_t,t-1}^{-1}\sum_{\substack{s\in[t-1]\\i_s\in\overline{V}_t}}\boldsymbol{x}_{a_s}\boldsymbol{x}_{a_s}^{\top}(\boldsymbol{\theta}_{i_s} - \boldsymbol{\theta}_{V_t}) \\
&\quad - \lambda\overline{\boldsymbol{M}}_{\overline{V}_t,t-1}^{-1}\boldsymbol{\theta}_{V_t}.
\end{aligned}
$$

Thus, with the same method in Lemma 5.7 but replace $\zeta = 4\epsilon_*\sqrt{\frac{1}{2\tilde{\lambda}_x}}$ with $\zeta_1 = 6\epsilon_*\sqrt{\frac{1}{2\tilde{\lambda}_x}}$, and with the previous reasoning, with probability at least $1 - 5\delta$, we have:

$$|\boldsymbol{x}_a^{\mathrm{T}}(\hat{\boldsymbol{\theta}}_{\overline{V}_t,t} - \boldsymbol{\theta}_{V_t})| \leq C_{a_t} + \frac{3\epsilon_*\sqrt{2d}}{2\tilde{\lambda}_x^{\frac{3}{2}}} \tag{72}$$

The lemma can be concluded.

## N    Proof of Lemma L.2

With the analysis in the proof of Lemma H.1, with probability at least $1 - \delta$:

$$\left\|\hat{\boldsymbol{\theta}}_{i,t} - \boldsymbol{\theta}^{j(i)}\right\|_2 \leq \frac{\beta(T_{i,t}, \frac{\delta}{u}) + \epsilon_* \sqrt{T_{i,t}}}{\sqrt{\lambda + \lambda_{\min}(\boldsymbol{M}_{i,t})}}, \forall i \in \mathcal{U}, \tag{73}$$

and the estimated error of the current cluster $\left\|\tilde{\boldsymbol{\theta}}^{j(i)} - \boldsymbol{\theta}^{j(i)}\right\|$ also satisfies this inequality. For set-based clustering structure, to ensure for each user there is only one $\zeta$-close cluster, we let:

$$\frac{\beta(T_{i,t}, \frac{\delta}{u}) + \epsilon_* \sqrt{T_{i,t}}}{\sqrt{\lambda + \lambda_{\min}(\boldsymbol{M}_{i,t})}} \leq \frac{\gamma_1}{6} \tag{74}$$

By assuming $\lambda < 2\log(\frac{u}{\delta}) + d\log(1 + \frac{T_{i,t}}{\lambda d})$, we can simplify it to

$$\frac{2\log(\frac{u}{\delta}) + d\log(1 + \frac{T_{i,t}}{\lambda d})}{2\tilde{\lambda}_x T_{i,t}} < \frac{1}{4}(\frac{\gamma_1}{6} - \epsilon_*\sqrt{\frac{1}{2\tilde{\lambda}_x}})^2 \tag{75}$$

which can be proved by $\frac{2\log(\frac{u}{\delta})}{2\tilde{\lambda}_x T_{i,t}} \leq \frac{1}{8}(\frac{\gamma_1}{6} - \epsilon_*\sqrt{\frac{1}{2\tilde{\lambda}_x}})^2$ and $\frac{d\log(1+\frac{T_{i,t}}{\lambda d})}{2\tilde{\lambda}_x T_{i,t}} \leq \frac{1}{8}(\frac{\gamma_1}{6} - \epsilon_*\sqrt{\frac{1}{2\tilde{\lambda}_x}})^2$. It's obvious that the former one can be satisfied by $T_{i,t} \geq \frac{8\log(u/\delta)}{\tilde{\lambda}_x(\frac{\gamma_1}{6} - \epsilon_*\sqrt{1/2\tilde{\lambda}_x})^2}$. As for the latter one, by [25] Lemma 9, we can get $T_{i,t} \geq \frac{8d\log(\frac{16}{\tilde{\lambda}_x\lambda(\frac{\gamma_1}{6} - \epsilon_*\sqrt{1/2\tilde{\lambda}_x})^2})}{4\tilde{\lambda}_x(\frac{\gamma_1}{6} - \epsilon_*\sqrt{1/2\tilde{\lambda}_x})^2}$. By assuming $\frac{u}{\delta} \geq \frac{16}{4\tilde{\lambda}_x\lambda(\frac{\gamma_1}{6} - \epsilon_*\sqrt{2/4\tilde{\lambda}_x})^2}$, the lemma is proved.

## O    Proof of Theorem L.1

After $2T_1$ rounds, in each phase, at most $u$ times split operations will happen, we use $u\log(T)$ to bound the regret generated in these rounds. Then in the remained rounds the cluster num will be no more than $m$.

For the instantaneous regret $R_t$ at round $t$, with probability at least $1 - 2\delta$ for some $\delta \in (0, \frac{1}{2})$:

$$\begin{aligned}
R_t &= (\boldsymbol{x}_{a_t^*}^{\mathrm{T}}\boldsymbol{\theta}_{i_t} + \boldsymbol{\epsilon}_{a_t^*}^{i_t,t}) - (\boldsymbol{x}_{a_t}^{\mathrm{T}}\boldsymbol{\theta}_{i_t} + \boldsymbol{\epsilon}_{a_t}^{i_t,t}) \\
&= \boldsymbol{x}_{a_t^*}^{\top}(\boldsymbol{\theta}_{i_t} - \hat{\boldsymbol{\theta}}_{\overline{V}_t, t-1}) + (\boldsymbol{x}_{a_t^*}^{\top}\hat{\boldsymbol{\theta}}_{\overline{V}_t, t-1} + C_{a_t^*, t}) - (\boldsymbol{x}_{a_t}^{\top}\hat{\boldsymbol{\theta}}_{\overline{V}_t, t-1} + C_{a_t, t}) \\
&\quad + \boldsymbol{x}_{a_t}^{\top}(\hat{\boldsymbol{\theta}}_{\overline{V}_t, t-1} - \boldsymbol{\theta}_{i_t}) + C_{a_t, t} - C_{a_t^*, t} + (\boldsymbol{\epsilon}_{a_t^*}^{i_t,t} - \boldsymbol{\epsilon}_{a_t}^{i_t,t}) \\
&\leq 2C_{a_t} + 2\epsilon_* + (12\epsilon_*\sqrt{\frac{1}{2\tilde{\lambda}_x}} + \frac{3\epsilon_*\sqrt{2d}}{\tilde{\lambda}_x^{\frac{3}{2}}})\mathbb{I}(\overline{V}_t \notin V)
\end{aligned} \tag{76}$$

where the last inequality holds due to the UCB arm selection strategy, the concentration bound given in LemmaL.3 and the fact that $\left\|\epsilon^{i,t}\right\|_\infty \leq \epsilon_*$.

Define such events. Let:

$$\mathcal{E}_2 = \{\text{All clusters } \overline{V}_t \text{ only contain users who satisfy } \left\|\tilde{\boldsymbol{\theta}}_i - \tilde{\boldsymbol{\theta}}_{\overline{V}_t}\right\| \leq \alpha_1(\sqrt{\frac{1 + \log(1 + T_{i,t})}{1 + T_{i,t}}} + \sqrt{\frac{1 + \log(1 + T_{\overline{V}_t, t})}{1 + T_{\overline{V}_t, t}}}) + \alpha_2\epsilon_*\}$$

$$\mathcal{E}_3 = \{r_t \leq 2C_{a_t} + 2\epsilon_* + 12\epsilon_*\sqrt{\frac{1}{2\tilde{\lambda}_x}} + \frac{3\epsilon_*\sqrt{2d}}{\tilde{\lambda}_x^{\frac{3}{2}}}\}$$

$$\mathcal{E}' = \mathcal{E}_2 \cap \mathcal{E}_3$$

From previous analysis, we can know that $\mathbb{P}(\mathcal{E}_2) \geq 1 - 3\delta$ and $\mathbb{P}(\mathcal{E}_3) \geq 1 - 2\delta$, thus $\mathbb{P}(\mathcal{E}' \geq 1 - 5\delta)$. By taking $\delta = \frac{1}{T}$, we can get:

$$\begin{aligned}
E(R_t) &= P(\mathcal{E})\mathbb{I}\{\mathcal{E}\}R_t + P(\bar{\mathcal{E}})\mathbb{I}\{\bar{\mathcal{E}}\}R_t \\
&\leq \mathbb{I}\{\mathcal{E}\}R_t + 5 \\
&\leq 2T_1 + 2\epsilon_* T + (12\epsilon_*\sqrt{\frac{1}{2\tilde{\lambda}_x}} + \frac{3\epsilon_*\sqrt{2d}}{\tilde{\lambda}_x^{\frac{3}{2}}})T + 2\sum_{2T_1}^{T} C_{a_t} + 5
\end{aligned} \tag{77}$$

Now we need to bound $2\sum_{2T_1}^{T} C_{a_t}$. We already know that after $2T_1$ rounds, in each phase $k$ after the first $u$ rounds, there will be at most $m$ clusters

Consider phase $k$, for simplicity, ignore the fist $u$ rounds. For the first term in $C_{a_t}$:

$$
\begin{aligned}
\sum_{t=T_{k-1}}^{T_k} \|\boldsymbol{x}_{a_t}\|^{-1}_{\boldsymbol{M}_{\overline{V}_{t,t-1}}} &= \sum_{t=T_{k-1}}^{T_k} \sum_{j=1}^{m_t} \mathbb{I}\{i \in \overline{V}_{t,j}\} \|\boldsymbol{x}_{a_t}\|_{\boldsymbol{M}_{\overline{V}_{t,j}}^{-1}} \\
&\le \sum_{j=1}^{m_t} \sqrt{\sum_{t=T_{k-1}}^{T_k} \mathbb{I}\{i \in V_{t,j}\} \sum_{t=T_{k-1}}^{T_k} \mathbb{I}\{i \in V_{t,j}\} \|\boldsymbol{x}_{a_t}\|^2_{\boldsymbol{M}_{\overline{V}_{t,j}}^{-1}}} \\
&\le \sum_{j=1}^{m_t} \sqrt{2T_{k,j} d \log(1 + \frac{T}{\lambda d})} \\
&\le \sqrt{2m(T_k - T_{k-1}) d \log(1 + \frac{T}{\lambda d})}
\end{aligned}
\tag{78}
$$

For all phases:

$$
\sum_{k=1}^{s} \sqrt{2m(T_{k+1} - T_k) d \log(1 + \frac{T}{\lambda d})} \le \sqrt{2 \sum_{k=1}^{s} 1 \sum_{k=1}^{s} (T_{k+1} - T_k) m d \log(1 + \frac{T}{\lambda d})}
\tag{79}
$$

$$
\le \sqrt{2m d T \log(T) \log(1 + \frac{T}{\lambda d})}
$$

Similarly, for the second term in $C_{a_t}$:

$$
\begin{aligned}
\sum_{t=T_{k-1}}^{T_k} \sum_{\substack{s \in [t-1] \\ i_s \in \overline{V}_t}} \epsilon_* |\boldsymbol{x}_{a_t}^{\mathrm{T}} \overline{\boldsymbol{M}}_{\overline{V}_{t,t-1}}^{-1} \boldsymbol{x}_{a_s}| &= \sum_{t=T_{k-1}}^{T_k} \sum_{j=1}^{m_t} \mathbb{I}\{i \in \overline{V}_{t,j}\} \sum_{\substack{s \in [t-1] \\ i_s \in \overline{V}_{t,j}}} \epsilon_* |\boldsymbol{x}_{a_t}^{\mathrm{T}} \overline{\boldsymbol{M}}_{\overline{V}_{t,j}}^{-1} \boldsymbol{x}_{a_s}| \\
&\le \epsilon_* \sum_{t=T_{k-1}}^{T_k} \sum_{j=1}^{m_t} \mathbb{I}\{i \in \overline{V}_{t,j}\} \sqrt{\sum_{\substack{s \in [t-1] \\ i_s \in \overline{V}_{t,j}}} 1 \sum_{\substack{s \in [t-1] \\ i_s \in \overline{V}_{t,j}}} |\boldsymbol{x}_{a_t}^T \overline{\boldsymbol{M}}_{\overline{V}_{t,j}}^{-1} \boldsymbol{x}_{a_s}|^2} \\
&\le \epsilon_* \sum_{t=T_{k-1}}^{T_k} \sum_{j=1}^{m_t} \mathbb{I}\{i \in \overline{V}_{t,j}\} \sqrt{T_{k,j} \|\boldsymbol{x}_{a_t}\|^2_{\boldsymbol{M}_{\overline{V}_{t,j}}^{-1}}} \\
&\le \epsilon_* \sum_{t=T_{k-1}}^{T_k} \sqrt{\sum_{j=1}^{m_t} \mathbb{I}\{i \in \overline{V}_{t,j}\} \sum_{j=1}^{m_t} \mathbb{I}\{i \in \overline{V}_{t,j}\} T_{k,j} \|\boldsymbol{x}_{a_t}\|^2_{\boldsymbol{M}_{\overline{V}_{t,j}}^{-1}}} \\
&\le \epsilon_* \sqrt{(T_k - T_{k-1})} \sum_{t=T_{k-1}}^{T_k} \sqrt{\sum_{j=1}^{m_t} \mathbb{I}\{i \in \overline{V}_{t,j}\} \|\boldsymbol{x}_{a_t}\|^2_{\boldsymbol{M}_{\overline{V}_{t,j}}^{-1}}} \\
&\le \epsilon_*(T_k - T_{k-1}) \sqrt{2m d \log(1 + \frac{T}{\lambda d})}
\end{aligned}
\tag{80}
$$

Then for all phases this term can be bounded by $\epsilon_* T \sqrt{2md \log(1 + \frac{T}{\lambda d})}$.

Thus the total regret can be bounded by:

$$
R_t \le 2\sqrt{2mTd \log(T) \log(1 + \frac{T}{\lambda d})} \left( \sqrt{2\log(T) + d \log(1 + \frac{T}{\lambda d})} + 2\sqrt{\lambda} \right)
$$

$$
+ 2\epsilon_* T \sqrt{2md \log(1 + \frac{T}{\lambda d})} + 2\epsilon_* T + 12\epsilon_* \sqrt{\frac{1}{2\tilde{\lambda}_x} T} + \frac{3\epsilon_* \sqrt{2d}}{\tilde{\lambda}_x^{\frac{3}{2}}} T + 2T_1 + u\log(T) + 5
$$

where $T_1 = 16u \log(\frac{u}{\delta}) + 4u \max\{\frac{16}{\tilde{\lambda}_x^2} \log(\frac{8d}{\tilde{\lambda}_x^2 \delta}), \frac{8d}{\tilde{\lambda}_x(\frac{\gamma_1}{6} - \epsilon_* \sqrt{\frac{1}{2\tilde{\lambda}_x}})^2} \log(\frac{u}{\delta})\}$

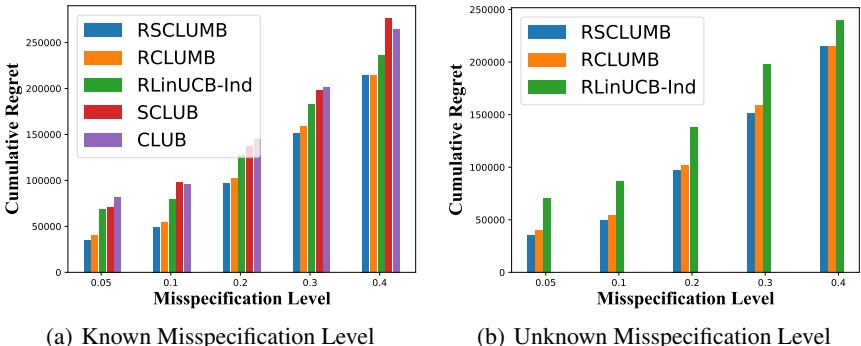

(a) Known Misspecification Level      (b) Unknown Misspecification Level

Figure 2: The cumulative regret of the algorithms under different scales of misspecification level.

## P   More Experiments

For ablation study, we test our algorithms' performance under different scales of deviation. We test RCLUMB and RSCLUMB when $\epsilon^* = 0.05, 0.1, 0.2, 0.3$ and $0.4$ in both misspecification level known and unknown cases. In the known case, we set $\epsilon^*$ according to the real misspecification level, and we compare our algorithms' performance to the baselines except LinUCB and CW-OFUL which perform worst; in the unknown case, we keep $\epsilon^* = 0.2$, and we compare our algorithms to RLinUCB-Ind as only it has the pre-spicified parameter $\epsilon^*$ among the baselines. The results are shown in Fig.2. We plot each algorithm's final cumulative regret under different misspecification levels. All the algorithms' performance get worse when the deviation gets larger, and our two algorithms always perform better than the baselines. Besides, the regrets in the unknown case are only slightly larger than the known case. These results can match our theoretical results and again show our algorithms' effectiveness, as well as verify that our algorithm can handle the unknown misspecification level.

