# OpenReview forum: "Online Clustering of Bandits with Misspecified User Models"
_NeurIPS.cc/2023/Conference — NeurIPS 2023 poster_

### Official Review · Reviewer_TDwD · 2023-07-06

**Soundness:** 3 good
**Presentation:** 3 good
**Contribution:** 3 good
**Rating:** 6
**Confidence:** 4

**Summary:**

This work proposes the clustering bandit problem with misspecified reward functions. It proposes the RCLUMB and RSCLUMB algorithms and their upper bounds match the state-of-the-art ones in some degenerate cases. Numerical results are provided to evaluate the performances of proposed algorithms.



==================

The score is updated to 6 after rebuttal.

**Strengths:**

1. It is good to study the 'combination' of clustering bandits and misspecified bandits, as they are both realistic topics.
2. The paper is well organized, and contributions are mostly well described. For instance, Lemma 5.7 is highlighted as a key result to bound the regret due to misclustering users.
3. The intuition of algorithm design of RCLUMB is in great details and easy to understand. Especially, in Section 4, the steps Cluster Detection, Cluster-based Recommendation, Update User Statistic and Update User Statistics are well explained.
4. Section 5 presents a clear description of the analytical analysis of RCLUMB. In detail, the 'Discussion and Comparison' and proof sckech for Theorem 5.3 are easy to follow.
5. The proposed RCLUMB algorithm is evaluated by numerical experiments in both synthetic and real data sets.

**Weaknesses:**

Overall, I believe this work is challenging and it provides a set of meaningful results and I still here is some possible venues for improvement:
1. It is stated that the CB algorithms which were designed for bandits without misspecification will cause a large regret. I believe it is true with high probability and some discussion is provided in Remark 1, but it would be appreciated if an upper bound on the regret of them in bandits with misspecification can be presented in the paper.
2. A lower bound may help us to learn whether the proposed algorithms are efficient, but is not provided in the work.

**Questions:**

Please refer to the *Weakness* section for questions/suggestions.

---

> ### Author Rebuttal · Authors · 2023-08-09
>
> # Responses to Reviewer TDwD:
>
> Thanks very much for the positive comments for improving our work.
> ## 1. About the regret upper bound of previous CB algorithms under model misspecifications:
> Thanks for giving this suggestion, which would strengthen the claim of theoretical outperformance of our algorithms. We have provided some discussions on this point; please kindly refer to Lines 269-270 and Appendix D: ``Discussions on why Trivially Combining Existing CB and MLB Works Could Not Achieve a Non-vacuous Regret Upper Bound". We also provide the discussions below.
>
> We consider discussing regret upper bounds for CB without considering misspecifications for three cases: (1) neither the clustering process nor the decision process considers misspecifications (previous CB algorithms); (2) the decision process does not consider misspecifications; (3) the clustering process does not consider misspecifications.
>
> For cases (1) and (2), the decision process could contribute to the leading regret. We consider the case where there are $m$ underlying clusters, with each cluster's arrival being $T/m$, and the agent knows the underlying clustering structure. For this case, there exist some instances where the regret upper bound $R(T)$ is strictly larger than $\epsilon_{*}T\sqrt{m\log T}$ asymptotically in $T$. Formally, in the discussion of ``Failure of unmodified algorithm" in Appendix E in [21],
>  they give an example to show that in the single-user case, the regret $R_1(T)$ of the classic linear bandit algorithms without considering misspecifications will have: $\displaystyle\lim_{T \rightarrow + \infty}\frac{R_1(T)}{\epsilon_*T\sqrt{m\log T}}=+ \infty$. In our problem with multiple users and $m$ underlying clusters, even if we know the underlying clustering structure and keep $m$ independent linear bandit algorithms with $T_i$ for the cluster $i \in [m]$ to leverage the common information of clusters, the best we can get is $R_2(T)=\sum_{i \in [m]}R_1(T_i)$. By the above results, if the decision process does not consider misspecifications, we have $\displaystyle\lim_{T \rightarrow + \infty}\frac{R_2(T)}{\epsilon_*T\sqrt{m\log T}}=\displaystyle\lim_{T \rightarrow + \infty}\frac{mR_1(T/m)}{\epsilon_*T\sqrt{m\log T}}=+ \infty$. Recall that the regret upper bound $R(T)$ of our proposed algorithms is of $O(\epsilon_*T\sqrt{md\log T}  + d\sqrt{mT}\log T)$ (thus, we have $\displaystyle\lim_{T \rightarrow + \infty}\frac{R(T)}{\epsilon_*T\sqrt{m\log T}}<+ \infty$), which gives a proof that that the regret upper bound of our proposed algorithms is asymptotically much better than CB algorithms in cases (1)(2).
>
> For case (3), if the clustering process does not use the more tolerant deletion rule in Line 10
>  of Algo.1, the gap between users linked by edges would possibly exceed $\\zeta$ ($
> \zeta = 2\epsilon_{*} \sqrt{\frac{2}{\tilde{\lambda}\_{x}}}$) even after $T_0$, which will result in a regret upper bound no better than $O(\epsilon_*u\sqrt{d}T)$. As the number of users $u$ is usually huge in practice, this result is vacuous. The reasons for getting the above claim are as follows. Even if the clustering process further uses our deletion rule considering misspecifications, and the users linked by edges are within $\zeta$ distance, failing to extract $1$-hop users (Line 5 in Algo.1) would cause the leading $O(\epsilon_*u\sqrt{d}T)$ regret term, as in the worst case, the preference vector $\theta$ of the user in $\tilde{V}\_t$ who is $h$-hop away from user $i_t$ could deviate by $h\zeta$ from $\theta_{i_t}$, where $h$ can be as large as $u$, and it would make the second term in Eq.(8) a $O(\epsilon_*u\sqrt{d}T)$ term. If we completely do not consider the misspecifications in the clustering process, the above user gap between users linked by edges would possibly exceed $\zeta$, which will cause a regret upper bound worse than $O(\epsilon_*u\sqrt{d}T)$.
>
> ## 2. About the lower bound:
>
> We also think a lower bound could help us to learn whether the proposed algorithms are efficient. Therefore, we have provided a lower bound for the CBMUM problem in Theorem 5.4, and we have made some discussions on the optimality of our regret upper bounds. Please kindly refer to Lines 255-262 and Appendix F ``Proof and Discussions of Theorem 5.4" for detailed discussions and proofs of the lower bound. We also list Theorem 5.4, the proof, and the discussions below.
>
>
> **Theorem 5.4** There exists a problem instance for the CBMUM problem such that for any algorithm $R(T)\geq \Omega(\epsilon\_*T\sqrt{d})\,.$
>
> **Proof and discussions**
>
> In the work [21], they give a lower bound for misspecified linear bandits with a single user. The lower bound of $R(T)$ is given by:
> $R_3(T)\geq \epsilon\_* T\sqrt{d}$. Therefore, suppose our problem with multiple users and $m$ underlying clusters where the arrival times are $T_i$ for each cluster, then for any algorithms, even if they know the underlying clustering structure and keep $m$ independent linear bandit algorithms to leverage the common information of clusters, the best they can get is $R(T)=\sum_{i \in [m]}R_3(T_i)\geq \epsilon_*\sum_{i \in [m]}T_i\sqrt{d}=\epsilon_*T\sqrt{d}$, which gives a lower bound of $O(\epsilon_*T\sqrt{d})$ for the CBMUM problem. Recall that the regret upper bound of our algorithms is of $O(\epsilon_*T\sqrt{md\log T}  + d\sqrt{mT}\log T)$, asymptotically matching this lower bound with respect to $T$ up to logarithmic factors and with respect to $m$ up to $O(\sqrt{m})$ factors, showing the tightness of our theoretical results (where
>  $m$ are typically very small for real applications).
>
>  We conjecture that the gap for the $m$ factor is due to the strong assumption that cluster structures are known to prove our lower bound, and whether there exists a tighter lower bound will be left for future work.
>
>  Finally, we thank you again for taking the time to review our paper, the strongly positive feedback, and the valuable suggestions for further improvements.

---

> > ### Comment · Reviewer_TDwD · 2023-08-20
> >
> > Thanks for your response. The score is updated to 6.

---

> > > ### Author Response · Authors · 2023-08-21
> > > **Thanks for your positive feedback**
> > >
> > > Dear Reviewer TDwD,
> > >
> > > Thanks very much for reviewing our response and providing positive feedback. Your dedicated commitment to the review process, positive comments on our work, and valuable suggestions are deeply appreciated.
> > >
> > > Sincerely,
> > >
> > > Authors of Paper 5784.

---

### Official Review · Reviewer_AnqL · 2023-07-07

**Soundness:** 4 excellent
**Presentation:** 4 excellent
**Contribution:** 3 good
**Rating:** 7
**Confidence:** 3

**Summary:**

The paper studies the contextual linear bandit setup with model misspecification in the framework of clustering bandits. This investigation revolves around a particular model misspecification framework, aiming to address the challenge posed by two independent sources of error: model misspecification and the bandit process itself. To tackle this challenge, the paper introduces two algorithms including RCLUMB and RSCLUMB. The former algorithm considers the feature similarity of users as clusters on graphs, while the latter treats it as a set. The authors establish regret upper bounds and present a corresponding lower bound to showcase the theoretical guarantees offered by both the proposed algorithms. Lastly, the paper provides empirical evaluations of the proposed algorithms by employing both synthetic and real-world data. These evaluations serve to validate and assess the effectiveness of the algorithms in practical scenarios.

**Strengths:**

The paper presents a distinctive approach by integrating two previously examined bandit setups, resulting in novel contributions in terms of problem formulation, proposed algorithm, and theoretical insights. The manuscript exhibits exemplary writing, effectively conveying the central idea. While the individual concepts of CB and MLB have been extensively studied, the combination of them as provided in the paper introduces a more practical setup, thereby amplifying the significance of the central problem. The authors of the paper explicitly state the assumptions, which either draw from previously established work or represent a milder version of the same, providing apt discussion concerning the parameters defined in the Theorem, ensuring a comprehensive understanding of their implications. To support the theoretical claims, the manuscript furnishes ample empirical evidence, underscoring the effectiveness of the proposed algorithms.

**Weaknesses:**

Concerns are highlighted with the following points :
1) Comparison between the two proposed algorithms: The main paper does mention two algorithms, but the paper seems to be lacking any discussion surrounding any theoretical comparison between the two. Can they be applied to the same problem? Scenarios to consider for highlighting the pros and cons of using one over another? Sample toy problem to support the claims? Any differences in the meta performance by using set nature over graph nature?
2) Justification of algorithmic choices :
 i) While 1-hop clusters would provide a more close network, they will also lead to lower number of data points. How is the comparison taken into account? It is possible that the increase in data by considering a 2-hop network could be better aligned with the current user. How does the paper handle this possibility?
 ii) Line 72 -- we design a more tolerant edge design algorithm. More tolerant in comparison to?
 iii) In order to ensure the theorem statements, do the model misspecification parameter $\epsilon^*$, noise variance $\sigma^2$, and magnitude of the feature vector need to be in some relation? How is the paper quantifying the relationship between the three?
3) Limitation and Failure: Would really appreciate a discussion on the limitation and failure cases of what the algorithm and problem setup is lacking and future directions of research with literature reference.

**Questions:**

1) Traditional notions of robustness typically involve outliers in data. In the case of the two algorithms mentioned in the paper, which include the term ``Robust'' in the titles, what is robustness tackling here? It is important to clarify what specific aspect of robustness they aim to address in this context.
2) Is there a specific rationale for starting with a complete graph and employing successive elimination rather than an empty graph and successive addition? Intuitively, I would assume the latter approach would yield superior rewards in the initial few rounds, as it would be more focused on the data provided by the sampled arm itself.
3) While the paper engages in a discussion regarding the selection of parameters within the range of (-0.2, 0.2), it is worth exploring other system parameters that influenced this decision-making process. Were there any additional system parameters that contributed to determining the range of parameters chosen?

**Limitations:**

The paper demonstrates commendable discussions concerning the limitations of the theorem statements and assumptions. However, it would be highly beneficial to have similar discussions regarding the problem setup in relation to the existing landscape of works in the bandit domain.
Further, by examining the gaps or areas that have not been sufficiently addressed, and highlighting how the proposed problem formulation fills those gaps, the authors can contextualize their work and reinforce its significance. Additionally, discussing the differences or similarities between the proposed problem setup and the previous approach can contribute to a more comprehensive evaluation of the paper's novelty and the implications of its findings.

---

> ### Author Rebuttal · Authors · 2023-08-09
>
> # Responses to Reviewer AnqL:
> Thanks for taking the time to review our paper, the positive comments, and valuable suggestions. Our responses are as follows.
> ## A. Responses to the weaknesses:
> ### 1. About the comparisons of our proposed algorithms:
> In the clustering of bandits literature, graph-based algorithms [11, 23, 26] and set-based algorithms [3, 25] are two standard approaches. Graph-based CB algorithms and set-based CB algorithms share the same order of regret upper bound, but set-based algorithms usually perform better in empirical evaluations. This is because graph-based algorithms use connected components to represent clusters. As a result, the learner may split two dissimilar users into two clusters only when it cuts every path between these two users. Even when
> the edge between two users is cut, the two users might still need to stay in the same cluster for a long time, making the clustering process longer than the set-based algorithms, which directly split dissimilar users into different clusters. More discussions of these two approaches can be found in [25]. Both of these two approaches are widely studied in CB literature since they share the same theoretical guarantees and similar design ideas. Therefore, to conduct systematic research on the CBMUM problem, we propose both graph-based RCLUMB and set-based RSCLUMB, showing that our ideas and techniques to deal with model misspecifications in CB are pretty general. As shown in Theorem 5.3 and Theorem L.1, both of these two proposed algorithms share the same nearly-optimal order of regret upper bounds. And in Section 6, RSCLUMB performs slightly better than RCLUMB, which is consistent with previous CB works. Both can be applied to the CBMUM problem and achieve outperformance theoretically and numerically. We will add more discussions and comparisons following your valuable advice.
> ### 2. About justification of algorithm choices:
> (i) About the choice of 1-hop:
>
> Thanks for this valuable comment. Theoretically, if we consider the worst-case regret upper bound, we do not need to worry that considering more hops than one hop would be better. In our work, we prove that by using this one-hop extraction, after some iterations, the algorithms can get a ``good partition" (Definition 5.5, which means the inferred cluster contains all the users with the same preference vector and users that are $\zeta$-close) with high probability. If we change the 1-hop to k-hop, then the second term in the regret upper bound shown in Eq.(7) will be k times worse because with k-hop, in the worst case, the algorithms might miscluster users with gaps of $k\zeta$. We will add more illustrations and discussions on this point.
>
> (ii) About the tolerant edge design:
>
> We mean that the edge deletion rule is more tolerant than previous CB works that do not consider model misspecifications [11, 23, 26]. We will make this clearer following your valuable advice.
>
> (iii) About the parameters:
> These parameters need not be in some relation to ensure the theoretical results in the theorem statements. Therefore, following previous works on MLB [10, 21, 27], we do not quantify the relation between them since it will not affect the theoretical results. We will make this point clearer following your suggestion.
>
> ### 3. About the limitation, failure cases, and future directions:
> Thanks very much for this valuable suggestion. One failure case we would think of is when the $\epsilon_*$ is not pre-specified, which could possibly be addressed by incorporating recent model selection methods [10, 27], but it is not the emphasis of this work as we are considering fundamental models on CB and MLB, so we rely on the typical previous assumption on MLB (Assumption 3.4).
>
> For future directions, we have mentioned 3 interesting aspects in the Conclusion (Lines 381-385), we list them here for your convenience: (1) Prove a tighter regret lower bound for CBMUM, (2) Incorporate recent model selection methods into our fundamental framework to design robust algorithms for CBMUM with unknown exact maximum model misspecification level, and (3) Consider the setting
> with misspecifications in the underlying user clustering structure rather than user models.
> ## B. Responses to the questions:
> ### 1. About the ``Robust":
> Thanks for the valuable comment. Following the first work on misspecified linear bandits [13], we use the same word ``robust" to mean that the algorithms are robust to model misspecifications. We will clarify that following your advice.
> ### 2. About starting with an empty graph:
> Thanks for this valuable question. The first work on CB [12] provides a detailed discussion of why we do not start with an empty graph and successive addition. In brief, this approach would fail to achieve a good regret upper bound without the prior knowledge of the minimum gap $\gamma$ between different clusters in Assumption 3.1. For more detailed discussions, please refer to Remark 2 of the paper [12]. We will add some discussions on this point following your advice.
>
> ### 3. About the selection of parameters:
> We select the deviations in the range of (-0.2, 0.2) in experiments is because, according to the work [13], the misspecifications from linearity are usually small, so we choose a relatively small range of deviations to test our algorithms' performance. We have also done some ablation studies on the parameter $\epsilon_*$ in Appendix P. The results clearly show the advantages of our methods in all cases. There are not any additional system parameters that contribute to determining the range of parameters chosen.
>
> Finally, we thank the reviewer again for the positive comments, detailed review, and valuable suggestions.

---

> > ### Comment · Reviewer_AnqL · 2023-08-19
> >
> > I am quite satisfied and happy with the detailed response from the authors. It helped me correct my view on certain aspects of the paper as well.

---

> > > ### Author Response · Authors · 2023-08-20
> > > **Thanks for your positive feedback and kind support**
> > >
> > > Dear Reviewer AnqL,
> > >
> > > We would like to express our sincere thanks for reviewing our response and providing positive feedback. Your dedication to the review process, recognition of our efforts, and valuable suggestions are all deeply appreciated.
> > >
> > > Sincerely,
> > >
> > > Authors of Paper 5784.

---

### Official Review · Reviewer_ovwY · 2023-07-08

**Soundness:** 1 poor
**Presentation:** 1 poor
**Contribution:** 1 poor
**Rating:** 1
**Confidence:** 4

**Summary:**

This paper considers a new bandit problem setting, clustering of bandits with misspecified user models. Here, the misspecified user model means the expected reward does not follow a perfect linear model. Authors give a regret upper bound of $O(\epsilon_*T\sqrt{d\log{T}}+d\sqrt{T}\log{T})$, which is basically a combination of cost from regular CB and the cost from bounded deviation, which is not surprising. And then authors also provide a "desperate" and linear lower bound due to the bounded deviation.

**Strengths:**

The only strength I can find is the lower bound, which conveys the message that this problem do not have any value as it will result in a linear lower bound. If I miss anything, please notify me.

**Weaknesses:**

1. The presentation is poor. i) the notation system is chaotic, ii) lack of description when introducing the algorithm.
2. Lack of novelty in problem setting and result. At least for me, this paper considers an A (CB) + B (MLB) problem, so as the result (upper bound).
3. The proof is, as well, a simple combination of CB and MLB.

**Questions:**

1. Could you provide some recent literature review about CB? The most related work [25] you mentioned was published 4 years ago, and there is a huge gap.
2. In line 123, for the matrix norm, the matrix $M$ should be PD. Otherwise, you should not call it a norm. See ref: https://en.wikipedia.org/wiki/Norm_(mathematics).
3. Could you please improve your notations, for example, in line 131 and 132,  $j(i)$, $i$, and $\ell$? Please check the consistency.
4. In line 133, since it is a contextual bandit problem, can it be extend to infinite arm set?
5. Line 146, what if the user do not come uniformly? For example, for social network, some users are heavy user, whereas some are not.
6. Line 152, I found the assumption of the bounded misspecification is useless, because the most worse case of this misspecification is a uniform distribution. Consequently, it is sub-Gaussian, and then can be absorbed into $\eta$. Please provide explanation.


**Limitations:**

See weakness before.

---

> ### Author Rebuttal · Authors · 2023-08-10
>
> # Responses to Reviewer ovwY
> Thanks for reviewing our paper. Our responses are as follows.
> ## Responses to strengths and weaknesses:
> ### A. About the strengths and novelty of the work:
> We strongly disagree our only strength is the lower bound and the argument that this is a trivial A (CB) + B (MLB) problem. Trivially combining previous CB and MLB works could not achieve a non-vacuous regret in the CBMUM problem. Note that in CB, misspecifications can cause erroneous clustering, which is not considered in previous MLB works. Detailed theoretical analysis of these claims is provided in Appendix D ``Discussions on why Trivially Combining Existing CB and MLB Works Could Not Achieve a Non-vacuous Regret Upper Bound".
>
> RCLUMB introduces 3 novel components to tackle the challenges: (1) extracting 1-hop users (Line 5 in Algo.1); (2) a tolerant edge deletion rule (Line 10 in Algo.1); (3) an enlarged confidence radius (Eq.(5)). We also propose a novel set-based algorithm RSCLUMB (Algo.2). Moreover, our most significant contribution is the novel theoretical analysis with less stringent assumptions. Below are our detailed contributions.
>
> **1. New problem formulation.** We are the first to study the CBMUM problem, which is more practical by removing the perfect linearity assumption in CB.
>
> **2. Novel algorithm designs.**
> We discuss the novelty of the above (1)-(3) key components in detail:
>
> (1) Extracting 1-hop users can avoid estimation error aggregation. Due to the estimation inaccuracy caused by deviations, users connected in the graph may not have exactly the same preferences. Following previous CB works to use connected components to form clusters would cause estimation inaccuracy aggregation in clusters. Detailed discussions are in ``Cluster Detection" in Section 4.
>
> (2)  Without the tolerant edge deletion rule considering deviations, the algorithm would delete edges erroneously, causing inaccurate user clustering. Theoretical details are in Appendix H.
>
> (3) The UCB index in previous CB works only considers the exploration bonus and would fail to fully capture the uncertainty in CBMUM. Therefore, we design a new UCB index in Eq.(5) incorporating additional uncertainties from misspecifications. Details can be found in Lemma 5.6 and Appendix I.
>
> **3. Novel theoretical analysis (most challenging).** This is our most significant contribution. Our proof techniques are novel in CB and MLB. The main challenge of the regret analysis in CBMUM is that due to the estimation inaccuracy caused by misspecifications, it is impossible to cluster all users exactly correctly, and it is non-trivial to bound the regret caused by misclustering  $\zeta$ -close users, whose techniques can not be found in any previous works on CB and MLB. Please refer to Appendix C ``Highlight of the Theoretical Analysis" for details.
>
> **4. Less stringent assumption.** We move a stringent assumption on the arm distribution in all previous CB works, which is an important novel contribution to CB research. We propose a relaxed assumption (Assumption 3.3) by removing the condition that the variance must be bounded by $\frac{\lambda^2}{8\log (4C)}$. Details can be found in Appendix B.1 and J.
>
>
> **5. Experimental outperformance.** Extensive experiments show superior performances of our algorithms.
>
> ### B. About the lower bound:
> We strongly disagree a linear lower bound means the problem has no value. In MLB literature, the lower bound is linear [21], but MLB is still an important hot topic (see the two recent works below), disproving the claim.
>
> (1) Does Sparsity Help in Learning Misspecified Linear Bandits? ICML 2023
>
> (2) No-Regret Linear Bandits beyond Realizability. UAI 2023
> ### C. About the presentation:
> We respectfully disagree that our presentation is poor. We have enough descriptions of our algorithm in Sections 1 and 4. We will further polish the writing in the final version.
> ## Responses to the questions:
> ### A. About the literature review about CB:
> We kindly remind you that the most recent CB works we mentioned are [3] and [26] (published in 2021 and 2022). To the best of our knowledge, no CB works consider model misspecifications. We will add more related CB works in the final version following your advice.
> ### B. About the matrix norm in Line 123:
> Thanks for the detailed comment. We will revise it to be just a notation instead of calling it a norm.
> ### C. About the notations:
> We will check the notations. However, in Lines 131 and 132, $j(i)$ denotes the index of the ground-truth cluster of user $i$; $i$ and $l$ denote two different users; we do not think there exists any inconsistency.
> ### D. About infinite arm set:
> Yes, our results can be directly extended to the infinite arm set since the regret bound in Theorem 5.3 does not depend on the size of the arm set but only depends on d.
> ### E. About general arrival distribution:
> Please refer to the discussions in Lines 156-158 and Appendix B.3. The uniform arrival only affects the $T_0$ term. For any arrival distribution with $p_{min}$ the minimal arrival probability of a user, $T_0$ becomes $O(1/p_{min} \log T)$. Since it is a lower-order term, it will not affect the main order of our regret bound.
> ### F. Explanation on the difference between misspecifications and noises:
> We disagree the assumption of the bounded misspecification is useless. This is a standard assumption in the MLB literature [10, 21, 27]. The uniform distribution is the best rather than the worst case, as it makes the expectation be 0 and can thus be absorbed into the sub-Gaussian noise, making the expected reward models perfectly linear. In general, the expectations of deviations are not 0 and can not be regarded as noises.
>
> Finally, thanks again for your effort in reviewing our paper and giving some advice. We hope our clarifications and explanations are clear and well address your concerns.

---

> > ### Comment · Reviewer_ovwY · 2023-08-11
> >
> > I just want to mention one thing, the regret bound $O(\epsilon_*T\sqrt{md\log{T}} + \textup{some not important thing})$. This is even worse than $O(T)$, $\textbf{again, a regret bound worse than}$ $O(T)$? (you should write $O(T)$ please!) Sorry, I just cannot find any value of this paper. I will lower my rating.

---

> > > ### Author Response · Authors · 2023-08-11
> > >
> > > First, we would like to kindly remind you that in your comment, you add an additional $O(\sqrt{T})$, which does not appear in our regret upper bound in Theorem 5.3.
> > >
> > > Second, though we agree that a linear regret upper bound may be vacuous in most other bandit research fields, we
> > > respectfully disagree that an upper bound with $O(T)$ term has no value in the research field of bandits with misspecifications. If you are familiar with the literature on misspecified settings and difficulties, you will notice that previous works on the classic misspecified linear bandits (MLB) setting all have a common linear term of $\widetilde{O}(\epsilon_*\sqrt{d}T)$ that is thought of as the price of misspecification [1, 2, 3, 4], which is inevitable as supported by the lower bound given in [1]. Apart from MLB, the works on misspecified Gaussian process bandits also have a linear term of $O(\epsilon_* T\sqrt{\gamma_T})$, and a lower bound argument also shows that the linear regret is unavoidable [5]. We also prove a lower bound for CBMUM, and our upper bound asymptotically nearly matches this lower bound, indicating that we achieve the nearly optimal result in the challenging CBMUM problem.
> > > In summary, as supported by extensive works, a linear term related to the model misspecification is necessary for bandits with misspecifications; therefore, we respectfully disagree that our regret upper bound with a linear term means our work is worth nothing.
> > >
> > > We sincerely appreciate your taking the time to review our paper and give us some comments. We hope our explanations are clear and can address your concerns.
> > >
> > > References
> > >
> > > [1] Learning with Good Feature Representations in Bandits and in RL with a Generative Model. ICML 2020
> > >
> > > [2] Adapting to Misspecification in Contextual Bandits with Offline Regression Oracles. ICML 2021
> > >
> > > [3] A Parameter-Free Algorithm for Misspecified Linear Contextual Bandits. AISTATS 2021
> > >
> > > [4] Adapting to misspecification in contextual bandits. Neurips 2020
> > >
> > > [5] Misspecified Gaussian process bandit optimization. Neurips 2021

---

> > > ### Comment · Reviewer_ANNG · 2023-08-11
> > > **The regret is sub-linear in the absence of mis-specification, and is only linear in the case of mis-specification (\epsilon > 0). That is the expected behaviour.**
> > >
> > > The bound shown in the paper  attains a linear bound in the presence of mis-specification while attaining sub-linear bound in the well specified case. There are lower bounds showing that linear regret cannot be avoided in the case of mis-specified bandits. So, in this light, I am not sure I follow the criticism by reviewer ovwY that the regret bound is weak.

---

> > > > ### Author Response · Authors · 2023-08-12
> > > >
> > > > Dear Reviewer ANNG,
> > > >
> > > > Thanks very much for kindly helping us make further clarification. We really appreciate your support.
> > > >
> > > > Sincerely,
> > > >
> > > > Authors of Paper 5784.

---

### Official Review · Reviewer_ANNG · 2023-07-10

**Soundness:** 3 good
**Presentation:** 3 good
**Contribution:** 3 good
**Rating:** 7
**Confidence:** 3

**Summary:**

This paper studies multi-agent linear bandits with cluster-structure and mis-specification. At each time, an user is chosen independently and an arm from a finite set of arms is to be recommended for this user. Subsequently, the decision maker observes a noisy and mis-specified reward. The users are assumed to have an underlying latent cluster structure with well separated clusters.

The goal of the decision maker is to minimize the total regret of all the decisions made over time. The main result of the paper is an algorithm in which the decision maker has a running estimate of the clustering of users which is used to make recommendations. The regret bound scales sub-linearly in the number of agents (which indicates the learning algorithm is able to exploit the underlying cluster structure) and optimally in the mis-specification bound.

**Strengths:**

1) The paper combines in a non-trivial manner, works on mis-specified linear bandits and clustered multi-agent bandits.

2) Technically, the paper moves past a restrictive technical assumption on the distribution of the arms first adopted by Gentile at.al. and followed by all subsequent works. This is a good contribution just in the study of clustered multi-agent bandits.

**Weaknesses:**

The only weakness is that the mis-specification upper bound \epsilon^* is both
1. needed for the algorithm as an input, and
2. The regret scales as this upper bound \epsilon^* as opposed to the true unknown mis-specification upper bound of the problem.



**Questions:**

Comment on the weakness in terms of knowledge of \epsilon^* can help elucidate why the assumption is needed or is non-trivial to overcome.

---

> ### Author Rebuttal · Authors · 2023-08-09
>
> # Responses to Reviewer ANNG:
> We are very grateful for your strongly positive comments and valuable suggestions. Below are our responses.
> ## 1. About the misspecification upper bound:
> Thanks for your valuable suggestions for adding more comments and discussion of the known misspecification upper bound. We will add more discussions for further improvement.
>
> The Assumption 3.4 of a pre-specified maximum misspecification level $\epsilon_*$ follows the standard work on misspecified linear bandits [21]. This $\epsilon_*$ can be an upper bound on the maximum misspecification level, not the exact maximum itself. In real-world applications, the deviations are usually small [13], and we can set a relatively big $\epsilon_*$ (e.g., 0.2) to be the upper bound. Our experimental results support this claim. As shown in our experimental results on real-data case 2 (Lines 363-365), even when $\epsilon_*$ is unknown, our algorithms still perform well by setting $\epsilon_* = 0.2$. We have done more experiments on the Movielens dataset with unknown $\epsilon_*$. The results can be found in Figure 1 in the global PDF. We can clearly notice that even when the exact maximum is unknown, we can choose relatively small $\epsilon_*$, and our proposed RCLUMB outperforms the baselines with all chosen small values of $\epsilon_*$ in real-world applications.
>
> Some recent studies [27, 10] use model selection methods to theoretically deal with an unknown exact maximum misspecification level in the single-user case, which is not the emphasis of this work. The work [10] also assumes that the learning agent has access to a regression oracle. And for the work [27], though their regret bound depends on the exact maximum misspecification level that needs not to be known by the agent, an upper bound of the exact maximum misspecification level is still needed. We are the first to initialize the study of the important CBMUM problem, and propose a general
> framework for dealing with model misspecifications in CB problems. Our study is based on fundamental models on CB [11, 25] and MLB [21], the algorithm design ideas and theoretical analysis are pretty general. We leave incorporating their methods to deal with an unknown exact maximum misspecification level but achieve the regret upper bound depending on the exact maximum as an interesting future work.
>
> Finally, thank you again for taking time to carefully review our paper, the strong appreaciation of our work, and valuable suggestions.

---

> > ### Comment · Reviewer_ANNG · 2023-08-11
> > **Thank you for your response.**
> >
> > I would encourage the authors to add the discussion from this response in the revised version of the draft.
> >
> > In addition, I will also encourage the authors to state that they do NOT need a restrictive assumption on the conditional variance that Gentile et.al. (2014) require which all subsequent works on clustering of multi-agent bandits employ. **It was shown by Lattimore et.al., that no distribution can  satisfy the conditional variance assumption!** In this light, moving past that assumption can be bubbled up as a contribution of the paper.

---

> > > ### Author Response · Authors · 2023-08-12
> > > **Thank you for your strong support and appreciation**
> > >
> > > Dear Reviewer ANNG,
> > >
> > > Thanks very much for your valuable suggestions, and for your kind help in further highlighting the significance of our contributions to removing the previous restrictive assumption in the clustering of bandits literature. We will add more discussions in the final version for further improvement following your advice. Finally, thank you again for your strong support and appreciation.
> > >
> > > Sincerely,
> > >
> > > Authors of Paper 5784.

---

### Official Review · Reviewer_oJ3g · 2023-07-11

**Soundness:** 3 good
**Presentation:** 2 fair
**Contribution:** 3 good
**Rating:** 6
**Confidence:** 3

**Summary:**

This paper proposes the clustering of bandits with misspecified user models problem. Clustering of bandits combines collaborative filtering with standard contextual bandits in order to cluster users into groups which share a common reward $\theta$. This paper modifies the setting by allowing users to have deviation vectors $ \epsilon^{i_t, t}$ so that the reward is $r_t = x_{a_t}^T\theta_{i_t} + \epsilon_{a_t}^{i_t, t} + \eta_t$. They then propose two algorithms for this setting, prove their performance, and do experiments to show the improved results.

**Strengths:**

1) Model misspecification is a problem in online clustering bandits
2) Experimental comparison is quite favorable towards the developed algorithms
3) The developed algorithms have proven regret guarantees

**Weaknesses:**

1) ~~RSCLUMB is not well motivated~~
2) Writing is a bit tricky to follow at times. For example, lines 309-310 are unclear to me are we moving the $\|\theta_{i_s} - \theta_{i_t}\|_2$ out and bounding it by the term on line 310? Or are we using the term from Lemma 5.7? Is the term from 5.7 substantially better?
3) ~~The experiments are missing the baseline of learning a reward $\theta$ for each user separately~~

**Questions:**

1) Does $\epsilon_{a}^{i_t, t}$ vary over time? The $t$ superscript and line 141 make it seem like it does, but it's not clear to me how the algorithm could learn a user's deviation vector which is unknown and different at each timestep. If it varies over time, how does  $\epsilon_{a}^{i_t, t}$ differ from the noise $\eta_t$?
2) Why do we talk about RCLUMB and RSCLUMB?
3) How do your algorithms compare to the baseline of just learning a reward $\theta$ for each user separately?

**Limitations:**

The authors have not discussed the limitations, though they are similar to those for other bandit papers

---

> ### Author Rebuttal · Authors · 2023-08-09
>
> # Responses to Reviewer oJ3g:
>
> Thanks for reviewing our paper. Our responses are listed below.
> ## A. Reponses to the weaknesses:
> ### 1. About the RSCLUMB:
> In the clustering of bandits literature, graph-based algorithms [11, 23, 26] and set-based algorithms [3, 25] are two standard approaches. Graph-based CB algorithms and set-based CB algorithms share the same order of regret upper bound, but set-based algorithms usually perform better in empirical evaluations. This is because graph-based algorithms use connected components to represent clusters. As a result, the learner may split two dissimilar users into two clusters only
> when it cuts every path between these two users. Even when
> the edge between two users is
> cut, the two users might still need to stay in the same cluster
> for a long time, making the clustering process longer than the set-based algorithms, which directly split dissimilar users into different clusters. More discussions of these two approaches can be found in [25]. Both of these two approaches are widely studied in CB literature since they share the same theoretical guarantees and similar design ideas. Therefore, to conduct systematic research on the CBMUM problem, we propose both graph-based RCLUMB and set-based RSCLUMB, showing that our ideas and techniques to deal with model misspecifications in CB are pretty general. We have some discussions on the motivation for proposing RSCLUMB in Lines 165-167 and Appendix K. We will add more discussions and illustrations following your valuable advice.
> ### 2. About the writing:
> In Lines 309-310, what we mean is that to upper bound the extra term in Line 308 (\\[\left|x_a^{\top}\overline{M}\_{\overline\{V}\_t,t-1}^{-1}\sum\_{s\in[t-1]\atop i_s\in \overline{V}\_t}x_{a_s}x_{a_s}^{\top}(\theta_{i_s}-\theta\_{i_t})\right|\\]) with condition $\left\\|\theta_{\ell}-\theta_{i_t}\right\\|\_2\\leq\zeta\,,\forall{\ell\in \overline{V}\_t}$, we can not trivially bound the term by \\[\left|x_a^{\top}\overline{M}\_{\overline{V}\_t,t-1}^{-1}\sum\_{s\in[t-1]\atop i_s\in \overline{V}\_t}x_{a_s}x_{a_s}^{\top}(\theta_{i_s}-\theta_{i_t})\right|\leq\left\\|x_a^{\top}\overline{M}\_{\overline{V}\_t,t-1}^{-1}\sum\_{s\in[t-1]\atop i_s\in \overline{V}\_t}x_{a_s}x_{a_s}^{\top}\right\\|_2\times \zeta\\], which is wrong and we should not do that. Therefore, we prove Lemma 5.7 with subtle theoretical analysis to bound the term. We will add more explanations to make it clearer.
> ### 3. About the baseline of learning a reward for each user separately:
>
> We kindly remind you that we have compared two baselines of learning a reward for each user separately: LinUCB-Ind and RLinUCB-Ind. LinUCB-Ind is the baseline that uses LinUCB to learn a reward for each user independently, and RLinUCB-Ind is the baseline that uses the RLinUCB that is robust to model misspecifications in the single-user case to learn a reward for each user independently. Please kindly refer to Lines 319-321, Figure 1, and all the discussions of experimental results.
>
> ## B. Responses to questions:
>
> ### 1. About the deviation:
> We use the t superscript to represent that the model deviations can be either time-varying or fixed; as long as the deviations are bounded, whether they are fixed or time-varying will not affect the algorithm designs and theoretical results, showing that the generalizability of our proposed methods. In misspecified linear bandits, the algorithms try to learn the $\theta$ accurately with the existence of deviations
>  instead of learning the deviation vectors [10, 13, 21, 27]. We kindly remind you that when the deviations vary with time, they are still completely different from the subgaussian noises. The reason is that the subgaussian noises must satisfy that the expectations are 0; however, the expectations of the deviations do not need to be 0, as long as they are not 0, they would make the expected reward model perturbed from linearity. The only common thing is that they can be time-varying. Hope this clarification addresses your concerns.
>
> ### 2. About RCLUMB and RSCLUMB:
> Please kindly refer to our response to the weakness above.
>
> ### 3. About the baseline of learning a reward for each user separately:
> Please kindly refer to our response to the weakness above.
>
> Finally, we thank you again for your effort in reviewing our paper and giving some comments and questions. We hope our clarifications and explanations are clear and well address your concerns.

---

> > ### Comment · Reviewer_oJ3g · 2023-08-15
> > **Response to Rebuttal**
> >
> > Ah, thanks! I had misunderstood some important aspects of the paper, and I found this response to be helpful. In light of this, I am revising my previous score.
> >
> > I would now summarize the core contribution of this paper as showing that where Clustering Bandits previously grouped users in order to pool data and get higher reward sooner, this paper shows that grouping strategies can also in fact help to overcome model misspecification, and attain a regret bound that agrees with the lower bound on misspecified bandit problems up to a logarithmic factor.

---

> > > ### Author Response · Authors · 2023-08-16
> > > **Thanks for your positive feedback**
> > >
> > > Dear Reviewer oJ3g,
> > >
> > > We are pleased to learn that our response effectively addressed your concerns. Your decision to adjust the rating is genuinely appreciated, and we wish to express our sincere gratitude for acknowledging the merit of our work.
> > >
> > > Finally, we want to extend our heartfelt thanks for your dedicated time and effort in reviewing our paper.
> > >
> > > Sincerely,
> > >
> > > Authors of Paper 5784.

---

### Official Review · Reviewer_1JFu · 2023-07-11

**Soundness:** 2 fair
**Presentation:** 3 good
**Contribution:** 2 fair
**Rating:** 3
**Confidence:** 4

**Summary:**

This paper introduces the problem of clustering of bandits with misspecified user models (CBMUM), where the expected rewards in user models can deviate from perfect linear models. The paper presents two robust CB algorithms, RCLUMB and RSCLUMB, that aim to deal with inaccurate user preference estimations and erroneous clustering caused by model misspecifications. The regret upper bounds of the algorithms are provided under introduced assumptions, and experiments show their outperformance over previous linear bandit algorithms.

**Strengths:**

- The paper studied a practical and important problem of clustering of bandits with misspecified user models, which is a more realistic scenario than assuming perfect linearity.
- The theoretical analysis provides regret upper bounds for the algorithms under introduced assumptions.

**Weaknesses:**

- The paper's theoretical results are significantly undermined by the inclusion of impractical assumptions. The analysis relies on the existence of a pre-specified maximum misspecification level, denoted as $\epsilon^*$, which is used as a hyper-parameter. However, in real-world scenarios, such prior knowledge is generally unavailable. Furthermore, the paper fails to offer any insights on how to determine this parameter and how it affects the model. As a result, the practical relevance and applicability of the findings are considerably weakened.
- The motivation behind the proposed algorithm is insufficient. The paper suggests clustering bandits with misspecified user models as a solution to handle situations where the reward function does not necessarily adhere to linearity. However, it fails to discuss the extensive research conducted on non-linear bandits in recent years, as mentioned in [1,2]. Moreover, the existing work on clustering in the non-linear setting, as discussed in [3], significantly undermines the perceived contribution and novelty of this paper.
- The experimental evaluation is limited. Only 1000 items and users are selected to perform the experiments, which is far from practical. The paper also fails to compare the proposed clustering algorithm with other recommender system algorithms [4]. Furthermore, the paper lacks an analysis of the learned clustering strategy, which could offer valuable insights into the practical pattern of clustering. Including such comparisons and analyses would enhance the overall quality and applicability of the experimental evaluation.

[1] Neural Contextual Bandits with UCB-based Exploration. ICML 2020

[2] Neural Thompson Sampling. ICLR 2021

[3] Neural Collaborative Filtering Bandits via Meta-Learning.

[4] Neural Collaborative Filtering. WWW 2017


/****************/

I couldn't include the authors in the comments below, so I've commented here. Thanks for the rebuttal. After carefully reading the points and others’ reviews, I‘m still not satisfied with the motivation, experimental evaluation, and the empirical selected misspecification level. For the motivation, the main benefit of the proposed bandit among neural bandits to handle non-linearity is efficiency. However, the efficiency of the proposed algorithm among other linear or non-linear bandits is not verified in the paper. The scalability of the used datasets is also quite small (maximum 10000 users only), which is difficult to support the claim of efficient. For the experimental evaluation, despite the dataset size, the choice to only rank a subset of items of each user is far from practical. The discussion with the recommender system literature is also insufficient as the online recommendation with the same setting are ignored. Furthermore, the significance of deriving the bound involving the term O(T) does not appear substantial. My rating will remain unchanged.

**Questions:**

Please refer to the weaknesses I mentioned above.

**Limitations:**

I'm not aware of any potential negative societal impact in the paper.

---

> ### Author Rebuttal · Authors · 2023-08-09
>
> # Responses to Reviewer 1JFu:
> Thanks for reviewing our paper and giving some suggestions. Below are our responses.
> ## 1. About the maximum misspecification level:
> Assumption 3.4 of a pre-specified max misspecification level $\epsilon_*$ follows the standard work in MLB literature [21]. $\epsilon_*$ can be an upper bound on the max misspecification level, not the exact maximum. In real-world applications, the deviations are usually small [13], and we can set a relatively big $\epsilon_*$ (e.g., 0.2) to be the upper bound. Our experimental results on real-data case 2 (Lines 363-365) show that even when $\epsilon_*$ is unknown, our algorithms still perform well by setting $\epsilon_* = 0.2$. We have done more experiments on the Movielens data with unknown $\epsilon_*$. The results can be found in Figure 1 in the global PDF. We can notice that if the exact maximum is unknown, we can choose some small $\epsilon_*$, and our RCLUMB outperforms the baselines with all chosen small values of $\epsilon_*$ on the real-world data.
>
> Some recent studies [27, 10] use model selections to deal with an unknown exact max misspecification level in the single-user case, which is not the emphasis of this work. The work [10] assumes access to a regression oracle; though the regret bound in [27] depends on the unknown exact max misspecification level, an upper bound of the exact max misspecification level is still needed. We are the first to initialize the study of the important CBMUM problem, and propose a general
> framework for dealing with misspecifications in CB. Our study is based on fundamental models on CB [11, 25] and MLB [21]; the algorithm designs and theoretical analysis are pretty general. We leave incorporating model selections to deal with unknown $\epsilon_*$ as an interesting future work.
>
> We have many discussions on $\epsilon_*$ (Lines 114-118, 158-160, 341-343, 363-366, and Appendix B.2). We also provide ablation studies with respect to $\epsilon_*$ in Appendix P.
>
>
> ## 2. About the distinctions of misspecified linear bandit (MLB) from neural bandit (NB):
>
> We agree that MLB and NB are both non-linear. However, we strongly disagree that these two lines of research are conflicted. We will cite these works, but we do not think they can undermine our contributions to any extent. Reasons are as follows.
>
> First, the settings are completely different. In MLB, reward models are almost linear with slight deviations. In many real-world applications, the reward models are indeed close to linearity [13, 21]. Therefore, MLB is an important and practical problem. On the other hand, NB aims to deal with situations where reward models are very complex and far from linearity. Though both are non-linear, they are two distinct research fields.
>
> Second, we emphasize NB's computation costs are very high. Therefore, all previous works on MLB [10, 13, 21, 27] do not consider NB because they are unnecessary and computationally heavy. Also, many recent works below on MLB contradict the claim that MLB lacks sufficient motivation given NB.
>
> (1) Does Sparsity Help in Learning Misspecified Linear Bandits? ICML 2023
>
> (2) No-Regret Linear Bandits beyond Realizability. UAI 2023
>
> ## 3. About the differences between [3] and ours:
> We claim the work [3] given by the reviewer can not undermine our contributions. First, as mentioned above, the setting of [3] is very different, and our methods are far more computationally efficient in CBMUM. Second, their theory relies on an additional ``Arm Separability" assumption on over-parameterized neural networks, which is not needed in ours. Further, their regret upper bound has a term of $\tilde{O}(\sqrt{uT})$, where $u$ is the number of users that can be huge, whereas the corresponding term in ours is $\tilde{O}(\sqrt{mT})$ ($m\ll u$ is the number of clusters), significantly less than $\tilde{O}(\sqrt{uT})$. Our contributions to the theoretical analysis in CB are significant and pretty general (detailed in Appendix C), which are not covered in [3].
>
> ## 4. About the experiments:
> First, we would like to stress our most significant contributions are theoretical analysis and algorithm designs. Second, our experimental evaluations are comparable to previous works in CB literature.
> ### 4.1 Size of the dataset:
> In CB literature, the data sizes in most works are not large and close to ours [23, 25, 26]. Following these works, we extract a proportion of the large datasets to be the data for experiments. The experiments are mainly used to verify the theoretical results. We agree performing experiments with a larger dataset would better reflect our outperformance.  We have done some experiments on an enlarged dataset extracted from Yelp (10,000 users and items). The results are shown in Figure 2 in the global PDF. Our algorithm outperforms the well-performed methods among all baselines on this larger dataset. We will conduct more experiments on larger data in a later version following your advice.
>
>
> ### 4.2 About the baseline in [4]:
> The baseline in [4] is completely an offline algorithm, lying on offline training of deep neural networks, and can not be applied in the online bandit setting. None of the previous CB works [11, 12, 23, 25, 26] compare with this baseline. We will cite it for discussions, but we respectfully disagree that including it as a baseline is necessary.
> ### 4.3 About the analysis of the learned clustering strategy:
> In Lemma H.1, we give a theoretical guarantee showing that after some iterations $T_0$, our algorithms can learn a good clustering strategy (``good partition"). Following previous CB works [11, 12, 23, 25, 26], we do not conduct experiments on the learned clustering strategy. We agree that adding more experimental results on the clustering strategy would give a better understanding. We will consider adding them in a future version.
>
> Finally, thanks again for taking the time to review our paper. We hope our clarifications are clear and address your concerns.

---

### Author Rebuttal · Authors · 2023-08-10

We appreciate all reviewers' effort in reviewing our paper and giving some comments and questions. We have done some experiments for your reference, please refer to the global pdf.

---

### Decision · Program_Chairs · 2023-09-21

**Decision:**

Accept (poster)

**Comment:**

Based on the discussions, the reviewers acknowledged that the main strengths were:

- The paper relaxes some assumption in prior Cluster of Bandits works
- The combination of techniques for Cluster of Bandits and Misspecified Linear Bandits is nontrivial
- A lower bound is given

The main remaining concerns were:
- Motivation: why considering using misspecified linear bandits instead of nonlinear bandits
- Dataset size in evaluation: experiments with 10000 users may not be large-scale enough
- Algorithm needs to know upper bound of misspecification level

Reviewers ovwY and 1JFu had some concerns on the O(T) term in the regret bound. However, from the AC (and Reviewer ANNG's) point of view, a $\epsilon \sqrt{d} T$-like term in the regret bound is totally expected in misspecified linear bandits. First, it is a standard term that appears in single-task misspecified linear bandit settings; second, such bound is meaningful: it can be significantly smaller than the trivial bound of O(R T), where R is the range of the expected reward.

Given the overall feedback from the reviewers, it seems that the main strengths outweigh the concerns.